# Empty Shelves or Lost Keys? Recall Is the Bottleneck for Parametric Factuality

**Nitay Calderon** [1 2]  **Eyal Ben-David** [1]  **Zorik Gekhman** [1]  **Eran Ofek** [1]  **Gal Yona** [1]

## Abstract

Standard factuality evaluations of LLMs treat all errors alike, obscuring whether failures arise from missing knowledge (empty shelves) or from limited access to encoded facts (lost keys). We propose a behavioral framework that profiles factual knowledge at the level of facts rather than questions, characterizing each fact by whether it is encoded, and then by how accessible it is: cannot be recalled, can be directly recalled, or can only be recalled with inference-time computation (thinking). To support such profiling, we introduce WikiProfile, a new benchmark constructed via an automated pipeline with a prompted LLM grounded in web search. Across 4 million responses from 13 LLMs, we find that encoding is nearly saturated in frontier models on our benchmark, with GPT-5 and Gemini-3 encoding 95–98% of facts. However, recall remains a major bottleneck: many errors previously attributed to missing knowledge instead stem from failures to access it. These failures are systematic and disproportionately affect long-tail facts and reverse questions. Finally, we show that thinking improves recall and can recover a substantial fraction of failures, indicating that future gains may rely less on scaling and more on methods that improve how models utilize what they already encode.

## 1. Introduction

Large Language Models (LLMs) encode a vast amount of factual information in their parameters (Joshi et al., 2017; Petroni et al., 2019; Kwiatkowski et al., 2019; Mallen et al., 2023), yet factual errors remain a persistent challenge (Wei et al., 2024a; Haas et al., 2025; Ravichander et al., 2025). While existing evaluations consider a response as either correct or incorrect, accuracy alone provides little insight

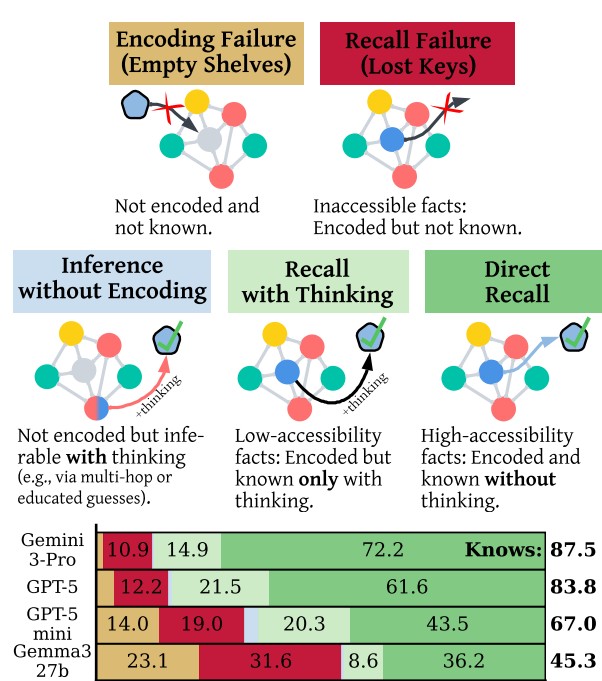

*Figure 1.* **Top:** We propose five knowledge profiles that characterize facts. **Bottom:** Percentages of these profiles across selected LLMs, revealing: (1) Scaling fills "empty shelves" by reducing encoding failures: **frontier LLMs encode nearly all facts** in our data. (2) **Recall failures remain abundant** despite scaling, leaving substantial room for improvement. (3) **Thinking acts as a recovery mechanism** of facts that would otherwise remain *"lost"*.

into the source of the error. If an LLM incorrectly answers the question *"Which famous band played their first gig at the Boardwalk club?"* (the answer is Oasis), can we tell why? One possibility is that the LLM *does not encode* this information in its parameters (e.g., due to limited data coverage or model capacity). Alternatively, a fact may be *encoded but not accessible* under different conditions than how it was learned. One example is the reversal curse (Berglund et al., 2024; Lin et al., 2024c), where an LLM can answer that Oasis played their first gig at the Boardwalk club, yet fails to answer who played their first gig there.

Encoding and recall failures are indistinguishable under accuracy metrics, yet they imply different limitations and solutions. Encoding failures call for pre-training interventions, such as scaling model size or data coverage. Recall failures suggest post-training interventions that often im-

---

[1]Google Research    [2]Technion – Israel Institute of Technology Correspondence to: Nitay Calderon <nitayc@google.com>, Gal Yona <galyona@google.com>. We will announce the release of WIKIPROFILE and add a link.

*Proceedings of the $43^{rd}$ International Conference on Machine Learning*, Seoul, South Korea. PMLR 306, 2026. Copyright 2026 by the author(s).

prove how models utilize what they already encode (Zhou et al., 2023; Gekhman et al., 2024; Lin et al., 2024a).

We therefore propose shifting the unit of analysis from individual questions to *facts*. We introduce *knowledge profiling*: a framework that categorizes facts into one of five profiles (see Figure 1), based on whether the fact is encoded, and by how accessible it is: cannot be recalled, can be directly recalled, or can be recalled only with inference-time computation (thinking). We use *thinking* to refer to inference-time techniques that elicit intermediate computations before the final answer, including both chain-of-thought (CoT) prompting and dedicated thought tokens generated by thinking-optimized LLMs. We operationalize the definitions of encoding and recall behaviorally (Holtzman & Tan, 2025). We measure *encoding* by priming the LLM to complete a factual proposition using a context that resembles how the fact was likely to be encountered during pre-training. We measure *recall*[1] by asking questions about these encoded facts across different contexts, including phrasings or relational orderings. This enables the study of frontier LLMs despite the inaccessibility of their weights and training data.

To support knowledge profiling, we introduce WIKIPRO-FILE, a new benchmark comprising 2,150 facts, each paired with 10 questions that probe encoding, recall, and recognition (multiple-choice). WIKIPROFILE is constructed via a fully automated multi-step pipeline that employs a prompted LLM grounded in web search to generate, refine, and verify questions. Unlike other benchmarks that rely on synthetic or schema-restricted knowledge-base triplets (Luo et al., 2023; Kumar et al., 2024; Yuan et al., 2024; Wu et al., 2025b), our facts are extracted from natural documents. This enables evaluation in more realistic settings, where facts are expressed indirectly, and may involve complex, context-dependent relations (Wang et al., 2025b).

Using WIKIPROFILE, we evaluate 13 LLMs, resulting in more than 4 million graded answers. We find that scaling primarily improves encoding, while recall remains a major bottleneck. For frontier LLMs, including Gemini-3-Pro and GPT-5, encoding is nearly saturated, with 95–98% of facts encoded. Yet the same models still cannot recall 25–33% of facts without thinking. In addition, we show that scaling model size alone increases what the model encodes but does not necessarily improve its recall.

Why do LLMs struggle to access knowledge they have encoded? Our findings suggest that recall is tightly coupled to the conditions under which facts were learned, degrading when queries diverge from training-time patterns. We manifest this through two cases: fact popularity and the reversal curse. While the encoding gap between popular and

long-tail facts for frontier LLMs is small (a few percentage points), the recall gap is substantially larger (often exceeding 25%). Reverse questions reveal a surprising pattern: LLMs struggle to answer reverse questions compared to direct ones, despite showing no disadvantage in recognizing the correct answer under multiple-choice evaluation.

Together, our results refine prior accounts of long-tail errors and the reversal curse (Kandpal et al., 2023; Berglund et al., 2024; Mallen et al., 2023; He et al., 2025). Whereas these failures are often interpreted as limitations of memorization or bidirectional encoding, our results suggest a different picture: rare facts are often encoded but inaccessible, and reverse facts can be recognized even when they cannot be generated. This reframes both phenomena as recall failures rather than *'missing knowledge'*. Our results also complement prior work on *'hidden knowledge'* (Gekhman et al., 2025; Orgad et al., 2025): what appears to be knowledge accessible only through model internals may reflect encoded facts whose recall is context-dependent, such that the right prompt or context can elicit (Jiang et al., 2020).

Given that encoding in frontier LLMs is nearing saturation while substantial headroom remains for recall, future improvements are likely to come from better utilization of existing knowledge. One mechanism that already demonstrates this is thinking: our results show that thinking recovers 40–65% of encoded-but-not-directly-known facts. The gains are most pronounced in challenging settings, such as long-tail facts and reverse questions. This parallels the "tip-of-the-tongue" phenomenon in human memory (Brown & McNeill, 1966; Schwartz, 2002), in which additional effort can surface knowledge that is present but momentarily inaccessible. Post-training and inference-time methods may thus play a central role in factuality by improving how models utilize what they already encode.

In summary, our contributions are: (1) We propose a behavioral framework that characterizes factual knowledge in LLMs by separating encoding from recall via knowledge profiles; (2) We introduce WIKIPROFILE, a benchmark for knowledge profiling, along with a fully automated pipeline for constructing such benchmarks; and (3) Through evaluation of 13 LLMs on Wikipedia-derived facts, we show that the bottleneck for factuality is recall, not encoding, and that thinking helps recover otherwise inaccessible knowledge.

**Conflict of Interest Disclosure:** The authors are employed by Google, which leads the development of the Gemini and Gemma model families, which were evaluated in this paper.

## 2. Knowledge Profiling

We focus on *factual knowledge* as the ability to answer short-form questions correctly without external tools. Our aim is to provide a systematic, model-agnostic characterization

---

[1]By analogy to human memory, we use *encoding* to denote parametric representation, and *recall* to accessing encoded facts.

of factual behavior that applies to both closed- and open-weight LLMs. To this end, we introduce two complementary operational notions, defined purely in terms of observable behavior: *encoding* and *knowledge*. Together, they allow us to determine whether errors arise from encoding or recall, thereby informing different paths to improvement.

## 2.1. Operationalizing Encoding and Knowledge

**Facts and Questions:** We define a *fact* as a proposition[2] involving an ordered pair of entities: a subject and an object. The roles of subject and object are determined by the source text from which the fact was extracted (e.g., a Wikipedia document): the subject is the entity that appears first in the text, and the object appears subsequently. A question whose answer is the object is termed a *direct question*, while a question whose answer is the subject is termed a *reverse question*. Examples are shown in Figure 2.

For each question $q$, we generate $n$ responses from the LLM and use a prompted LLM grader (autorater) to compare each response to the gold answer and label it as correct or incorrect.[3] Let $c_q$ and $i_q$ denote the number of responses to question $q$ graded as correct and incorrect. We define the *question grade* as the accuracy over gradable responses:

$$g(q) = \frac{c_q}{c_q + i_q}.$$

**Encoding:** The notion of *encoding* has been defined in multiple ways in prior work, typically referring to information stored in the model's internal parametric representations. Existing approaches to measuring encoding often rely on access to internal states, a requirement that does not align with our focus on evaluating frontier LLMs. Moreover, the extent to which these methods reliably capture whether a model truly encodes a fact remains an open question (Hase et al., 2023; Ma et al., 2023; Huang et al., 2024; Wei et al., 2024b; Chen et al., 2025; Haller et al., 2025).

We therefore adopt an *encoding-via-memorization* perspective: an LLM **encodes** a fact if it can correctly reproduce that fact in a pre-training-like context. Intuitively, successful reproduction under training-like conditions provides sufficient evidence that the fact is encoded (though it may not be known). Since encoding, under this definition, is assessed behaviorally, we operationalize it using tasks that strongly prime the model with the context in which the fact $f$ originally appeared. We denote the set of such encoding tasks

---

[2]Although facts are often described as subject-relation-object triplets, in our data they may involve multiple or complex relations and are therefore treated as *propositions*.

[3]The grader may assign 'partially' or 'other' labels when a response cannot be reliably evaluated. See §4.

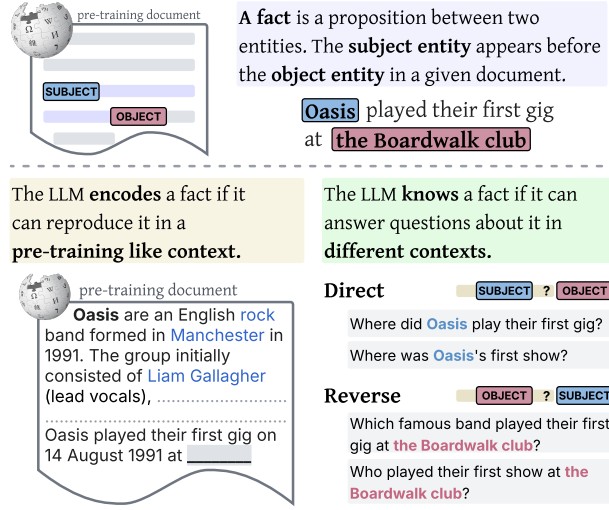

*Figure 2. Top:* We extract facts from Wikipedia, a predominant source of pre-training data. *Left:* We measure encoding by prompting the LLM to reproduce facts within their original context, testing whether they are stored in the model's parameters. *Right:* We measure knowledge by asking questions across varied phrasings and relational directions, with and without thinking.

by $\mathcal{E}_f$. We say an LLM *encodes* the fact $f$ if:

$$\exists q \in \mathcal{E}_f, \quad g(q) > \tau,$$

where $\tau$ is a predefined threshold. We set $\tau = 0.5$ in all experiments as a natural majority threshold. We show in Appendix D.2 that our findings are robust to this choice.

In our setup, $\mathcal{E}_f$ consists of two tasks. The first task is *proposition completion* (see Figure 2), in which the model is given the *left context* (the entire source text truncated immediately before the object entity), and is asked to complete the factual statement. This task directly mimics the pre-training objective for which the LLM was optimized. Notably, the answer (the object entity) does not appear in the left context.

The second task, which we refer to as *contextual questioning*, uses the same left context but replaces the final sentence with a direct question about the fact. We include both tasks because completion-based evaluation can be ambiguous for post-trained LLMs, which are optimized to answer questions rather than continue pre-training documents. LLMs may continue a sentence in multiple plausible ways that do not explicitly state the target fact. The contextual questioning task mitigates this ambiguity by explicitly eliciting the target fact in question form while preserving the original source context, which primes the model.

We deliberately exclude *thinking* from the encoding evaluation. Correct reproduction can arise not only from direct encoding but also from inference, including multi-hop reasoning or educated guessing based on other encoded facts. To avoid conflating encoding with inference, encoding is

always measured without thinking. Knowledge, in contrast, is evaluated both with and without thinking.

**Knowledge:** We say that an LLM **knows** a fact if it can correctly answer questions about it across a range of semantically equivalent contexts. This definition captures the intuition that factual knowledge should be robust to superficial changes, such as in phrasing and to alternative relational directions (Elazar et al., 2021; Meng et al., 2022; Zucchet et al., 2025). Let $\mathcal{Q}_f$ denote a set of questions associated with a fact $f$. We say an LLM *knows* a fact $f$ if:

$$\forall q \in \mathcal{Q}_f, \quad g(q) > \tau,$$

The contrast with encoding is intentional: encoding uses existential quantification ($\exists$) because reproducing a fact in any priming context suffices as evidence of storage, whereas knowledge uses universal quantification ($\forall$) because robust recall should not depend on phrasing or query direction. In our setup, $\mathcal{Q}_f$ consists of four questions: two direct and two reverse, each in a different phrasing. This tests invariance to both surface form and relational direction.

## 2.2. Knowledge Profiles

Given our operational definitions, each fact can be characterized along three dimensions: whether it is *encoded*, whether it is *known without thinking*, and whether it is *known only with thinking*. The interaction of these dimensions yields qualitatively distinct *knowledge profiles*. In particular, we identify five profiles, illustrated in Figure 1.

**Encoding Failure:** The fact is neither encoded nor known, with and without thinking. A high rate of encoding failures indicates limitations in model capacity or pre-training data coverage, suggesting interventions such as increasing model size or expanding the training set.

**Recall Failure:** The fact is encoded but not known, even with thinking. This indicates that the fact is stored but inaccessible at inference time, and calls for improvements beyond pre-training, such as in post-training.

**Direct Recall:** The fact is encoded and known without thinking. This straightforward recall reflects highly accessible knowledge, which is desirable in many applications (e.g., when integrating facts into long-form responses).

**Recall with Thinking:** The fact is encoded but known only with thinking. This indicates that the fact is not readily accessible and requires additional computation to be recalled. This profile can be viewed as a recovery mechanism for otherwise inaccessible facts.

**Inference without Encoding:** The fact is not encoded but is known when thinking is enabled. In this case, correctness

may arise from inference over other encoded facts, such as multi-hop reasoning or educated guessing. It is also possible that the fact is encoded but missed by our encoding tasks. This profile is less reliable; relying on it may promote hallucinations, and it occurs less frequently in practice.

Finally, we do not define a profile for facts that are not encoded but known without thinking. While such cases are possible, they occur in fewer than 0.5% of instances across all evaluated models and are likely attributable to sampling noise or grader error. We therefore exclude this profile.

## 3. The WIKIPROFILE Benchmark

To operationalize the knowledge profiles, we need a benchmark that supports measuring both encoding and knowledge. We hence introduce WIKIPROFILE, a benchmark for evaluating and profiling factual knowledge in LLMs. Since profiling involves measuring encoding, we extract facts from Wikipedia, a trusted reference that pre-training pipelines are likely to prioritize, making it both a strong candidate for encoded knowledge and a reliable ground truth. WIKIPROFILE consists of 2150 facts, each with ten corresponding questions. These include two questions for measuring encoding (proposition-completion and contextual questions); four questions for measuring knowledge (direct and reverse questions, each instantiated with two distinct phrasings); and four multiple-choice variants for measuring fact verification capabilities. Examples of tasks are in Table 1.

All questions in WIKIPROFILE have a single, unambiguous gold answer, enabling automatic validation with an LLM grader (see §4). WIKIPROFILE is constructed through a fully automated pipeline (Gemini-2.5-Pro with thinking; Comanici et al. 2025) that may be applicable to other corpora for domain-specific knowledge profiling.

**Pipeline Overview:** The full pipeline spans dozens of steps and fifteen prompts; here we provide a high-level overview (Figure 3), with complete details and prompts in Appendix A and E. All prompts in the pipeline were carefully developed through manual optimization and error analysis on a held-out subset of tens of documents, from which the examples used in the prompts are drawn.

**Fact Extraction:** To extract facts, we sample 10,000 Wikipedia pages, record their visit counts, and categorize them into nine topics (see Figures 9 and 10 for distributions). Next, we perform NER to identify entities and their types (e.g., DATE, LOCATION, PERSON; see Figure 11). For each document, we select up to 3 candidate object entities (not from the first sentences), each of which defines a distinct fact. These candidates are selected based on strict criteria: an entity is suitable only if it represents a non-trivial, non-guessable, and time-independent (i.e., not subject to

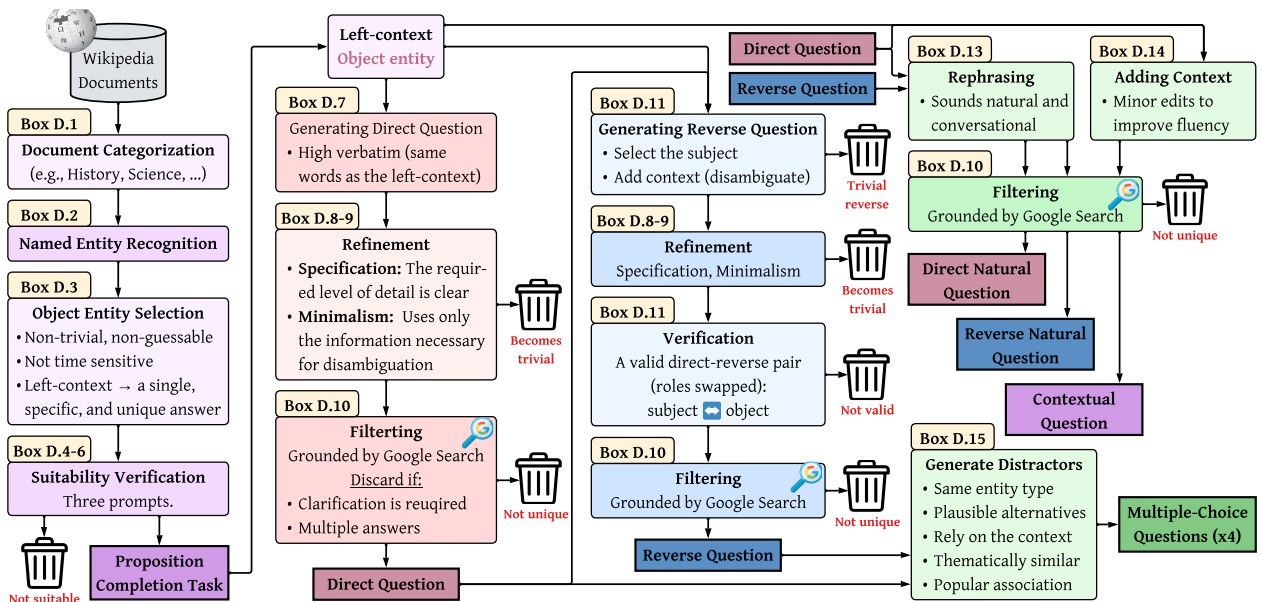

*Figure 3.* **The WikiProfile Creation Pipeline:** We propose a fully automated pipeline based on prompted LLMs. The yellow boxes denote the specific prompts used at each step (see §E). *Left (purple):* Fact extraction and construction of the proposition completion task. *Center (red and blue):* Construction of direct and reverse questions via generation, refinement, and filtering (grounded by Google Search). *Right (green):* Creation of remaining questions (natural phrasing, contextual, and multiple-choice versions) based on the direct/reverse pairs. Facts are discarded throughout the pipeline if their associated questions are rejected. Full details in Appendix A. The encoding tasks are 'Proposition Completion' and 'Contextual Question'. The knowledge tasks are 'Direct/Reverse', 'Direct/Reverse Natural' questions.

change) completion of its left context. Furthermore, the left context must constrain the completion to a single, unambiguous answer. After a final verification step, we downsample to 5,000 candidates, balancing categories and entity types. The median is 100 words, the mean is 116 words ($\pm 56$ std).

**Question Generation:** We generate direct and reverse questions through a three-step process of generation, refinement, and filtering, ensuring that each question is unambiguous, specific, minimal, and has a unique answer. Direct questions are generated first, and reverse questions are then derived by swapping subject and object roles. Because our goal is to test encoding, we initially generate high-verbatim questions that closely match the source text; we then create natural rephrasings of each to test robustness to phrasing variation. Contextual questions are constructed by appending the direct question to its left context, providing maximal priming. For each question, we also generate a multiple-choice variant with three plausible distractors matched by entity type and thematic similarity. All questions undergo filtering grounded in Google Search. We prompt an LLM with each question and discard cases where multiple answers are returned or clarification is needed. This filtering is strict: if any question is rejected, the entire fact is discarded, leaving us with approximately 2,200 facts.

Finally, we manually validated facts and their associated questions, discarding fewer than 2% of facts and yielding a final set of 2,150 facts. See Appendix A.4 for details.

## 4. Experimental Setup

**LLMs:** We evaluate 13 LLMs from five families, covering commercial and open-weight models: Gemini-3 and Gemini-2.5 (Pro and Flash variants) (Comanici et al., 2025); GPT-5.2, GPT-5, and GPT-5-mini (OpenAI, 2025b); GPT-4.1 and GPT-4.1-mini (OpenAI, 2025a); and Gemma3 (1B, 4B, 12B, 27B) (Kamath et al., 2025). Each model is evaluated with and without thinking. Gemini-2.5, Gemini-3, and GPT-5 are thinking-optimized; we use default settings when thinking is enabled and set effort/budget to zero to disable it. For GPT-4.1 and Gemma3, we use CoT prompting.

**Response Generation:** For each example (an incomplete text or a question), we generate eight responses per LLM with a temperature of 1. In Appendix D.4, we justify the choice of eight responses. We employ three prompts, one for each task format (completion, closed-book QA, and multiple-choice QA). Each prompt instructs the LLM to produce concise, answer-only responses (see Box E.17). For multiple-choice questions, answer choices are shuffled each time, ensuring that each option is the correct answer exactly twice. In total, we evaluate $\approx 4.5$ million responses ($13 \times 2$ LLMs $\times 2150$ facts $\times 10$ tasks $\times 8$ responses).

**Response Evaluation:** We use two prompted LLM graders (autoraters) (Wei et al., 2024a): one for the completion tasks (Box E.18) and one for the closed-book QA tasks (Box E.19). Both graders use Gemini-2.5-Pro with thinking.

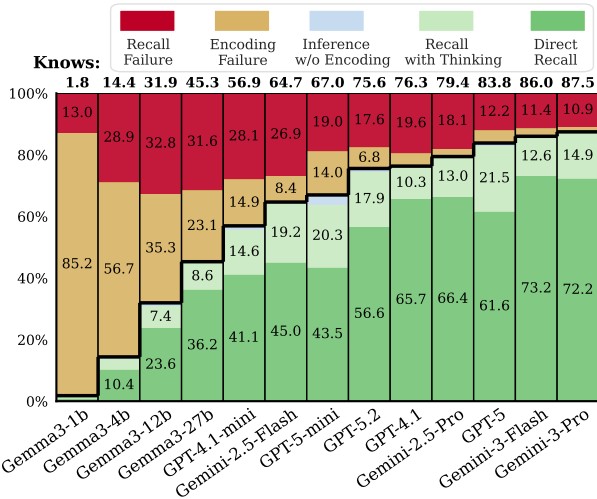

Figure 4. **Knowledge Profiles:** Distribution of the five profiles across 13 LLMs (percentages). The black line marks *potential knowledge*: the fraction of facts known with or without thinking ('Direct Recall'+'Recall with Thinking'+'Inference without Encoding'). As shown, encoding failures decrease sharply with scale, while recall failures persist even in frontier models.

The graders assign one of four labels: CORRECT, INCORRECT (factual contradiction), PARTIALLY (correct but at a different granularity; Yona et al. 2024), or OTHER (unattempted or unverifiable answers). Including PARTIALLY improved grader stability in preliminary experiments.

As defined in §2.1, the question grade is computed using CORRECT and INCORRECT responses only, excluding PARTIALLY and OTHER, since assigning them weights is subjective (Kalai et al., 2025). In practice, such responses are rare (less than 5%; see Table 2). When all responses are PARTIALLY or OTHER, the fact is excluded from aggregate statistics. Alternative handling strategies yield negligible differences (see our analysis in Appendix D.1).

Finally, in Appendix D.3, we assess cross-family grader consistency by comparing graders based on Gemini-2.5-Pro and GPT-5, and find 98.2% agreement. The remaining disagreements primarily involve OTHER or PARTIALLY labels, which are excluded from our analysis; thus, our results are not sensitive to the choice of underlying LLM.

## 5. Results

Our results suggest that factual errors arise not from *missing knowledge* (empty shelves), but from *inaccessible knowledge* (lost keys). We first characterize this recall bottleneck by examining knowledge profiles across all 13 evaluated models (§5.1). We then ask why recall fails, identifying two systematic patterns, fact popularity and question directionality, where encoded knowledge remains difficult to access (§5.2). Finally, we examine thinking as a mechanism that

compensates for these weaknesses, recovering facts that direct generation cannot surface (§5.3). Due to the analysis scale, we occasionally report results for frontier LLMs only; complete results and details are provided in Appendix C.

### 5.1. The Recall Bottleneck

We begin by examining knowledge failures, namely, facts that are not known. These correspond to two profiles: *Encoding Failures* and *Recall Failures*. Figure 4 shows the distribution of all five knowledge profiles across 13 LLMs.

**Encoding is saturated; recall is not.** For frontier LLMs such as Gemini-3-Pro and GPT-5, factual encoding is near saturation, with 95-98% of facts encoded. Yet these models fail to directly recall 26–34% of the facts, or 11–12% even with thinking. Accordingly, recall failures account for more than 70% of GPT-5.2's errors and a larger share in stronger models, suggesting recall is indeed a bottleneck.

**Scaling improves encoding more than recall.** The Gemma3 model family illustrates this clearly. Increasing model capacity and size from 1B to 27B parameters reduces encoding failures from 85% to 23%. However, as model scale increases, a growing fraction of errors shifts from encoding failures to recall failures, peaking at 33% (with thinking) and 40% (without). In this family, scaling increases what the model stores; it does not necessarily improve what it can access at inference time.

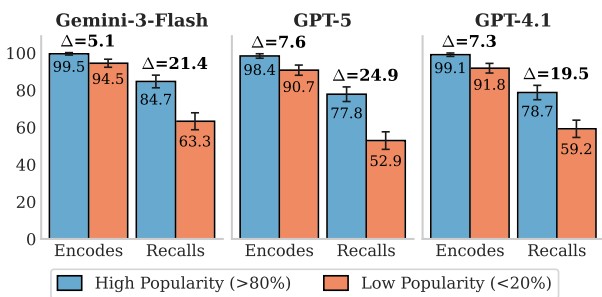

Figure 5. **Fact Popularity:** We compare two popularity tiers (bottom 20% vs. top 20%) in terms of encoding rates and direct recall rates (knowing encoded facts without thinking). The Δ indicates the gap between tiers. As shown, it is narrow for encoding but wide for recall. Figure 16 presents all LLMs.

### 5.2. Why Recall Fails

Having established that recall is the bottleneck, we now ask: what makes it difficult? To answer this, we next focus on encoded facts. We show that recall remains tied to the conditions under which facts were acquired. During pre-training, facts are encountered in specific contexts, phrasings, and orderings. When queries diverge from the training-time patterns, recall becomes harder. We examine two manifestations of this: fact popularity and question directionality.

**Rare facts are encoded but hard to recall.** We compare facts from low-popularity Wikipedia pages (bottom 20% by page views, used as a proxy for rare facts) with those from high-popularity pages (top 20%). Figure 5 presents, for the two tiers, the percentage of encoded facts and, among encoded facts, the percentage that are directly recalled. Conditioning on encoded facts isolates recall from encoding, allowing us to assess whether rare facts are harder to recall even when they are encoded. First, we find that rare facts are encoded at high rates approaching those of popular facts. Second, while the *encoding gap* between popular and rare facts is modest (e.g., for Gemini-3-Pro, 99.5% vs. 94.5%, $\Delta = 5$), the *recall gap* is far larger (84.6% vs. 63.3%, $\Delta = 21.3$); this pattern holds for all frontier LLMs. For smaller models like Gemma3-4B, encoding gaps are much larger; this is consistent with our broader finding that scaling primarily improves encoding.

**Reframing the long-tail problem.** Prior work established that LLMs struggle with long-tail (rare) facts, typically interpreted as a capacity limitation (Kandpal et al., 2023; Mallen et al., 2023; Sun et al., 2024). Our results suggest a complementary picture: rare facts *are* encoded; however, recalling them is the challenge. Currently, the bottleneck has shifted from knowledge acquisition to utilization. Consequently, improving factuality in frontier LLMs will likely require interventions that enhance recall, not only scale.

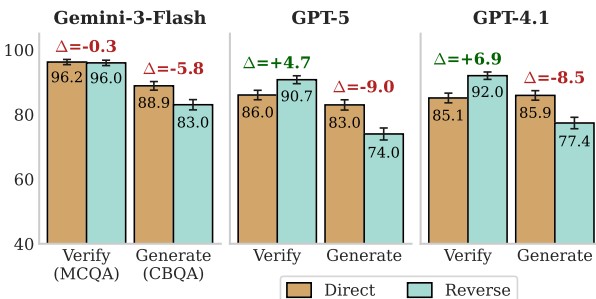

*Figure 6.* **The Reversal Curse:** We compare direct recall rates (knowing encoded facts without thinking) on direct and reverse questions across two tasks: verification (multiple-choice) and generation (closed-book). The $\Delta$ denotes the gap between the direct and reverse settings. We find that LLMs handle reverse questions effectively in verification but struggle in generation. Figure 16 presents results for all LLMs.

**Reverse questions are verifiable but hard to recall.** Figure 6 shows that all LLMs (except for Gemini-2.5-Flash) exhibit lower recall rates for reverse questions (e.g., GPT-5: 82.9% direct vs. 74% reverse). This asymmetry is consistent with the "reversal curse" documented by Berglund et al. (2024). Our results suggest a refinement of this view. If LLMs truly lack bidirectional knowledge, they should struggle with reverse questions regardless of format. To test this, we compare closed-book generation with multiple-choice

questions, in which the correct answer is presented among distractors (verification). In verification, reverse questions are no harder than direct ones: for 9 models, reverse is actually *easier*; for the remaining 4, performance is similar. This dissociation suggests that LLMs *are aware* of the bidirectional association of the fact; they can recognize the correct answer when presented with it. Their failure lies in recalling it when the direction does not match the training data. In this view, the reversal curse is a recall phenomenon rather than a lack of bidirectional association.

**Phrasing is not a factor.** Our definition of *knows* requires correctness across phrasings and directions. We verify that phrasing does not confound our results by conducting hypothesis tests comparing the *distributions* of performance under high-verbatim and natural phrasings. Across 104 hypothesis tests (13 LLMs, with and without thinking, across four task pairs) and after FDR correction, we find no significant effects (see Appendix C.3 for details).

## 5.3. Thinking as a Recovery Mechanism

We now turn to the question of what enables the recovery of otherwise inaccessible knowledge. To this end, we examine the potential of *thinking* to fill this role. Thinking-optimized LLMs such as Gemini-3, Gemini-2.5, and GPT-5 models allocate additional computation to thinking by default; for non-thinking models (Gemma3 and GPT-4.1), we elicit similar behavior through CoT prompting.

**Thinking targets recall weaknesses.** Figure 7 presents the improvement from thinking **on encoded facts** (i.e., recall gains), broken down by popularity and directionality. Thinking yields larger gains precisely where direct recall is weakest. For Gemini-3-Pro, thinking improves the recall of rare facts by 20.1 points, versus 11.3 for popular facts; for reverse questions, GPT-5 gains 19 points, versus 12 for direct questions. Consequently, thinking narrows both the popularity gap (from $\Delta = 21.4$ to 12.5) and the directionality gap (from $\Delta = 9$ to 2).

**Why does thinking help?** Thinking gains are remarkable because our facts are single-hop; they require no complex multi-step reasoning. Why should thinking help recall a simple fact such as *"Where did Oasis play their first show?"* We consider three candidate mechanisms (with additional details in Appendix B.1) through which thinking improves performance: (i) *Response diversity*: thinking increases output variance, improving the chance that at least one sample is correct (a sampling effect); (ii) *Inference*: thinking enables multi-step derivation from other encoded facts; (iii) *Recall facilitation*: thinking helps the model locate and retrieve a specific encoded fact that direct generation fails to access.

Our analysis in Appendix B.1 rules out response diversity as the main mechanism that improves factuality for

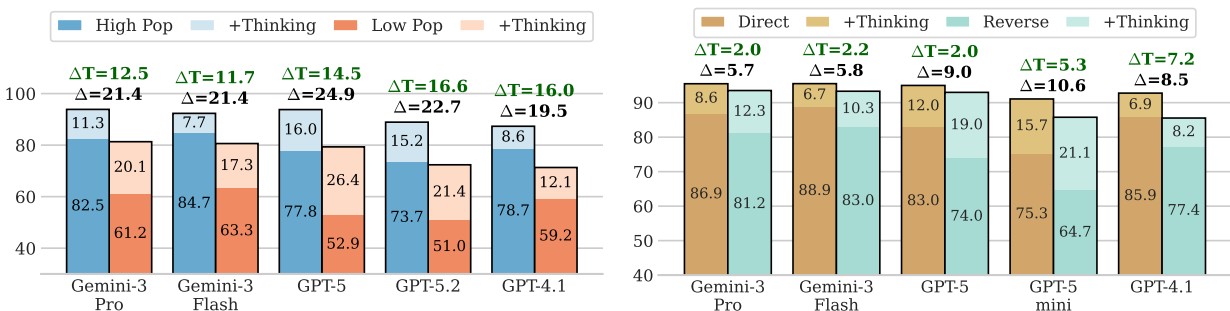

*Figure 7.* **Thinking and Recall:** We examine the impact of thinking on recall (knowing encoded facts). On the *left*, we compare two popularity tiers (bottom 20% vs. top 20%); On the *right*, we compare direct and reverse questions. The popularity or directional gaps are denoted by $\Delta$ (no thinking) and $\Delta T$ (with thinking). As shown, thinking narrows the gaps ($\Delta T < \Delta$). Figure 17 presents all LLMs.

thinking-optimized LLMs. When thinking is enabled, models become more consistent in their correctness, producing a higher fraction of correct samples for the same question. This pattern is incompatible with increased output variance. We therefore believe that thinking primarily facilitates recall of encoded knowledge. Figure 8 provides evidence that supports this: thinking recovers 40–65% of encoded but not directly known facts in thinking-optimized LLMs, but only 5–20% of non-encoded facts. This dependence on encoding status is expected under recall facilitation. It is also partially expected under inference, since non-encoded facts may lack the related encoded premises that inference requires. Nonetheless, our facts are single-hop, and complex deductive reasoning is unnecessary to derive the correct answer. For single-hop facts, recall facilitation is the more parsimonious explanation: the model already stores the answer and need not derive it from other knowledge. We therefore conclude that thinking primarily helps models access knowledge that they have already encoded. Gekhman et al. (2026) concurrently show that thinking expands parametric recall through computational buffering and priming.

**The cost of thinking.** While thinking improves recall, it is computationally expensive. Across frontier models, 10–20% of facts are accessible *only* via thinking. Determining exactly when to invoke thinking is fundamentally a metacognitive problem, as it requires the model to recognize when direct recall is insufficient before triggering more intensive computation (Yona et al., 2026). Improving direct recall could help side-step this issue until models develop improved metacognitive abilities.

### 5.4. Connections to Human Cognition

The role of thinking in supporting recall resonates with well-studied phenomena in human memory. The *tip-of-the-tongue* phenomenon describes states in which a person is confident they know something but cannot immediately produce it (Brown & McNeill, 1966; Schwartz, 2002). Such states are often resolved through deliberate effort: think-

ing of related concepts or mentally retracing the context in which the information was encountered. The parallel to our findings is suggestive: encoded-but-not-known facts in LLMs may represent a functional analogue, in which the information is stored but the query is insufficient to trigger recall. Thinking provides the additional computation that, like deliberate effort in humans, bridges the gap.

Separately, our verification results (in multiple-choice settings, §5.2) echo the *feeling-of-knowing* phenomenon: people often predict they will recognize an answer even when they cannot recall it (Hart, 1965; Nelson & Narens, 1990). The dissociation, in which LLMs can verify (recognize) answers to reverse questions but cannot generate (recall) them, mirrors this asymmetry in human memory. While we do not claim that LLMs implement human-like cognitive processes, the functional parallels suggest that the gap between encoding and recall may be a general property of systems in which the conditions for storing information diverge from the conditions for recalling it.

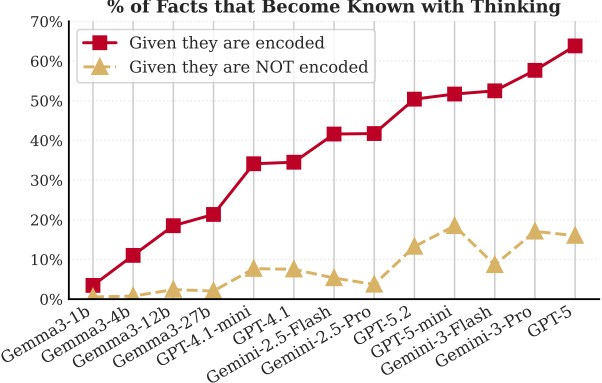

*Figure 8.* **Recovering Depends on Encoding:** We report the percentage of not-known facts that become known with thinking, conditioned on whether the fact is encoded (red) or not (yellow). Thinking recovers 40–65% of encoded facts in thinking-optimized LLMs, but only 5–15% of non-encoded facts. This dependence on encoding is consistent with the recall facilitation mechanism.

# 6. Related Work

**Encoding and Memorization:** Our behavioral definition of encoding connects to work on memorization in LLMs (Carlini et al., 2023; Huang et al., 2024; Lu et al., 2024; Menta et al., 2025; Wang et al., 2025a). In particular, our encoding criterion aligns with the probabilistic framework of Hayes et al. (2025). Memorization work typically focuses on privacy or data leakage concerns, whereas we use encoding-via-memorization for knowledge profiling.

**Latent Knowledge:** A separate line of work uses internal representations to argue that LLMs possess "latent knowledge" or "hidden knowledge": facts they encode but fail to express in generation (Kadavath et al., 2022; Burns et al., 2023; Gottesman & Geva, 2024; Gekhman et al., 2025; Orgad et al., 2025; Simhi et al., 2025). One possible reason is that the post-training phase may introduce new knowledge that undermines the use of the knowledge acquired during pre-training (Gekhman et al., 2024; Kaplan et al., 2026). Our results offer a complementary behavioral perspective: what appears to be latent knowledge may reflect context-dependent recall. Facts surface when queries match training conditions (as in our encoding tasks) but become harder to access when they diverge (as in our knowledge tasks).

**The Reversal Curse:** Berglund et al. (2024) documented that LLMs trained on "A is B" often fail to answer "What is B?". Subsequent work has attributed this to the autoregressive objective (Guo et al., 2024; Kitouni et al., 2024; Lv et al., 2024), training dynamics (Zhu et al., 2024), or data asymmetries (Golovneva et al., 2024; He et al., 2025), with proposed solutions focusing on pre-training modifications (Golovneva et al., 2024; Yu et al., 2025) or architectural changes (Wang & Sun, 2025). By comparing verification to generation, we show that LLMs can recognize reverse answers they cannot produce, suggesting the curse reflects a *recall asymmetry* rather than a failure to learn bidirectional associations. Thinking can mitigate the curse, providing a path to improvement that prior work has not explored.

**Thinking and Factuality:** The effect of CoT prompting (Wei et al., 2022; Wang et al., 2024b) and reasoning-optimized models (Marjanovic et al., 2025) on factuality remains contested. Some studies report improved accuracy via increased diversity (Wang et al., 2023; Yang et al., 2025), while others find higher hallucination rates (Wang et al., 2024a; Yao et al., 2025) or attribute gains to multi-hop reasoning (Zhang et al., 2025). Effects on consistency are similarly mixed, with some studies reporting improvements (Kim et al., 2025; Wu et al., 2025a) and others reporting degradation (Cheng et al., 2025a;b). A contemporaneous work by (Gekhman et al., 2026) also demonstrated that thinking can expand the boundary of parametric knowledge recall, identifying two key mechanisms underlying this ef-

fect: a computational buffer and factual priming. Our work offers a unifying perspective: thinking functions as a recall mechanism, not just a reasoning mechanism. Its effect is most pronounced under challenging conditions, such as rare facts and reverse questions.

# 7. Discussion

In this work, we propose a framework for knowledge profiling and apply it to profile 13 LLMs across a benchmark of naturally occurring facts from Wikipedia. Our results point to recall, not encoding, as the primary bottleneck for factuality in frontier LLMs. Across the models we evaluated, encoding is approaching saturation; the strongest LLMs encode over 95% of facts in our benchmark. The challenge lies in accessing what has been learned. Recall failures are not random; they concentrate on rare facts and reverse questions, conditions where queries diverge from training-time contexts. This pattern suggests that LLMs store facts in ways that remain tied to their acquisition, making recall sensitive to surface-level query characteristics.

This shift from encoding to recall has implications beyond short-form factuality. Distinguishing between encoding and recall failures provides a framework that could extend to long-form generation, multi-step reasoning, and other domains where accessing learned knowledge is critical. In the age of retrieval-augmented generation (RAG) and tool-using agents, it is tempting to view parametric knowledge as secondary, something RAG can compensate for. But parametric knowledge is essential for fluency, speed, and integration across contexts. Improving how models access what they already encode is, therefore, a meaningful target.

The gap between encoding and recall might be bridged through intervention during *pre-training*, for example, by augmenting training data with self-generated question-answer pairs (Lin et al., 2025), or during *post-training*, as recent studies suggest that alignment teaches models how to better utilize knowledge acquired during pre-training (Zhou et al., 2023; Gekhman et al., 2024; Lin et al., 2024a). Our findings also suggest that *inference-time* techniques, such as thinking, can be effective in aiding parametric recall.

Our study has several limitations worth noting. WikiProfile is scoped to Wikipedia, an encyclopedic domain of relatively prominent facts, and our findings may not generalize to other corpora. Our operational definition of knowledge is likewise bounded: we measure recall along two axes, directionality and phrasing, for single-hop facts, and do not claim to exhaustively characterize all forms in which a fact can be queried. Finally, fully profiling a frontier model on WikiProfile costs $\pm\$500$, though this can be reduced substantially by omitting multiple-choice questions or using fewer samples per question.

## Impact Statement

This paper presents work aimed at understanding factual errors in LLMs. Our findings could inform deployment decisions by clarifying when models are likely to fail (e.g., with rare facts or reverse questions) and when inference-time computation may be beneficial. We do not foresee negative societal consequences specific to this work.

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

# A. WikiProfile – Full Details

In this section, we describe WikiProfile and its construction in detail. WikiProfile comprises ten tasks defined for each fact extracted from a Wikipedia source text; their properties and examples are summarized in Table 1. The tasks span three formats: text completion, closed-book question answering, and multiple-choice question answering. Two tasks measure encoding (proposition completion and a contextual question), and four tasks measure knowledge (direct and reverse questions, each with two phrasings). Each knowledge task additionally has a multiple-choice variant.

WikiProfile is constructed using a fully automated pipeline powered by a prompted LLM, Gemini-2.5-Pro with thinking enabled (Comanici et al., 2025). Our goal is to provide a framework that can be readily applied to other corpora, including domain-specific ones, enabling researchers to identify knowledge profiles and characterize LLM behavior. Prompts were developed through manual optimization on a small subset of approximately 100 documents and facts, which is excluded from the final dataset. This process involved iterative error analysis and refinement.

## A.1. Tasks and Properties

We begin by outlining the properties that guided the design of WikiProfile.

**Required Properties**   All questions must satisfy the following properties to ensure methodological rigor and enable comparison against a single verifiable gold answer:

- **Answerable:** A correct answer exists.
- **Unambiguous:** The question is self-contained and precisely phrased.
- **Specific:** The expected answer type and level of detail are clearly specified.
- **Single-answer:** Exactly one answer is correct.
- **Not time-sensitive:** The answer is stable over time.

**Controlled Properties**   These properties define systematic variations across tasks. By manipulating them, we examine factors that affect factual knowledge.

- **Format:** Completion, closed-book, or multiple-choice.
- **Context:** The amount of auxiliary information provided, ranging from full source context (used to measure encoding) to minimal context required for disambiguation.
- **Verbatim:** The degree of linguistic overlap with the source text, spanning exact phrasing (for completion tasks), high-verbatim questions that reuse source phrases and terms, and low-verbatim, natural user-like queries.
- **Direction:** The relational direction of the question, defined by the ordering of the entities in the source text.

*Table 1.* The properties of the ten tasks in WikiProfile, including examples. All tasks were generated from the Wikipedia page for 'Love and Money'. The extracted fact is: *'The band Love and Money was formed by three former members of Friends Again, along with the bassist Bobby Paterson'*.

| Task | Format | Context | Verbatim | Direction |
|---|---|---|---|---|
| **Completion** | Completion | Full | Exact | Direct |

"Love and Money are a Scottish rock/soul/funk band, formed in 1985 in Glasgow, Scotland. The band was formed by three former members of Friends Again (singer-songwriter and guitarist James Grant, drummer Stuart Kerr and keyboardist Paul McGeechan) along with bassist ..."

| Task | Format | Context | Verbatim | Direction |
|---|---|---|---|---|
| **Contextual** | Closed-book | Full | High | Direct |

"Love and Money are a Scottish rock/soul/funk band, formed in 1985 in Glasgow, Scotland. The band was formed by three former members of Friends Again (singer-songwriter and guitarist James Grant, drummer Stuart Kerr and keyboardist Paul McGeechan). What is the name of the bassist who, along with them, formed the band Love and Money?"

| Task | Format | Context | Verbatim | Direction |
|---|---|---|---|---|
| **Direct** | Closed-book | Minimal | High | Direct |

"What is the name of the bassist who, along with three former members of Friends Again, formed the band Love and Money?"

| Task | Format | Context | Verbatim | Direction |
|---|---|---|---|---|
| **MC Direct** | Multiple-choice | Minimal | High | Direct |

A. Bobby Paterson B. James Grant C. Neil Cunningham D. Stuart Kerr

| Task | Format | Context | Verbatim | Direction |
|---|---|---|---|---|
| **Direct Natural** | Closed-book | Minimal | Natural | Direct |

"Who was the bassist that formed the band Love and Money with the three guys from Friends Again?"

| Task | Format | Context | Verbatim | Direction |
|---|---|---|---|---|
| **MC Dir Nat** | Multiple-choice | Minimal | Natural | Direct |

A. Bobby Paterson B. James Grant C. Neil Cunningham D. Stuart Kerr

| Task | Format | Context | Verbatim | Direction |
|---|---|---|---|---|
| **Reverse** | Closed-book | Minimal | High | Reverse |

"What is the name of the band that was formed by bassist Bobby Paterson and three former members of Friends Again?"

| Task | Format | Context | Verbatim | Direction |
|---|---|---|---|---|
| **MC Reverse** | Multiple-choice | Minimal | High | Reverse |

A. Love and Money B. Friends Again C. Set the Tone D. Deacon Blue

| Task | Format | Context | Verbatim | Direction |
|---|---|---|---|---|
| **Rev Natural** | Closed-book | Minimal | Natural | Reverse |

"Which band was formed by Bobby Paterson with three guys from Friends Again?"

| Task | Format | Context | Verbatim | Direction |
|---|---|---|---|---|
| **MC Rev Nat** | Multiple-choice | Minimal | Natural | Reverse |

A. Love and Money B. Friends Again C. Set the Tone D. Deacon Blue

**Tasks**   Each fact is represented as a relation between a subject and an object entity, where the subject precedes the object in the source text. The *left context* consists of all text preceding the object entity.

We use two tasks to measure encoding:

- **Completion:** Proposition completion of the left context.
- **Contextual:** A high-verbatim question posed over the same left context.

Completion can be ambiguous for post-trained LLMs, which may generate plausible continuations without stating the target fact. The contextual question explicitly elicits the fact while preserving the original context, mitigating this issue.

We use four tasks to measure knowledge:

- **Direct:** A closed-book question, whose answer is the object entity.
- **Direct Natural:** A conversational rephrasing of the Direct question.
- **Reverse:** An inverted question where the object becomes the subject.
- **Reverse Natural:** A conversational rephrasing of the

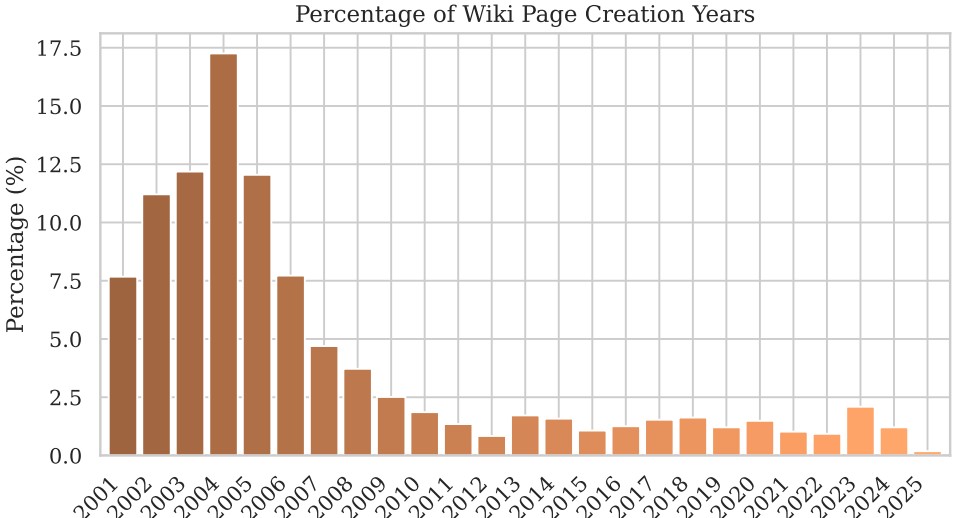

*Figure 9.* Distribution of Wikipedia pages according to their year of creation.

Reverse question.

Finally, we reformulate each of the four closed-book questions as multiple-choice questions with four answer options. The distractors are designed to be plausible and realistic. These tasks evaluate sensitivity to task format and, crucially, the ability to verify facts. This shift from generation to verification probes a distinct form of factual knowledge.

### A.2. Fact Extraction Process

We aim to extract facts that are likely encoded by LLMs (to enable measurement of encoding even for small LLMs) yet remain non-trivial. Notably, we intentionally avoid extracting facts in a schema-based triplet format (subject, relation, object). Our goal is to generate non-trivial questions that are not limited to simple structural relations and may involve more complex ones (e.g., *"What is the condition that caused Erika Nordby to spend two hours without a heartbeat before she was revived?"*).

We therefore extract facts from Wikipedia pages, which were likely a key component of LLM pre-training data. We sample 10,000 Wikipedia page summaries (first paragraphs), along with page popularity statistics.[4] We retain only documents with at least 50 words. The distribution of page creation dates is shown in Figure 9. Pages are then categorized into nine Wikipedia Vital Article categories.[5] The distribution of the categories is presented in Figure 10.

Fact extraction proceeds in three stages. First, the LLM

---

[4]Popularity is based on page views from 2023-02 to 2025-06, according to https://analytics.wikimedia.org/published/datasets/country_project_page/00_README.html

[5]https://en.wikipedia.org/wiki/Wikipedia:Vital_articles/Level/3

performs NER, identifying up to eight entities per document and assigning one of 19 predefined types (Box E.2). The purpose of this stage is to generate a diverse set of entities that can serve as answers. The distribution of the entity types is presented in Figure 11.

Second, the LLM selects up to three entities suitable for proposition completion. Suitability is defined by whether the left context constrains the answer to a *single, specific, and unique correct entity*, such that alternative answers would be factually inconsistent. In addition, the entity is neither time-sensitive nor trivial, making the proposition-completion task challenging. For additional details, including exact criteria and examples, see the prompt in Box E.3. This selection step yields 33,709 candidate object entities. Third, candidates are verified using three prompts, and only those deemed suitable are retained (Boxes E.4, E.5, E.6). This verification process results in 12,031 entities (35.6%). Finally, we downsample the dataset to 5,000 entities by balancing fact categories and entity types.

### A.3. Question Generation Process

We use three stages of generation, refinement, and filtering to ensure that questions satisfy the required properties. The refinement stage resolves ambiguity and enforces specificity, for example, by clarifying the required answer type and adding minimal disambiguating context when needed. The filtering stage is grounded in Google Search to recall plausible alternative answers and to discard questions that admit multiple answers.

We first apply these three stages to construct the *Direct* question. Based on the Direct question, we generate the *Reverse* question, which is then refined and filtered. From the Direct question, we also generate the *Direct Natural* and

*Contextual* questions, which are subsequently filtered. Similarly, we generate and filter the *Reverse Natural* questions. Finally, we generate the multiple-choice variants. Below, we describe the three-stage process used to construct the *Direct* question, and then elaborate on the remaining types.

**Generation**   The *Direct* questions are generated by instructing the LLM to design a question based on the left context such that the only correct answer is the object entity. The question uses only the information necessary for disambiguation (*minimal context*) while preserving exact phrasing from the source text (*high verbatim*). We also specify a set of requirements in the prompt (Box E.7), including ensuring that the question is self-contained and does not include pronouns. We then refine the Direct questions using two additional prompts: one to improve specificity and one to enforce minimalism.

**Refinement**   The specification prompt (Box E.8) instructs the LLM to evaluate a question's specificity: whether it clearly defines the required level of detail in the answer, and to revise it if necessary. The LLM should recommend rejecting a question if it cannot be fixed, either because revising it would leak a non-trivial clue that makes the question too easy or because the question is too vague and would require a full rewrite. This stage rejected 45 questions, leaving Direct questions for 4955 facts. The minimalism prompt (Box E.9) instructs the LLM to revise a question if it contains non-essential information that can be removed while preserving disambiguation and specificity.

**Filtering**   Some questions may still be vague or ambiguous, or admit multiple answers, according to sources outside the Wikipedia context. To filter these cases, we use Gemini-2.5-Pro with thinking grounded in Google Search. Each question is answered with search grounding, and a prompted LLM (Box E.10) evaluates whether the question is suitable. If the grounded answer requests clarification, provides multiple possible answers, or varies by interpretation, the question is rejected. This process removes 10.6% of Direct questions. Filtering applies to all five question types: once a fact is discarded due to any of its questions, it is not reconsidered. The number of removed questions is reported in the relevant paragraphs below.

**Reverse Questions**   For reverse questions, the object entity becomes the new subject, and the subject entity of the Direct question becomes the answer. Given the left context, the Direct question, and the target answer, we instruct the LLM to identify the subject entity and generate a reverse question that satisfies all required properties (Box E.11). In some cases, additional information from the left context is required to ensure that the reverse question is unambiguous and has a single correct answer. For example, consider the

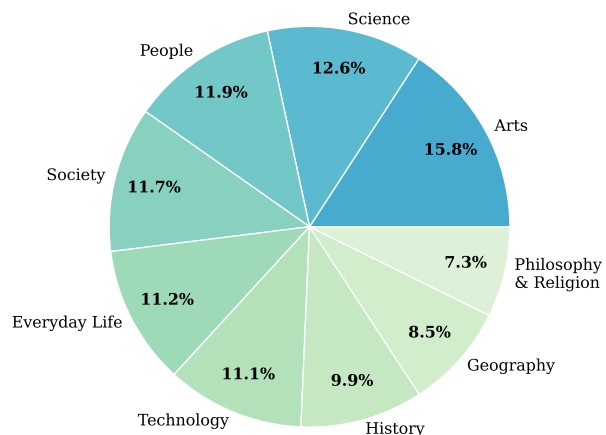

*Figure 10.* Distribution of Wikipedia page categories.

direct question: *"At which event did the film 'There's Still Tomorrow' premiere?"* with the answer *'The 18th Rome Film Festival'*. The subject entity is the film *'There's Still Tomorrow'*. A naive reverse question such as *"Which film premiered at the 18th Rome Film Festival?"* is unsuitable, since multiple films could have premiered there. The LLM should therefore incorporate additional context, such as the director's name, yielding: *"Which film directed by Paola Cortellesi premiered at the 18th Rome Film Festival?"*

The LLM may also reject generating a reverse question if it cannot do so without making it trivial (see the fourth example in Box E.11). This occurs in 2.2% of cases. After generation, we apply the refinement stage, which improves specification and enforces minimalism, rejecting an additional 3.6% of Reverse questions. We then proceed to filtering. Before applying Google Search–grounded filtering, we verify that the *Direct* and *Reverse* questions form a valid pair and are non-trivial, using the prompt in Box E.12. This step removes 31.5% of facts. Finally, the filtering stage removes an additional 13.1% of facts.

**Natural and Contextual Questions**   We instruct the LLM to rewrite each *Direct* and *Reverse* question as *Direct Natural* and *Reverse Natural*, following Box E.13. The *Contextual* question is constructed using Box E.14.[6] It combines the left context (excluding the final incomplete sentence) with the high-verbatim direct question, with minor edits for fluency and to avoid redundancy.

We then repeat the Google Search–based filtering for the *Direct Natural*, *Reverse Natural*, and *Contextual* questions. This removes an additional 5.5% of facts, leaving 2357 facts for which all five question types are available. Finally, we remove facts with lengthy questions (over 40 words) or short

---

[6]We do not generate contextual versions for reverse questions, as these tend to be too easy and the context often includes the answer, frequently the page title.

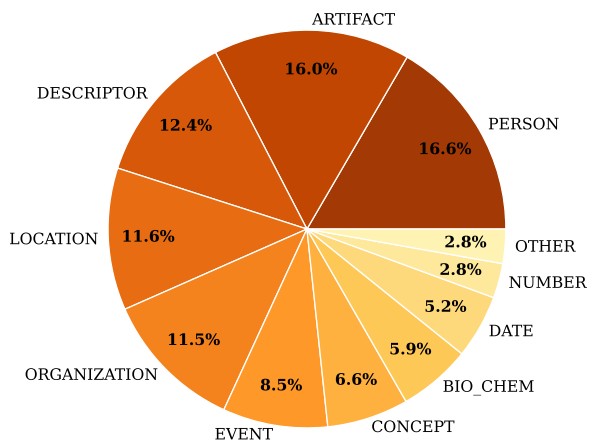

*Figure 11.* Distribution of object entity types. For this figure, we grouped together some types: ARTIFACT (WORK OF ART, PRODUCT), BIO_CHEM (ORGANISM, CONDITION, SUBSTANCE), CONCEPT (IDEA, LANGUAGE, LAW), DESCRIPTOR (CLASSIFICATION, GROUP, IDENTIFIER, TITLE).

left contexts (below 30 words), yielding 2200 facts.

### A.4. Manual Quality Validation

To ensure high data quality, we manually validate questions by adapting the protocol of Nahum et al. (2024), which prioritizes instances in which an LLM ensemble fails. Specifically, we manually examine a fact and its ten associated questions if the four frontier LLMs (Gemini-3-Pro/Flash, GPT-5, and GPT-4.1; evaluated with and without thinking) unanimously fail to correctly answer at least one of the questions, i.e., $g(q) \leq 0.5$. This process flagged 237 facts (10.7%). Upon manual review, we identified 43 ($<2\%$) facts and their associated questions as low quality, and they were subsequently removed from the benchmark. To further validate our data, we examined a subset of 50 facts in which three of the four models provided incorrect answers. We found only 3 errors in this subset. Together, these results suggest that low-quality questions are uncommon in the remaining data and that the benchmark meets a high standard.

## B. Thinking and Recall

In §5.3, we argued that thinking primarily facilitates recall of encoded knowledge. Here we present the full analysis, including formal definitions, detailed results across all models, and a discussion of limitations.

### B.1. Three Possible Mechanisms

We consider three mechanisms by which thinking may improve factual performance on our benchmark. Each mechanism makes distinct empirical predictions, which we evaluate in the subsequent sections.

**(i) Response diversity.** Thinking may alter the output distribution by increasing its variance. With higher variance, the probability that at least one of $n$ samples is correct increases – a sampling effect analogous to the relationship between response diversity and pass@$k$ (Wang et al., 2023; Yang et al., 2025). Importantly, this mechanism does not reflect improved access to knowledge; rather, it is a statistical consequence of broader exploration of the output space.

*Example:* Without thinking, a model asked "Where did Oasis play their first show?" might produce the same incorrect answer in 8 out of 8 samples. Thinking trajectories, by incorporating additional sampling steps, may lead to additional answers, some correct, most not, yielding, say, 2 out of 8 correct samples. Availability of the correct answer has improved (from 0 to 2 correct), but robustness has not: the model does not reliably converge to the correct answer.

*Prediction:* Response diversity increases *availability* (the chance of at least one correct answer across samples) but cannot increase *robustness* (consistent correctness), since higher variance cannot produce reliable convergence to the same correct output.

**(ii) Inference.** Thinking may enable multi-step derivation from other encoded facts. By chaining together related knowledge, the LLM can produce an answer that it does not store directly. This mechanism relies on the availability of related encoded premises, not on the encoding status of the target fact itself.

*Example:* A model may not encode that Oasis played their first gig at the Boardwalk. However, through thinking, it might chain together other related encoded facts: "Oasis formed in Manchester" → "the Boardwalk was a famous music club in Manchester during the 1990s" → "the Boardwalk hosted early gigs by emerging bands." This inference produces the correct answer without direct recall.

*Prediction:* Inference gains are regardless of whether the specific target fact is encoded, since it operates over *other related* encoded facts. We therefore expect inference to produce correct answers even for non-encoded facts.

**(iii) Recall facilitation.** Thinking may help the model locate and recall a fact that is encoded but not accessible via direct generation. The fact is stored in the model's parameters, as verified by the encoding tasks, but the bare question fails to activate its recall. Thinking provides additional computation that bridges this gap.

*Example:* An LLM can complete the relevant Wikipedia proposition about Oasis's first gig when given the left context. However, when asked "Where did Oasis play their first show?" without context, direct generation fails. With

thinking, the model can generate intermediate context (e.g., recalling details about the band's early history in Manchester) and perform additional computations that prime recall of the stored answer.

*Prediction:* Recall facilitation, by definition, requires the target fact to be encoded. It should therefore benefit only encoded facts. Unlike response diversity, it can improve both availability and robustness, since successful recall of a stored fact yields consistent correctness.

### B.2. Measuring Availability and Robustness

To operationalize the distinction between availability and robustness, we propose correct $\geq p$, a metric that captures both aspects along a continuous spectrum. It is defined as the proportion of questions whose estimated probability of correctness across samples is at least $p$

For each question, we draw $n$ independent samples and observe $k$ correct responses, yielding an empirical correctness rate $\hat{p} = k/n$. We then estimate the probability that, in a repeated run of $n$ samples, at least a $p$ fraction of the answers would be correct. Let $m = \lceil pn \rceil$. Using a binomial model $K \sim \text{Binomial}(n, \hat{p})$, then:

$$\hat{P}(K \geq m) \;=\; \sum_{j=m}^{n} \binom{n}{j} \hat{p}^{\,j} \, (1 - \hat{p})^{\,n-j}.$$

Aggregating over questions, correct $\geq p$ measures the fraction of questions for which this estimated probability exceeds 0.5, i.e., $\hat{P}(K \geq \lceil pn \rceil) > 0.5$.

The metric makes the trade-off between availability and robustness clear. At low values of $p$ (e.g., $p = 0.125$, corresponding to at least 1 correct out of 8), the metric asks whether the model can produce any correct answer, which is analogous to pass@k. At high values of $p$ (e.g., $p = 0.875$), it asks whether the model is correct almost all of the time.[7]

### B.3. Ruling Out Response Diversity

Figure 12 presents correct $\geq p$ for all 12 LLMs, comparing performance with and without thinking across the full range of thresholds. At low thresholds ($p < 0.5$, corresponding to **Availability**), thinking improves correct $\geq p$ for all LLMs. This is consistent with all three mechanisms, including response diversity. At high thresholds ($p > 0.5$, corresponding to **Robustness**), the picture diverges across model families. For thinking-optimized LLMs (Gemini-3-Pro, Gemini-3-Flash, Gemini-2.5-Pro, GPT-5, GPT-5.2, GPT-5-mini), thinking also improves robustness: the +Thinking curve re-

---

[7] With $n = 8$ samples, correct $\geq p$ is insensitive to changes at very low correctness probabilities (roughly $p \lesssim 0.1$), since such cases frequently yield zero correct samples. Thus, availability gains in the extreme low-probability tail may be underestimated, while robustness at high $p$ remains well-estimated.

mains above the Base curve across the full range of $p$. This pattern is incompatible with response diversity as the sole explanation, since increased output variance cannot produce greater consistency in correctness.

For Gemma3 models (4B, 12B, 27B), the pattern reverses: thinking (elicited via CoT prompting) *degrades* robustness, with the curves crossing at intermediate $p$. GPT-4.1 occupies a middle ground: CoT prompting yields modest gains in robustness. Our results suggest that, unlike thinking-optimized LLMs, CoT prompting mainly increases response diversity, and its effectiveness in enhancing inference or recall may depend on model capacity or training.

### B.4. Evidence for Recall Facilitation

Having ruled out response diversity for thinking-optimized LLMs, we now ask whether the remaining gains reflect inference or recall facilitation. Figure 8 reports, for each model, the fraction of facts that become known only with thinking, conditioned on encoding status.

The results show a strong dependence on encoding: for thinking-optimized LLMs, thinking recovers 40–65% of of encoded but not directly known facts, compared to less than 20% of non-encoded facts. This dependence is expected under recall facilitation; however, it does not, by itself, rule out inference. Nonetheless, the cumulative evidence favors recall facilitation as the primary mechanism by which thinking improves the factuality of thinking-optimized LLMs:

- Thinking improves robustness, ruling out diversity (§B.3).
- Thinking gains concentrate mostly on encoded facts, consistent with recall facilitation (Figure 8).
- The facts in our benchmark are single-hop: complex deductive reasoning is unnecessary to derive the correct answer, making inference a less parsimonious explanation than recall of an encoded fact.

### B.5. The Role of Thinking-Specific Training

Our evaluation pools two distinct implementations of thinking: *native thinking*, where the model was optimized during training to use thinking tokens (Gemini-3, Gemini-2.5, GPT-5 families), and *CoT prompting*, where thinking is elicited at test time without dedicated training (Gemma3, GPT-4.1). The results suggest that these implementations yield qualitatively different effects. Native thinking consistently improves both availability and robustness, consistent with recall facilitation. CoT prompting produces mixed results: for Gemma3 models, it improves availability but degrades robustness, consistent with a response-diversity effect; for GPT-4.1, it yields modest gains in both.

This distinction has practical implications. The recall facilitation mechanism, in which thinking helps the model access encoded knowledge, may depend on thinking-specific train-

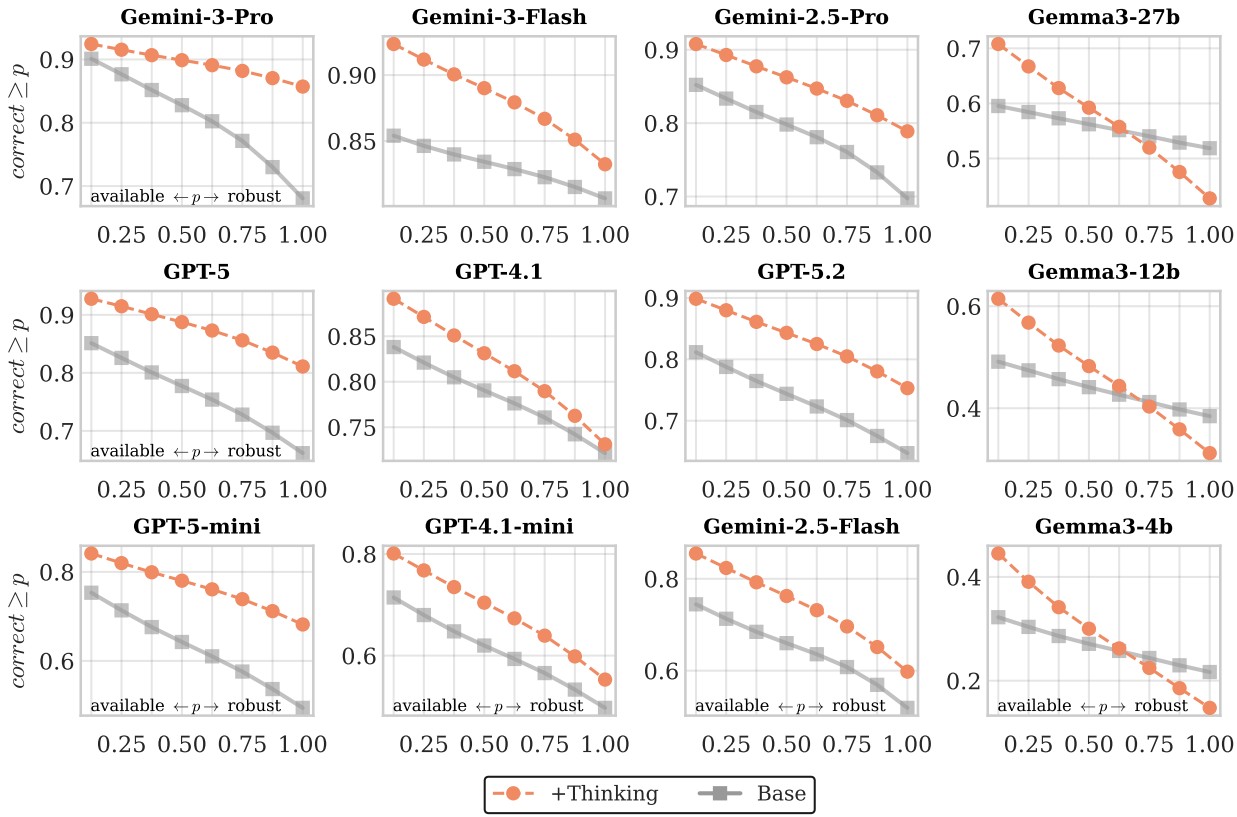

*Figure 12.* We report $correct \geq p$ for 12 LLMs, defined as the proportion of questions whose estimated probability of correctness across samples is at least $p$. Low values of $p$ reflect availability (the model produces at least one correct answer), while higher values reflect robustness (the model is correct consistently). Different LLMs exhibit distinct patterns; each column corresponds to a different pattern.

ing rather than merely on producing additional tokens at inference time. If so, the benefits of thinking for factuality are not a free lunch obtainable through prompting alone; they require dedicated training investment. We note, however, that our experimental design cannot fully disentangle the effects of thinking-specific training from other differences between model families (e.g., model size, data, post-training procedures). The pattern is suggestive but not conclusive.

## C. Additional Results

In this section, we clarify how each analysis, figure, and table is computed, and report results for all evaluated LLMs to support the main-text trends.

### C.1. Profiles

Figure 4 presents the profile distributions for the 13 LLMs, and Figure 13 provides a line plot that highlights trends. To estimate the distributions, we exclude facts with non-gradable questions (see Appendix D.1), then assign each fact a profile and compute the share of each profile. The *Knows* value sums the *Direct recall*, *recall with Thinking*, and *Inference without Thinking* profiles. It therefore slightly

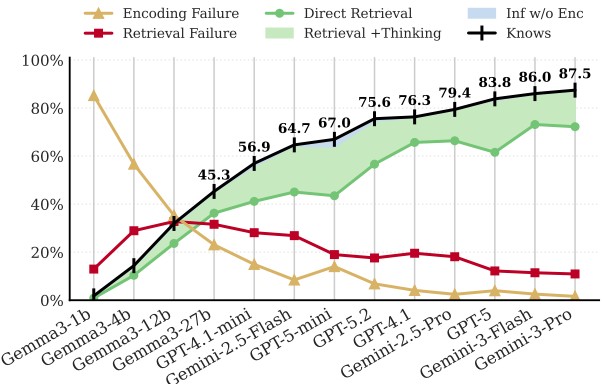

*Figure 13.* **Trends Identified by Profiles:** 'Knows' equal to the sum of 'Direct Recall' (green line), 'Recall with Thinking' (light green shade), and 'Inference without Encoding' (light blue shade). LLMs are sorted according to 'Knows'. Figure 4 presents distributions of the profiles.

overestimates the LLM's actual performance (with thinking enabled), since some facts become unknown when thinking is enabled. As such, *Knows* can be viewed as an LLM's knowledge potential.

## C.2. Fact Popularity

Popularity tiers (bottom 20%, top 20%) are defined based on Wikipedia page visit counts for the pages from which facts are extracted. This serves as a reasonable but imperfect proxy for fact incidence, and we expect the observed trends to be even stronger with a more direct measure. Figure 16 compares the encoding and recall gaps between the two popularity tiers across all 13 LLMs. Encoding performance is measured as the proportion of encoded facts within each tier, while recall performance is measured as the proportion of known facts conditioned on being encoded. This conditioning isolates recall from encoding, allowing us to assess whether rare facts are harder to recall even when they are encoded. As shown, for most LLMs (10 out of 13; excluding Gemma3 12b, 4b, and 1b), the recall gap is substantially larger than the encoding gap.

## C.3. Phrasing

A large body of work suggests that LLMs are sensitive to prompt phrasing (Deng et al., 2023; Lin et al., 2024b; Mizrahi et al., 2024; Habba et al., 2025; Lior et al., 2025). In contrast, in §5.2, we find that LLMs are insensitive to question rephrasing in our setup. Here we elaborate on this analysis. We conduct hypothesis tests comparing the distributions of performance across phrasings, treating success as a binary variable (1 if the question grade exceeds 0.5, 0 otherwise). In total, we perform 104 tests, covering 13 LLMs (with and without thinking) and four question pairs (direct vs. direct natural, reverse vs. reverse natural, and their multiple-choice variants). After applying False Discovery Rate (FDR) correction using the Benjamini–Yekutieli procedure with $q = 0.05$ (Benjamini & Yekutieli, 2001), none of the tests are significant. This indicates that, in our setup, phrasing does not confound knowledge assessment.

## C.4. Reverse Questions

Figure 16 reports recall performance conditioned on facts being encoded, for both multiple-choice and closed-book questions, under direct and reverse directions, across 13 LLMs. Each bar shows the fraction of encoded facts that are known under a given direction and task format. A fact is counted as known under the direct (respectively, reverse) condition only if the grades for both corresponding questions (direct and direct natural; reverse and reverse natural) exceed $g(q) > 0.5$. The same criterion is applied to the multiple-choice variants. Figure 16 shows that, for all LLMs, the gap between direct and reverse questions is much smaller for verification than for generation. For 9 out of 13 LLMs, reverse verification is in fact easier.

Our analysis comparing direct and reverse questions focuses on encoded facts, as this allows us to isolate recall failures.

One might argue that this conditioning introduces selection bias, since the encoding tasks share the same answer entity as direct questions. We therefore conduct several robustness analyses. First, all key results replicate when we do not condition on encoding: reverse generation remains substantially harder than direct generation, the gap is much smaller or absent for verification (with reverse verification often easier), and thinking mitigates the gap. Second, we control for answer type (e.g., person, location, etc) by computing performance separately for each type and then comparing reverse to direct questions; the same trends persist.

Finally, we analyze error attribution over all not-known facts and over those that are encoded. We decompose the sources of error into three categories: failure on direct questions only, failure on reverse questions only, or failure on both. As shown in Figure 14, across all LLMs, errors from reverse questions consistently exceed those from direct questions. For smaller LLMs, the dominant error source is failing 'both' questions. As scale increases, the share of 'both' decreases, and 'only reverse' becomes the dominant failure mode (40-50% are 'only reverse' and an additional 15-30% are 'both').

## C.5. Thinking

Figure 17 examines the impact of inference-time thinking on recall across 13 LLMs, focusing on popularity and directionality, respectively. In both analyses, we condition on encoded facts and decompose recall into baseline performance (facts known without thinking, given they are encoded) and the additional share that becomes known only with thinking. As shown, thinking mitigates the popularity gap for 8 out of 13 LLMs (excluding GPT-4.1-mini and the Gemma3 models) and the directional gap for 9 out of 13 LLMs (excluding Gemini-2.5-Flash and Gemma3 models). In both cases, the mitigation is substantial for frontier LLMs, that is, the stronger proprietary models.

# D. Methodological and Design Choices

The goal of this section is to describe and justify methodological details and design choices that influence our evaluation and analysis.

## D.1. Handling Partially and Other Grades

The question grade is computed as accuracy over CORRECT and INCORRECT responses only, excluding responses labeled PARTIALLY or OTHER. For example, if a question receives 3 CORRECT, 1 INCORRECT, 2 PARTIALLY, and 2 OTHER responses, its grade is $\frac{3}{3+1} = 0.75$. We exclude PARTIALLY and OTHER from grading because assigning them weights is subjective (Kalai et al., 2025).

We next describe how we handle PARTIALLY and OTHER

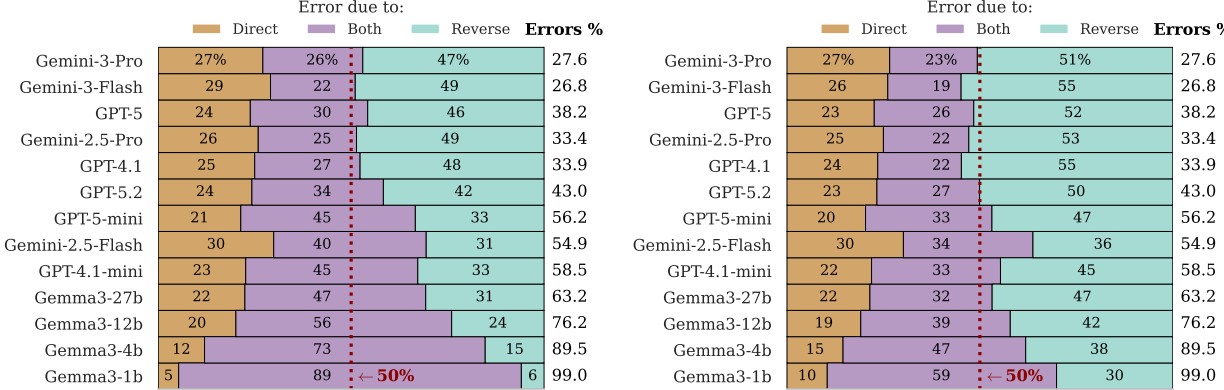

*Figure 14.* We decompose errors into three categories: failures on only direct questions, only reverse questions, or both. The dashed red line marks 50%. The left bar plot shows results for LLMs without thinking on all facts, while the right bar plot shows results for encoded facts. As shown, errors from reverse questions exceed those from direct questions, regardless of encoding.

responses at the fact level. We evaluated three strategies; Table 2 reports the resulting profile distributions and the fraction of excluded facts.

The first strategy excludes a fact if any of its six profiling questions is non-gradable, meaning all responses are PARTIALLY or OTHER. This approach removes a substantial number of facts, around 10% for frontier LLMs and over 20% for Gemma3 models.

The second strategy groups questions into task pairs (encoding, direct, and reverse) and excludes a fact only if an entire pair is non-gradable. This substantially reduces exclusions to 1–4% for nearly all LLMs, while producing profile distributions nearly identical to the first strategy.

The third strategy extends the second by assigning PARTIALLY responses a weight of 0.51, allowing facts to be considered known even when all responses are PARTIALLY. While this further reduces exclusions to below 1% and yields results very similar to the second strategy, it introduces subjective judgment that is harder to justify. We therefore adopt the second strategy throughout the paper.

Finally, in §2.2, we do not define a profile for facts that are not encoded but are known without thinking. Such cases are rare (less than 0.5% across all models; see the "Direct Inference" column in Table 2) and are likely due to sampling noise or grader error; excluding this profile does not change the results.

### D.2. Threshold Selection

Recall that both our definitions of *encodes* and *knows* rely on a threshold $g(q) > \tau$. A natural choice is $\tau = 0.5$, which corresponds to correctness being more likely than incorrectness. However, the threshold is a design choice, and different values can yield different profile distributions.

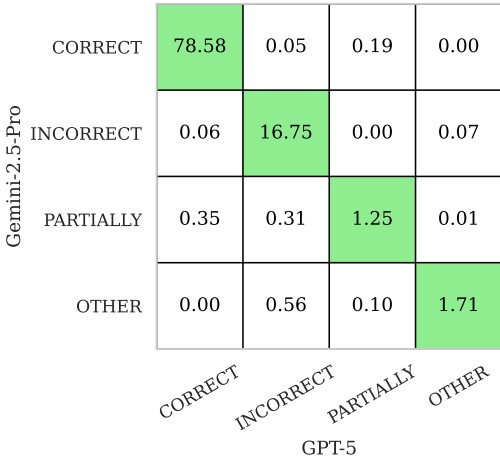

*Figure 15.* Confusion matrix between graders based on Gemini-2.5-Pro and GPT-5. Entries are percentages. Overall agreement is 98.2%, with most disagreements occurring when Gemini-2.5-Pro predicts OTHER or PARTIALLY.

To assess sensitivity to this choice, Figure 18 presents profile distributions for five additional reasonable values of $\tau$. While the exact proportions of profiles vary with $\tau$, our main conclusions are robust. Specifically, (1) encoding is nearly saturated for frontier LLMs, even under a conservative threshold of $\tau = 0.99$; (2) recall failures account for a substantial fraction of errors, and their relative impact increases under stricter thresholds; and (3) thinking consistently acts as a recovery mechanism, with over 10% of facts known only with thinking across all threshold choices.

### D.3. Grader Evaluation

Our evaluation relies on a prompted LLM grader, a common practice in short-form factuality evaluation (Wei et al., 2024a; Haas et al., 2025). We use Gemini-2.5-Pro with thinking as the primary grader. The prompt is adapted from

SimpleQA (Wei et al., 2024a) and refined through manual error analysis with Gemini-2.5-Pro. In this subsection, we assess robustness to the choice of the grader's underlying LLM by comparing our grader to a GPT-5-based grader with thinking (medium effort). We uniformly sample 4,160 responses across six tasks and all evaluated LLMs. The two graders agree on 98.2% of responses.

Figure 15 shows the confusion matrix between the two graders. Most disagreements occur when Gemini-2.5-Pro assigns OTHER or PARTIALLY, while GPT-5 assigns a definitive label. To better understand these disagreements, we manually examine them; representative examples are shown in Box D.1. In most cases, Gemini-2.5-Pro yields the more accurate label, which is unsurprising given that the prompt is optimized for it. In addition, it occasionally draws on external knowledge.

Overall, since the graders agree on the vast majority of responses and the remaining disagreements primarily involve OTHER or PARTIALLY labels, which are excluded from our analysis, we conclude that our results are not sensitive to the choice of the grader's underlying LLM.

### D.4. Multiple Response Generation

For each task, we generate $n = 8$ responses per question for three reasons. First, our goal is to assess factual knowledge at the level of facts; a single response is a noisy estimator and may misclassify a fact as not encoded or not known due to sampling variance. Second, because our questions are not single-token predictions, probability-based estimates are not well defined, and token-level likelihoods are inaccessible for most proprietary LLMs, making repeated sampling a natural alternative. Finally, multiple responses are required to distinguish availability (at least one correct response) from robustness (all responses correct), which is part of our analysis of inference-time thinking.

To assess the effect of multiple sampling and justify our choice of $n = 8$, we analyze how the fraction of facts with $g(q) > 0.5$ varies with $n$ across four tasks: two encoding tasks (completion and contextual) and two knowledge tasks (direct and reverse). For each task and model, we perform 1,000 bootstrap resamples, each time subsampling $n$ responses per question and recomputing the fraction of facts with $g(q) > 0.5$. Figure 19 shows the width of the 90% empirical bootstrap confidence interval (95th–5th percentiles) as a function of $n$. Across the evaluated models, the interval width falls below 1% for all tasks at $n = 8$, indicating that the estimated fraction is stable to within $\pm 0.5\%$. We therefore use $n = 8$ throughout the paper.

*Table 2.* Profile distributions under three strategies for handling PARTIALLY and OTHER responses. The three rightmost columns report the share of 'Excluded' facts, decomposed into 'OTHER or PARTIALLY' responses and 'Direct Inference' cases. The columns 'Knows (+Think)' and 'Knows' are computed with 'Direct Inference'.

| LLM | Encodes | Knows (+Think) | Knows | Encoding Failure | Recall Failure | Direct Recall | Recall w/ Thinking | Inference w/o Encoding | Excluded | OTHER or PARTIALLY | Direct Inference |
|---|---|---|---|---|---|---|---|---|---|---|---|
| **PARTIALLY: Assign NaN. OTHER: Assign NaN. NaNs Strategy: Ignore the entire fact.** | | | | | | | | | | | |
| Gemini-3-Pro | 98.3 | 87.4 | 72.4 | 1.5 | 11.2 | 72.3 | 14.9 | 0.1 | 5.2 | 5.1 | 0.1 |
| Gemini-3-Flash | 98.1 | 86.6 | 73.4 | 1.8 | 11.7 | 73.3 | 13.1 | 0.1 | 9.5 | 9.5 | 0.0 |
| Gemini-2.5-Pro | 98.1 | 79.7 | 66.1 | 1.9 | 18.5 | 66.0 | 13.5 | 0.1 | 7.4 | 7.3 | 0.1 |
| Gemini-2.5-Flash | 92.8 | 65.8 | 46.0 | 6.9 | 27.4 | 46.0 | 19.5 | 0.3 | 13.3 | 13.2 | 0.0 |
| GPT-5.2 | 92.8 | 75.7 | 56.8 | 6.3 | 18.1 | 56.6 | 18.0 | 0.9 | 8.2 | 8.0 | 0.2 |
| GPT-5 | 95.9 | 84.5 | 62.0 | 3.3 | 12.4 | 61.8 | 21.8 | 0.7 | 6.8 | 6.6 | 0.2 |
| GPT-5-mini | 83.5 | 67.9 | 44.2 | 13.4 | 19.0 | 44.0 | 20.6 | 3.1 | 7.2 | 7.0 | 0.2 |
| GPT-4.1 | 96.0 | 77.3 | 64.9 | 3.7 | 19.1 | 64.9 | 12.0 | 0.4 | 9.2 | 9.1 | 0.1 |
| GPT-4.1-mini | 85.7 | 58.1 | 41.7 | 13.4 | 28.7 | 41.5 | 15.4 | 0.9 | 11.1 | 10.9 | 0.2 |
| Gemma3-27b | 80.1 | 47.3 | 37.3 | 19.5 | 33.4 | 37.0 | 9.7 | 0.4 | 21.0 | 20.7 | 0.3 |
| Gemma3-12b | 68.0 | 33.4 | 24.6 | 31.1 | 35.6 | 24.5 | 7.9 | 1.0 | 27.2 | 27.1 | 0.1 |
| Gemma3-4b | 46.3 | 16.0 | 11.0 | 53.5 | 30.6 | 10.9 | 4.9 | 0.2 | 27.2 | 27.1 | 0.1 |
| Gemma3-1b | 17.0 | 1.8 | 1.1 | 82.7 | 15.5 | 1.1 | 0.4 | 0.3 | 45.8 | 45.8 | 0.0 |
| **\*\*Selected\*\* PARTIALLY: Assign NaN. OTHER: Assign NaN. NaNs Strategy: Ignore only if both tasks are NaNs. \*\*Selected\*\*** | | | | | | | | | | | |
| Gemini-3-Pro | 98.1 | 87.7 | 72.4 | 1.6 | 10.9 | 72.2 | 14.9 | 0.3 | 1.3 | 1.1 | 0.2 |
| Gemini-3-Flash | 97.2 | 86.1 | 73.2 | 2.5 | 11.4 | 73.2 | 12.6 | 0.2 | 3.2 | 3.1 | 0.1 |
| Gemini-2.5-Pro | 97.5 | 79.6 | 66.6 | 2.5 | 18.1 | 66.4 | 13.0 | 0.1 | 1.3 | 1.2 | 0.2 |
| Gemini-2.5-Flash | 91.1 | 64.8 | 45.1 | 8.4 | 26.9 | 45.0 | 19.2 | 0.5 | 1.9 | 1.8 | 0.1 |
| GPT-5.2 | 92.1 | 75.9 | 57.0 | 6.8 | 17.6 | 56.6 | 17.9 | 1.0 | 2.0 | 1.6 | 0.3 |
| GPT-5 | 95.3 | 84.1 | 61.8 | 4.0 | 12.2 | 61.6 | 21.5 | 0.8 | 1.6 | 1.3 | 0.2 |
| GPT-5-mini | 82.8 | 67.3 | 43.8 | 14.0 | 19.0 | 43.5 | 20.3 | 3.2 | 1.8 | 1.5 | 0.3 |
| GPT-4.1 | 95.6 | 76.7 | 66.1 | 4.1 | 19.6 | 65.7 | 10.3 | 0.3 | 2.5 | 2.1 | 0.4 |
| GPT-4.1-mini | 83.8 | 57.3 | 41.5 | 14.9 | 28.1 | 41.1 | 14.6 | 1.2 | 2.6 | 2.2 | 0.3 |
| Gemma3-27b | 76.4 | 45.8 | 36.8 | 23.1 | 31.6 | 36.2 | 8.6 | 0.5 | 3.5 | 3.0 | 0.5 |
| Gemma3-12b | 63.8 | 32.1 | 23.8 | 35.3 | 32.8 | 23.6 | 7.4 | 0.9 | 3.8 | 3.6 | 0.1 |
| Gemma3-4b | 42.9 | 14.6 | 10.5 | 56.7 | 28.9 | 10.4 | 3.6 | 0.4 | 5.3 | 5.1 | 0.2 |
| Gemma3-1b | 14.3 | 1.9 | 1.0 | 85.2 | 13.0 | 0.9 | 0.5 | 0.5 | 10.0 | 9.8 | 0.1 |
| **PARTIALLY: 0.51 Weight. OTHER: Assign NaN. NaNs Strategy: Ignore only if both tasks are NaNs.** | | | | | | | | | | | |
| Gemini-3-Pro | 98.2 | 88.7 | 74.3 | 1.5 | 9.8 | 74.2 | 14.2 | 0.2 | 0.2 | 0.2 | 0.0 |
| Gemini-3-Flash | 97.5 | 87.0 | 75.4 | 2.3 | 10.9 | 75.2 | 11.4 | 0.2 | 0.5 | 0.3 | 0.2 |
| Gemini-2.5-Pro | 97.7 | 81.0 | 68.5 | 2.2 | 16.9 | 68.3 | 12.4 | 0.1 | 0.1 | 0.0 | 0.1 |
| Gemini-2.5-Flash | 92.3 | 67.1 | 47.2 | 7.3 | 25.8 | 47.0 | 19.5 | 0.5 | 0.8 | 0.6 | 0.2 |
| GPT-5.2 | 92.6 | 76.3 | 57.8 | 6.4 | 17.5 | 57.6 | 17.5 | 1.0 | 0.7 | 0.6 | 0.2 |
| GPT-5 | 95.7 | 85.3 | 63.4 | 3.5 | 11.3 | 63.2 | 21.1 | 0.7 | 0.4 | 0.1 | 0.2 |
| GPT-5-mini | 84.1 | 68.7 | 45.1 | 12.9 | 18.8 | 44.7 | 20.6 | 3.0 | 0.8 | 0.4 | 0.4 |
| GPT-4.1 | 96.1 | 77.5 | 67.2 | 3.7 | 19.1 | 66.9 | 10.0 | 0.2 | 0.5 | 0.2 | 0.3 |
| GPT-4.1-mini | 85.2 | 58.4 | 42.7 | 13.5 | 28.5 | 42.4 | 14.4 | 1.3 | 1.0 | 0.6 | 0.4 |
| Gemma3-27b | 78.8 | 47.2 | 38.6 | 20.8 | 32.7 | 37.9 | 8.1 | 0.4 | 1.8 | 1.2 | 0.7 |
| Gemma3-12b | 66.5 | 33.5 | 25.7 | 32.7 | 34.1 | 25.4 | 7.1 | 0.8 | 1.7 | 1.4 | 0.3 |
| Gemma3-4b | 48.3 | 15.6 | 11.4 | 51.5 | 33.2 | 11.1 | 4.0 | 0.2 | 3.2 | 2.8 | 0.3 |
| Gemma3-1b | 21.6 | 2.5 | 1.4 | 78.0 | 19.7 | 1.3 | 0.6 | 0.4 | 7.9 | 7.7 | 0.2 |

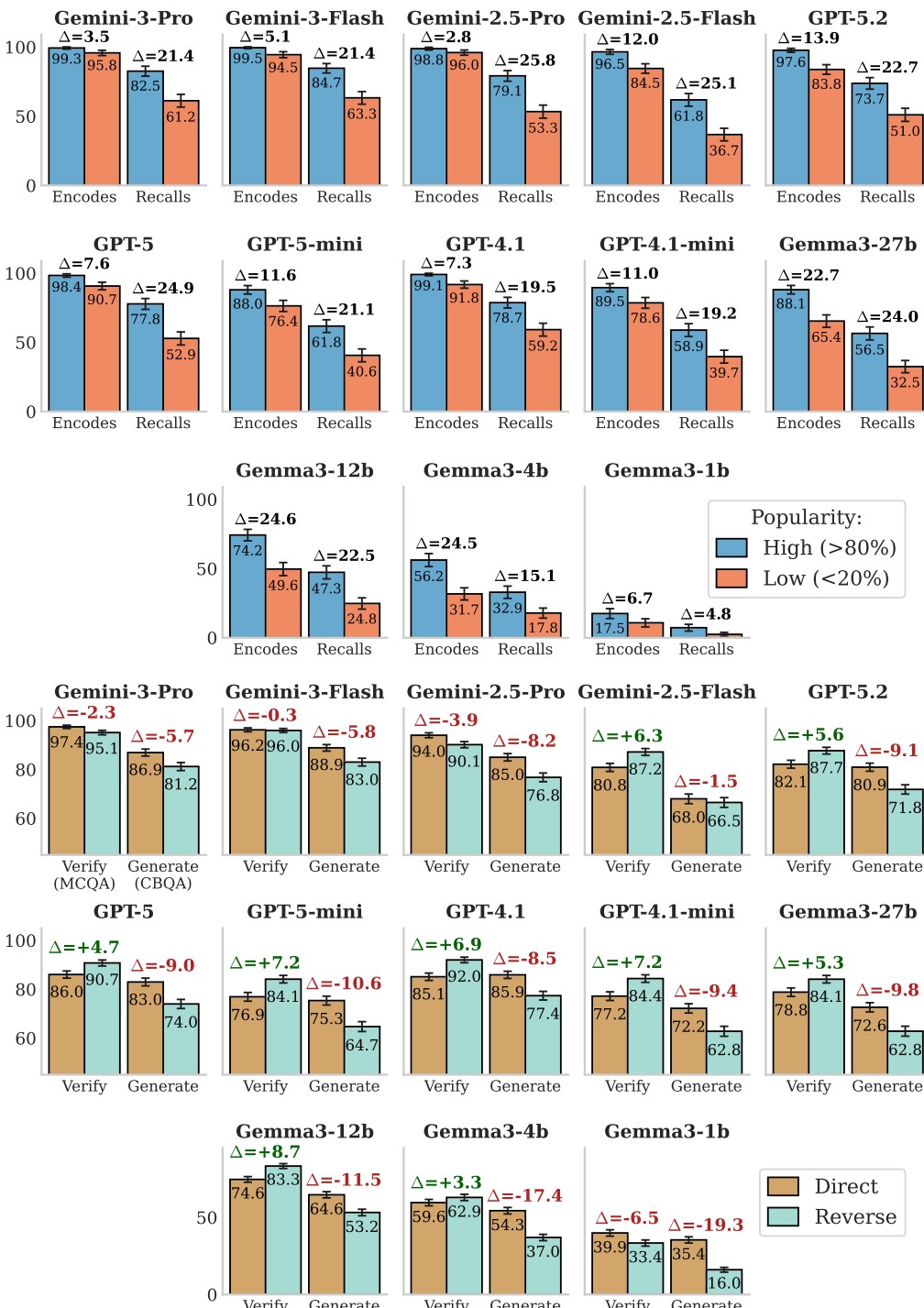

Figure 16. *Top:* We compare two popularity tiers (bottom 20% vs. top 20%) in terms of encoding rates and direct recall rates (knowing encoded facts without thinking). The Δ indicates the gap between tiers. *Bottom:* We compare direct recall rates on direct and reverse questions across two tasks: verification (multiple-choice) and generation (closed-book). The Δ denotes the gap between the directions.

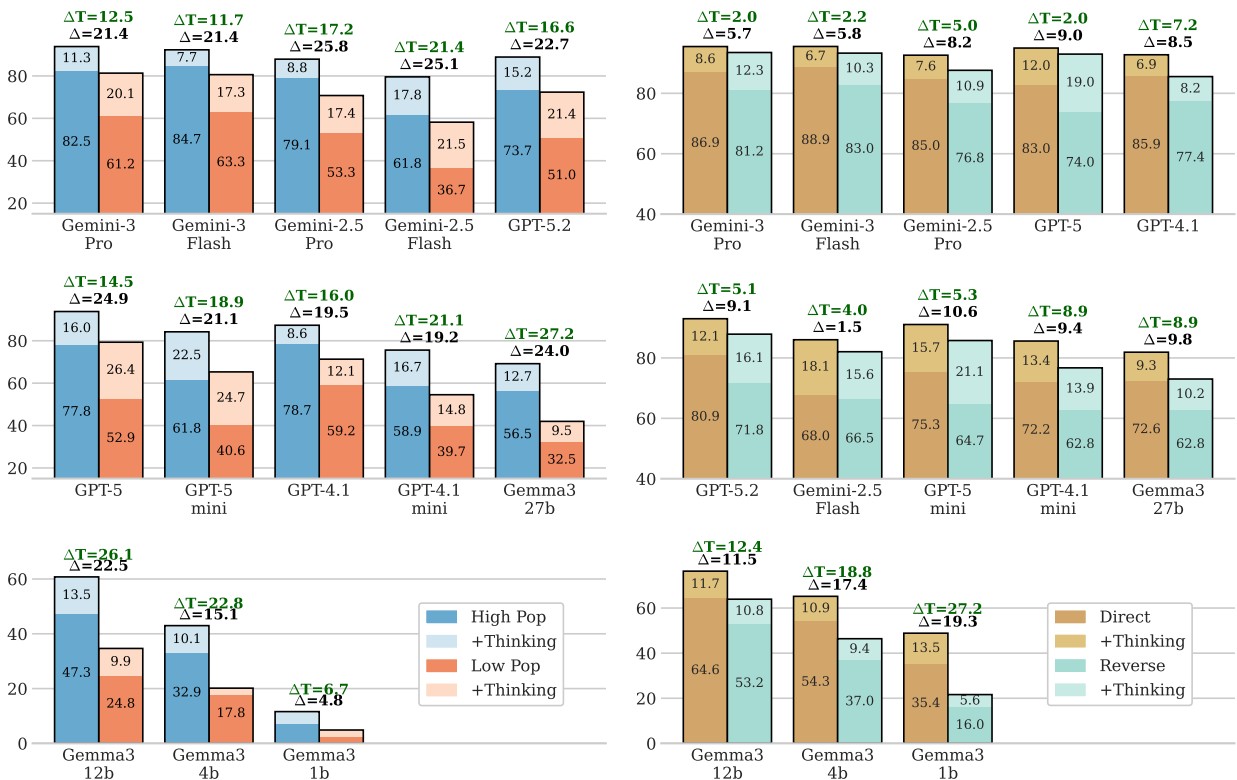

*Figure 17.* We examine the impact of thinking on retrieval (knowing encoded facts). On the *left*, we compare two popularity tiers (bottom 20% vs. top 20%), while on the *right*, we compare direct and reverse questions. The popularity or directional gaps are denoted by $\Delta$ (no thinking) and $\Delta T$ (with thinking).

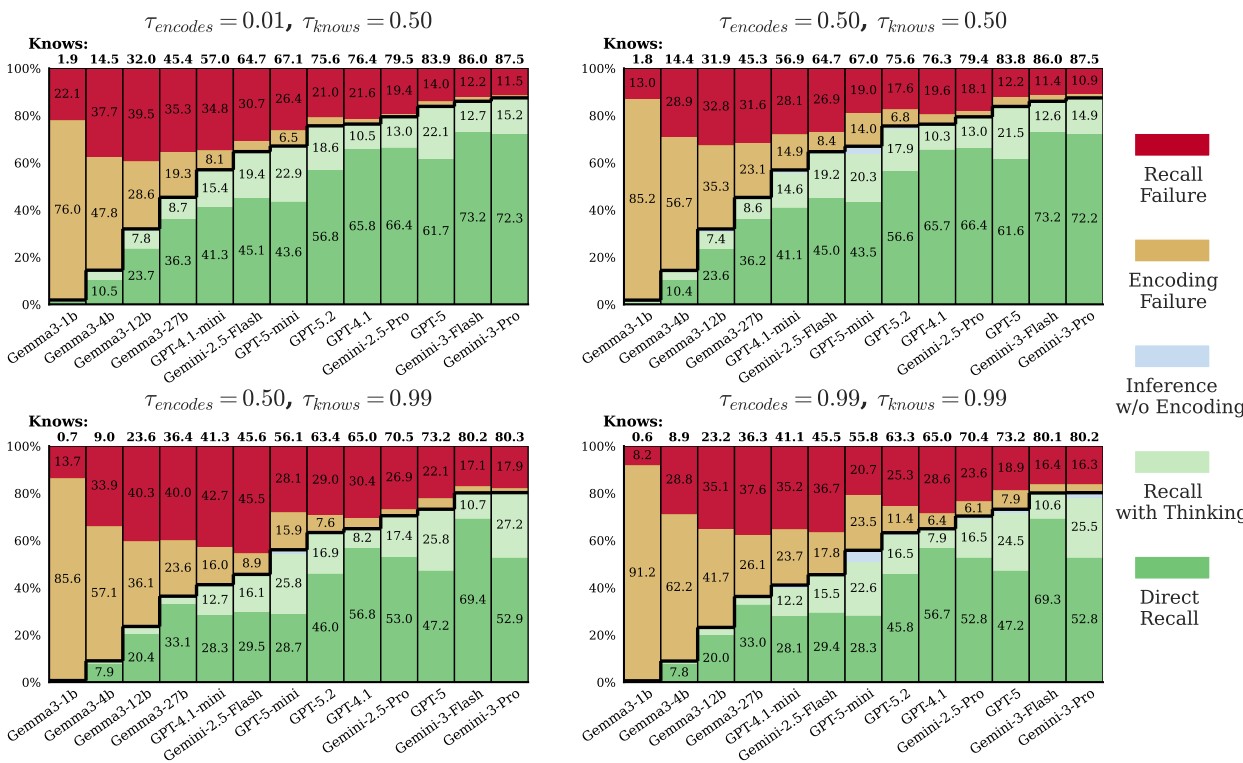

*Figure 18.* Distributions (percentages) of knowledge profiles across 13 LLMs under different $g(q) > \tau$ thresholds for encoding and knowledge. While the exact profile percentages vary with $\tau$, our main findings are robust to the choice of threshold.

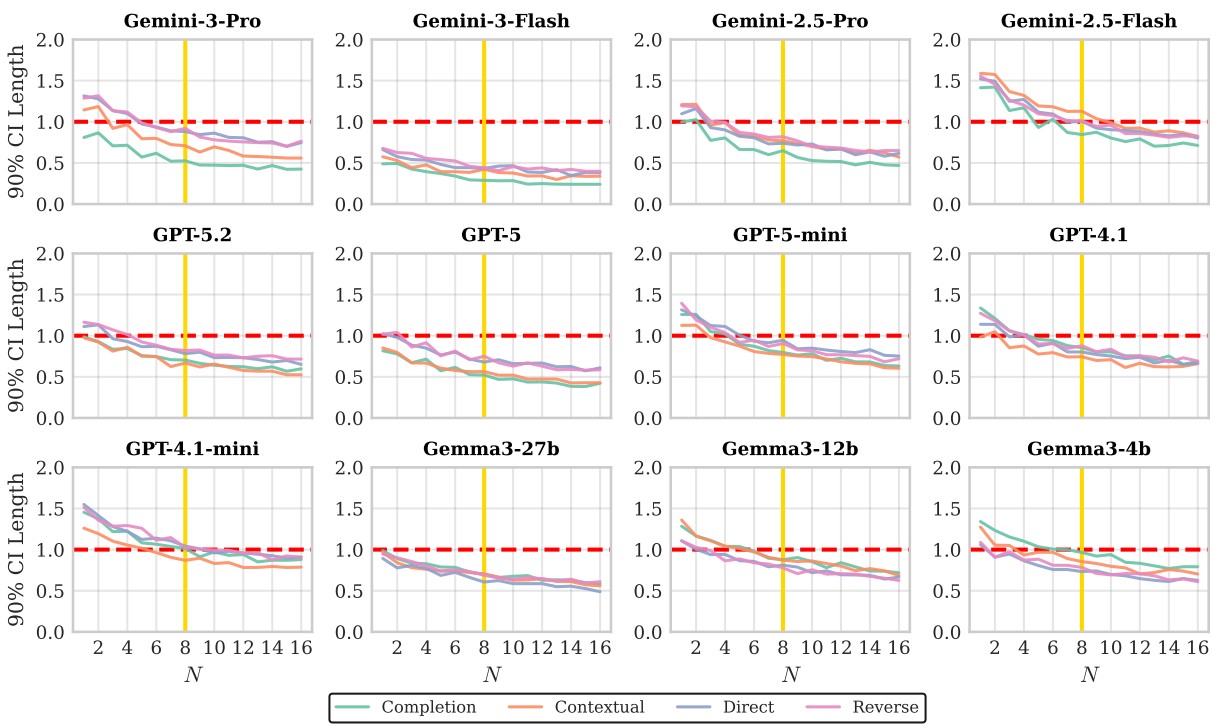

*Figure 19.* Sensitivity of the fraction of facts with $g(q) > 0.5$ to the number of sampled responses $n$, shown as the width of the 90% empirical bootstrap confidence interval (95th–5th percentiles) across eight LLMs. As shown, the interval width drops below 1% at $n = 8$.

---

**Box D.1: Analysis of Disagreements Between Graders**

**Direct Question:** Who developed the circuit for the Cockcroft–Walton generator in 1919?
**Gold Answer:** Heinrich Greinacher
**Prediction of Gemini-3-Pro:** The circuit itself was essentially discovered by Heinrich Greinacher, a Swiss physicist, in 1919.
**Better Grader:** Gemini (CORRECT vs GPT-5 INCORRECT)
**Rationale:** The prediction correctly identifies Greinacher; GPT-5 likely penalized the shift from "developed" to "discovered".

- - - - - - - - - - - - - - - - - - - - - - - - - - - - - - - - - - - - - - - - - - - - - - - - - - -

**Direct Natural Question:** About how many iPhone 11s had sold as of March 2022?
**Gold Answer:** 159.2 million
**Prediction of Gemini-2.5-Pro+Thinking:** Approximately 160 million units.
**Better Grader:** Gemini (CORRECT vs GPT-5 INCORRECT)
**Rationale:** Rounding 159.2 to "Approximately 160" is a permissible and align with the guidelines.

- - - - - - - - - - - - - - - - - - - - - - - - - - - - - - - - - - - - - - - - - - - - - - - - - - -

**Reverse Question:** What district within the Lahore Division of Punjab, Pakistan has a total area of 3,995 square kilometres?
**Gold Answer:** Kasur District
**Prediction of Gemini-2.5-Pro+Thinking:** Kasur
**Better Grader:** Gemini (CORRECT vs GPT-5 PARTIALLY)
**Rationale:** Dropping "District" is explicitly allowed by the guidelines.

- - - - - - - - - - - - - - - - - - - - - - - - - - - - - - - - - - - - - - - - - - - - - - - - - - -

**Reverse Question:** What is the name of the female version of Taskmaster who appears in Black Widow?
**Gold Answer:** Antonia Dreykov
**Prediction of Gemini-2.5-Flash:** Taskmaster
**Better Grader:** Gemini (OTHER vs GPT-5 INCORRECT)
**Rationale:** The prediction does not contradict the question (she is Taskmaster).

- - - - - - - - - - - - - - - - - - - - - - - - - - - - - - - - - - - - - - - - - - - - - - - - - - -

**Reverse Question:** Whose activities did the Federal Bureau of Investigation (FBI) program COINTELPRO attempt to subvert by placing a counterintelligence operative in the local Panthers organization?
**Gold Answer:** Fredrick Allen Hampton
**Prediction of GPT-5-mini+Thinking:** The Black Panther Party
**Better Grader:** Gemini (PARTIALLY vs GPT-5 INCORRECT)
**Rationale:** Identifying the group (Black Panthers) instead of the leader (Hampton) is a "more general version" of the target.

- - - - - - - - - - - - - - - - - - - - - - - - - - - - - - - - - - - - - - - - - - - - - - - - - - -

**Reverse Natural Question:** Who watched 'The UFO Incident' on the night Mike Rogers admitted the job was seriously behind schedule?
**Gold Answer:** Walton and Rogers
**Prediction of GPT-4.1+Thinking:** Travis Walton and the other logging crew members (including Mike Rogers) watched 'The UFO Incident' together that night.
**Better Grader:** Gemini (CORRECT vs GPT-5 INCORRECT)
**Rationale:** Including the rest of the crew does not contradict the presence of Walton and Rogers.

- - - - - - - - - - - - - - - - - - - - - - - - - - - - - - - - - - - - - - - - - - - - - - - - - - -

**Direct Question:** What is the compound wall that surrounds the St. Antony's Syro-Malabar Catholic Forane Church at Ollur called?
**Gold Answer:** Elephant Wall
**Prediction of Gemma3-4b+Thinking:** Perimeter Wall
**Better Grader:** Gemini (OTHER vs GPT-5 INCORRECT)
**Rationale:** "Perimeter Wall" is missing the specific name, but it is not a factual contradiction (it *is* a perimeter wall).

- - - - - - - - - - - - - - - - - - - - - - - - - - - - - - - - - - - - - - - - - - - - - - - - - - -

**Reverse Question:** What statue depicting Vairocana Buddha is located within the Fodushan Scenic Area?
**Gold Answer:** Spring Temple Buddha
**Prediction of Gemma3-1b+Thinking:** Fodushan Scenic Area, specifically the Temple of the White Horse, where the statue of Vairocana Buddha is located.
**Better Grader:** Gemini (INCORRECT vs GPT-5 OTHER)
**Rationale:** The prediction places the statue in a completely different temple (White Horse).

**Direct Question:** Who built the town, originally called Beodericsworth, on a grid pattern?
**Gold Answer:** Abbot Baldwin
**Prediction of Gemma3-1b+Thinking:** The people of Beodericsworth built the town.
**Better Grader:** Gemini (OTHER vs GPT-5 INCORRECT)
**Rationale:** "The people built it" is a vague truism that fails to answer the "Who" (Architect/Leader) but doesn't contradict the answer.

- - - - - - - - - - - - - - - - - - - - - - - - - - - - - - -

**Direct Natural Question:** Which institution was the fossil specimen "Apex" loaned to in December 2024?
**Gold Answer:** American Museum of Natural History
**Prediction of Gemini-2.5-Pro:** According to the provided information, the fossil specimen "Apex" was loaned in December 2022, not December 2024. Therefore, I cannot answer your question.
**Better Grader:** Gemini (INCORRECT vs GPT-5 OTHER)
**Rationale:** The prediction explicitly contradicts the question (claiming it is wrong).

- - - - - - - - - - - - - - - - - - - - - - - - - - - - - - -

**Direct Question:** What existing highway was US 1 mostly assigned to when the road system was laid out?
**Gold Answer:** Atlantic Highway
**Prediction of Gemma3-1b+Thinking:** US 1 was primarily assigned to transport goods.
**Better Grader:** Gemini (OTHER vs GPT-5 INCORRECT)
**Rationale:** Stating the purpose ("transport goods") instead of the highway name is irrelevant (Other).

- - - - - - - - - - - - - - - - - - - - - - - - - - - - - - -

**Reverse Natural Question:** What's the name of the Hindu temple complex in Delhi that opened on November 6, 2005?
**Gold Answer:** Swaminarayan Akshardham
**Prediction of Gemini-2.5-Pro:** Akshardham
**Better Grader:** GPT-5 (CORRECT vs Gemini PARTIALLY)
**Rationale:** "Swaminarayan" is a modifier and dropping it is allowed.

- - - - - - - - - - - - - - - - - - - - - - - - - - - - - - -

**Direct Question:** What was the name of the exposition at which Westinghouse Electric showcased its AC system to illuminate the "White City"?
**Gold Answer:** 1893 Columbian Exposition
**Prediction of Gemini-2.5-Flash:** The World's Columbian Exposition
**Better Grader:** GPT-5 (CORRECT vs Gemini PARTIALLY)
**Rationale:** "World's Columbian Exposition" is the formal name for the "1893 Columbian Exposition".

- - - - - - - - - - - - - - - - - - - - - - - - - - - - - - -

**Reverse Question:** On which album did bassist Danny Thompson have his first collaboration on record with John Martyn?
**Gold Answer:** The Road to Ruin
**Prediction of Gemma3-4b+Thinking:** Sure
**Better Grader:** GPT-5 (OTHER vs Gemini INCORRECT)
**Rationale:** "Sure" is a refusal or nonsense filler.

## E. Prompts

### E.1. Facts Extraction Prompts

**Box E.1: Prompt for Classifying Pages into Categories**

```
You will be provided with a document and its title.
Your task is to analyze the document and assign it to the single, most appropriate
    category from the predefined list below.

----

**Categories and Descriptions:**

*   **People:** Covers specific individuals, their biographies, life stories,
```

```
         achievements, and influence.
    *    **History:** Covers past events and historical periods.
    *    **Geography:** Covers Earth's physical features, locations, countries, and
         cities.
    *    **Arts:** Covers all forms of creative expression, including literature, music,
         visual arts, film, and performance.
    *    **Philosophy & Religion:** Covers fundamental principles, belief systems, ethics
         , religions, and mythological narratives.
    *    **Everyday Life:** Covers the practical and routine aspects of human life,
         including food, clothing, family, sports, and hobbies.
    *    **Society:** Covers the structures and systems that govern human communities,
         such as politics, law, economics, and language.
    *    **Science:** Covers the study of the natural and physical world, including
         biology, physics, chemistry, and mathematics.
    *    **Technology:** Covers the application of scientific knowledge, including
         engineering, computers, and inventions.
    *    **Other:** Covers topics that serve as a catch-all and do not clearly fit into
         any of the other categories.

    ----

    **Output Format:**

    Your response must consist of **ONLY** one category name. Do not add any explanation
        , punctuation, or other text.

    Title: {page_title.}
    Content: {summary}
```

---

### Box E.2: Prompt for Performing Named Entity Recognition

```
    You will be provided with two parts of a paragraph.
    Your task is to perform Named Entity Recognition on the *second* part of the
        paragraph.
    Please follow these instructions carefully.

    ### **Instructions:**

    1.   Read the two parts provided at the end of this prompt.
    2.   From the second part, identify and select up to a maximum of **eight** entities.
    3.   Do not select lengthy entities (more than five words) or entities that are not
         named entities.
    4.   Select specific core entities and avoid overly descriptive phrases that combine
         multiple separate concepts (e.g., select "documentary photographer" and not "
         feminist social documentary photographer").
    5.   For each entity you select, you must assign an entity type from the following
         specific list.
    6.   You should return a new version of the second part of the paragraph. Copy-paste
         the original second part and replace the entities you chose with the format
         `{{{{entity_name; entity_type}}}}`

    ----

    ### **Entity Types:**

    *    **PERSON:** An individual human, whether real or fictional.
    *    **ORGANIZATION:** A structured company, institution, or collective with a common
          purpose.
    *    **LOCATION:** A specific geographical place, region, or feature on Earth.
    *    **EVENT:** A notable occurrence or incident that happens at a particular time
```

```
          and place.
  *    **WORK_OF_ART:** A specific creative work, such as a book, movie, song, or
       painting.
  *    **PRODUCT:** A commercially produced good, food, service, or piece of technology
       .
  *    **ORGANISM:** A non-human biological life form, including animals, plants, and
       microbes.
  *    **SUBSTANCE:** A material with a distinct physical or chemical composition.
  *    **DATE:** A specific point in time, including full dates, years, or historical
       periods.
  *    **NUMERIC_VALUE:** A number representing a specific quantity, measurement, or
       other quantifiable value.
  *    **CONDITION:** A specific state of being, typically a medical, physical, or
       operational status.
  *    **TITLE:** An official position, role, award, or honorific held by a person or
       group.
  *    **LANGUAGE:** Natural languages, dialects, writing systems, including phrases.
  *    **GROUP:** A collective of people defined by a shared nationality, ethnicity, or
        cultural affiliation.
  *    **LAW:** A specific, formally enacted rule, regulation, or statute created by a
       governing body.
  *    **IDEA:** An abstract concept, belief system, named phenomenon, or cultural
       practice.
  *    **CLASSIFICATION:** A specific type, genre, format, class or list used to
       categorize other entities.
  *    **IDENTIFIER:** A unique name, code, or alias used as a formal reference or
       label for another entity
  *    **OTHER:** A catch-all category for named entities that do not fit into any of
       the other defined types.

  ----

  ### **Required Output Format:**

  Copy-paste the second part of the paragraph. In its new version, replace the
      entities you chose with the format `{{{{entity_name; entity_type}}}}`, for
      example: {{{{tennis player; TITLE}}}}. The `entity_name` must be exactly as it
      appears in the original text.

  ----

  **Example:**

  *Part 1:*
  ...

  *Part 2*
  `Liam Tarquin Broady (BROH-dee; born 4 January 1994) is a British professional
      tennis player who competes mainly on the ATP Challenger Tour. He reached a
      career high ranking of world No. 93 on 25 September 2023.`

  *Correct Output:*
  {{{{Liam Tarquin Broady; PERSON}}}} (BROH-dee; born {{4 January 1994; DATE}}) is a
      {{{{British; GROUP}}}} professional {{{{tennis player; TITLE}}}} who competes
      mainly on the {{{{ATP Challenger Tour; EVENT}}}}. He reached a career high
      ranking of world No. {{{{93; NUMERIC_VALUE}}}} on 25 September 2023.

  Explanation:
  1.   **Liam Tarquin Broady**: This is the full name of an individual human, therefore
       , it as a **PERSON**.
  2.   **4 January 1994**: This is a specific point in time, a full **DATE**.
  3.   **British**: This word describes a nationality or cultural affiliation.
       According to the definitions, this is a **GROUP**.
```

4.  **tennis player**: This is a title describing a person's profession or role. The
    correct type is **TITLE**.
5.  **ATP Challenger Tour**: This is a named series of tennis tournaments, which is
    a type of recurring **EVENT**.
6.  **93**: This is a number representing a specific quantity (a world ranking),
    therefore, it is a **NUMERIC_VALUE**.

**Instruction End**

**Please process the following paragraph and identify up to eight entities:**
{paragraph}

---

**Box E.3: Prompt for Selecting Object Entities**

You will be provided with a paragraph annotated with entities in the format of `{{{{
    entity_name}}}}`.
Your task is to select up to {n_entities} **OBJECT** entities for the pre-training
    completion task (causal language modeling).
Each OBJECT entity should complete a natural "fill-in-the-blank" based on the
    preceding text: the *left-context*.
Specifically, the left-context should clearly indicate **what kind of entity** (e.g
    ., a person, country, or biological species) is expected at that position.
However, correctly identifying the **specific entity value** (e.g., "South Africa",
    "banana plants") should require recalling a world fact.
These selected entities serve as factual anchors for generating our suite of
    completion-based and QA-based robustness tests.

### **Instructions for selecting suitable OBJECT entities**

* Select OBJECT entities that involve recalling factual knowledge, and is not
    trivial to guess.
* The left-context (the text that precedes the entity) must suggest **what kind** of
     entity is expected (e.g., a date, a country, a person's name, a title, a
    profession, a species, an event, a number, etc).
* The left-context should contain enough information to strongly suggest a single,
    unique, unambiguous completion.
* OBJECT entities whose specific value can be predicted solely based on the "left-
    context" or are too easy to guess are considered poor choices.
  * *Example:* `Bang bang chicken (Chinese: ...) is a popular dish in {{{{Chinese
    cuisine}}}}` -> {{{{Chinese cuisine}}}} is a poor choice because the Chinese
    name makes the answer too obvious.
  * *Example:* `Airlink is based in Johannesburg, {{{{South Africa}}}}` -> {{{{South
    Africa}}}} is a poor choice because it can be easily guessed from the mention
    of Johannesburg, without requiring any knowledge about Airlink.
* The OBJECT should have a strong relationship with an entity mentioned earlier in
    the text.
  * *Example:* `{{{{Panama disease}}}}... infects {{{{banana plants}}}}` -> Knowing
    what Panama disease is makes "banana plants" a highly predictable object of
    the verb "infects".
* Do not select OBJECT entities from the start of a sentence. Ideally, the OBJECT
    entity should appear at the end of a sentence or complete clause.
* Ensure the left-context provides sufficient disambiguation to yield a unique and
    specific completion. If the blank could plausibly be filled by multiple entity
    types, it is not a suitable choice.
  * *Example:* The left-context "Michael Jordan was born on..." is ambiguous, as
    multiple people share this name (e.g., the basketball player and the ML
    researcher). In contrast, "Michael Jeffrey Jordan was born on..." and "Michael
     Jordan, the famous basketball player, was born on..." clearly indicate a
    single referent, enabling a unique factual prediction.
  * *Example:* `Liam Tarquin Broady is a...` -> This is a poor choice. The context

```
          is too broad and could be completed with multiple entities, such as a
          profession (tennis player), nationality (British), or a descriptor (left-
          handed).
* The immediate context preceding the entity should constrain the expected entity
     type, strongly signaling what kind of entity is likely to follow. The OBJECT
     entity can follow a preposition that narrows down the possibilities.
   * *Example:* `...born on {{{{4 January 1994}}}}` -> The phrase "born on" strongly
        suggests a date.
   * *Example:* `...caused by the fungus {{{{Fusarium oxysporum f. sp. cubense}}}}`
        -> The phrase "the fungus" clearly indicates that a specific fungal name
        should follow.
   * *Example:* `The Virginia Plan (also known as the {{{{Randolph Plan}}}})...` ->
        The phrase "also known as" implies an alternative name is expected.
* Do not consider any information that appears after the OBJECT entity when making
     your selection.
* If the uniqueness or the disambiguation of the entity type relies on information
     that appears after the entity, it is a poor OBJECT choice.
   * *Example:* `It is their first album to feature {{{{Matt Wachter}}}} on bass.` ->
        {{{{Matt Wachter}}}} is a poor choice. The phrase "It is their first album to
        feature..." does not constrain the prediction. The disambiguating detail "on
        bass" comes *after* the entity, which violates the instruction that the **
        OBJECT** must be inferable solely from the left-context.
* You must select up to {n_entities} suitable OBJECT entities and return them in a
     list, sorted from the most to the least suitable.

----

### **Examples**

### **Text 1**
{{{{Panama disease}}}} (or Fusarium wilt) is a plant {{{{disease}}}} that infects
     {{{{banana plants}}}} ({{{{Musa spp.}}}}). It is a wilting disease caused by the
      fungus {{{{Fusarium oxysporum f. sp. cubense}}}} (Foc). During the {{{{1950s
      }}}}, an outbreak of Panama disease almost wiped out commercial {{{{Gros Michel
      }}}} banana production.

*    {{{{Panama disease}}}} - It appears in the first clause of the first sentence
     and is therefore ineligible. This choice is **poor**.
*    {{{{disease}}}} - This entity does not require recalling a world fact; it can be
      predicted from the context clues "Fusarium wilt" and "infects." This choice is
     **poor**.
*    {{{{Musa spp.}}}} - While recalling the scientific name for banana plants is a
     good test of knowledge, the use of parentheses makes the answer less certain.
     The context could also imply other related species. This choice is **good**.
*    {{{{banana plants}}}} - The context "a plant disease that infects..." combined
     with the subject "Panama disease" creates a very strong expectation for this
     specific plant, a well-known fact. This choice is **excellent**.
*    {{{{Fusarium oxysporum f. sp. cubense}}}} - The preceding phrase "...caused by
     the fungus..." requires a specific, factual answer. Knowing the fungus that
     causes Panama disease is a clear test of world knowledge. This choice is **
     excellent**.
*    {{{{1950s}}}} - The phrase "During the..." is too vague. It could be completed
     with many different time periods, making the answer unpredictable. This choice
     is **poor**.
*    {{{{Gros Michel}}}} - The context "...wiped out commercial..." strongly points
     to a specific type of banana. However, it doesn't uniquely identify the variety
     without further knowledge, so it's not perfectly constrained. This choice is **
     good**.

### **Text 2**
Liam Tarquin Broady (BROH-dee; born {{{{4 January 1994}}}}) is a {{{{British}}}}
     professional {{{{tennis player}}}} who competes mainly on the {{{{ATP Challenger
      Tour}}}}. He reached a career high ranking of world No. {{{{93}}}} on 25
```

```
    September 2023.

*   {{{{4 January 1994}}}} - The entity appears in the first clause of the first
    sentence, making it ineligible per the instructions. This choice is **poor**.
*   {{{{British}}}} - The left-context "...is a..." is too generic and does not
    sufficiently narrow down the possibilities. This choice is **poor**.
*   {{{{tennis player}}}} - The context "...is a British professional..." strongly
    implies a profession. Knowing who Liam Broady is leads to a single, specific
    prediction. This choice is **excellent**.
*   {{{{ATP Challenger Tour}}}} - The context "who competes mainly on the..." points
    to a tournament or tour. However, the word "mainly" introduces uncertainty,
    meaning the answer is not guaranteed to be unique. This choice is **good**.
*   {{{{93}}}} - The preceding phrase "...career high ranking of world No..."
    creates a very specific slot for a number. Recalling a player's exact ranking is
    a great test of factual knowledge. This choice is **excellent**.

### **Text 3**
Happiness for Beginners is a {{{{2023}}}} American {{{{romantic comedy}}}} film
    starring {{{{Ellie Kemper}}}} and {{{{Luke Grimes}}}}, an adaptation of the {{{{
    Katherine Center}}}} 2015 novel of the same name. The novel was adapted for the
    screen and directed by {{{{Vicky Wight}}}}.

*   {{{{2023}}}} - This entity appears in the first clause, making it ineligible.
    Additionally, the context "...is a..." is too general to uniquely predict a year
    . This choice is **poor**.
*   {{{{Ellie Kemper}}}} - The phrase "film starring" clearly points to an actor.
    However, since a film can have multiple stars, the specific actor named first is
     not uniquely predictable. This choice is **good**.
*   {{{{Luke Grimes}}}} - The context "...starring Ellie Kemper and..." creates a
    strong expectation for a co-star, a common structure in this genre. This
    requires specific knowledge about the film. This choice is **excellent**.
*   {{{{Katherine Center}}}} - The crucial context clue, "novel," appears *after*
    the entity. Therefore, the answer cannot be predicted from the left-context
    alone. This choice is **poor**.
*   {{{{Vicky Wight}}}} - The left-context "...adapted for the screen and directed
    by..." clearly and uniquely signals that a director's name is required. This
    choice is **excellent**.

----

### **Required Output Format**

Return a Python list of selected **OBJECT** entities, sorted from most suitable to
    least. Each entity must be written in the format `{{{{entity_name}}}}`, exactly
    as it appears in the original text.
Example:
["{{{{entity1_name}}}}", "{{{{entity2_name}}}}", "{{{{entity3_name}}}}"]

**Instruction End**

**Please select up to three OBJECT entities from the following annotated paragraph,
    and return them in a list sorted by the most suitable OBJECT to the least:**
{tagged_paragraph}
```

**Box E.4: Prompt for Verifying Object Entity Suitability – with Gold Target**

```
You will be provided with a *left-context* (a non-completed sentence or paragraph)
    and a proposed **OBJECT** entity that completes it.
Your task is to verify whether this OBJECT entity is a **suitable factual completion
    **, meaning:
```

1. The left-context provides enough semantic cues to **strongly constrain** what type of entity should come next (e.g., a date, person, award, location, number, title, etc.).
2. There is a **unique and specific** gold entity completion which is the OBJECT entity:

   * There are **no multiple plausible alternative completions** that refer to different entities.
   * If a language model predicts a different entity, it would **clearly contradict** the information conveyed by the *left-context* and the OBJECT.
   * The OBJECT entity is not underdetermined, vague, or overly general.

----

### **Guidelines for Determining Suitability**

A suitable OBJECT entity must meet **all** of the following:

* The left-context clearly constrains the **type** of entity expected next (e.g., a food, a date, a chemical stabilizer, a person's name, a specific event).
* The OBJECT entity fits seamlessly into the left-context to form a coherent, factual statement.
* The left-context likely **narrows the answer to a unique entity** and any alternative completion would conflict with the facts conveyed so far.

The OBJECT is **not suitable** if:

* The left-context is too vague or underspecified to narrow the completion to a single, specific entity.
* There are multiple plausible completions of the same type (e.g., many possible awards, people, or ingredients).
* The constraints on the OBJECT rely on **right-context** (i.e., text that appears after the entity).
* The OBJECT appears to fit syntactically or thematically, but the left-context provides **no strong evidence** that it is the only valid answer.

*Clarification on Uniqueness:* uniqueness refers to the **semantic identity** of the OBJECT entity and not to surface forms (the string used to represent an entity). The OBJECT is still considered unique even if it has other forms that:

* Differ in capitalization, punctuation, or grammatical form (e.g., pluralization, articles).
* Omit non-essential modifiers (e.g., "Pulitzer Prize for Fiction" vs. "2003 Pulitzer Prize for Fiction").
* Use a common variant or alias (e.g., "United States" vs. "U.S.").

However, if multiple **distinct** entities (e.g., different stabilizers, places, awards, or people) could reasonably complete the left-context, then the OBJECT is **not suitable**.

----

### **NOT SUITABLE Examples**

**Left-context:**
Garfield is portrayed as a lazy, fat, cynical, and self-absorbed orange tabby Persian cat. He is noted for his love of lasagna and *
**OBJECT:** pizza
**Explanation:** The left-context does not constrain the answer to be a food. Other plausible completions include "naps", "Mondays", or "sleep". The OBJECT is not uniquely inferable.

**Left-context:**
Glass noodles, sometimes called cellophane noodles, are a type of transparent noodle
    made from starch (such as mung bean starch, potato starch, sweet potato starch,
    tapioca, or canna starch) and water. They originated in China. A stabilizer
    such as chitosan or *
**OBJECT:** alum
**Explanation:** The left-context signals that a stabilizer is expected, but it does
    not sufficiently constrain which one. Multiple valid completions exist, so the
    OBJECT is not unique.

**Left-context:**
Rayagada railway station is located at the heart of the city and now it has the
    longest platform in Odisha (Platform No :- 3 measuring 910m), the 6th in the
    country and the
**OBJECT:** 7th
**Explanation:** The left-context does not clarify what the "7th" refers to. The
    comparison class is underspecified (7th in what? globally? regionally?), so the
    answer is ambiguous.

----

### **SUITABLE Examples**

**Left-context:**
Adrian Darnell Griffin Sr. (born July 4, 1974) is an American professional
    basketball coach and former player who most recently served as the head coach
    for the Milwaukee Bucks of the National Basketball Association (NBA). He played
    in the NBA as a shooting guard and small forward from 1999 to 2008. Griffin grew
    up in *
**OBJECT:** Wichita
**Explanation:** The phrase "grew up in" combined with known biographical details
    provides strong context for a specific city. Other completions would clearly
    contradict the known fact.

**Left-context:**
Andrea Barrett (born November 16, 1954) is an American novelist and short story
    writer. Her collection Ship Fever won the 1996 U.S. National Book Award for
    Fiction, and she received a MacArthur Fellowship in 2001. Her book Servants of
    the Map was a finalist for the 2003 *
**OBJECT:** Pulitzer Prize for Fiction
**Explanation:** While there are many awards, the combination of book title, year,
    and prior accolades strongly constrains the answer. The context likely narrows
    the answer to a unique award.

**Left-context:**
Bering Air Flight 445 was a scheduled domestic flight from Unalakleet Airport to
    Nome Airport which crashed on February 6, 2025 while flying over the Norton
    Sound. The flight was aboard a *
**OBJECT:** Cessna 208B Grand Caravan EX
**Explanation:** The left-context refers to a known historical event. There is only
    one aircraft associated with this flight, making the answer uniquely
    identifiable.

**Left-context:**
George Mark Collie (born January 18, 1956) is an American country music singer,
    songwriter, musician, actor, record producer, and fundraiser for Type 1 diabetes
    study. He has won awards and acclaim for his music, his acting, and his
    philanthropy. His singing career has included five major-label albums: four for

```
    MCA Nashville and one for Giant Records. Sixteen of his singles have charted on
      *
**OBJECT:** Hot Country Songs
**Explanation:** The specific number of charting singles and genre context constrain
     the answer to a unique Billboard chart. Alternate completions would contradict
    the left-context.

----

### **Required Output Format**

Respond only with "SUITABLE" or "NO".

**Instruction End**

**Please process the following left-context and the OBJECT entity:**

**Left-context:** {left_context}
**OBJECT:** {object_entity}
```

```
You will be provided with a *left-context* (a non-completed sentence or paragraph).
    Your task is to verify whether this left-context is a **suitable factual
    completion task**, meaning:

1. The left-context provides enough semantic cues to **strongly constrain** what
    type of entity should come next (e.g., a date, person, award, location, number,
    title, etc.).
2. There is a **unique and specific** gold entity completion:

  * There are **no multiple plausible alternative completions** that refer to
     different entities.
  * If a language model predicts a different entity than the gold entity, it would
     **clearly contradict** the information conveyed by the *left-context* and the
     gold entity.

----

### **Guidelines for Determining Suitability**

A suitable left-context must meet **all** of the following:

* The left-context clearly constrains the **type** of entity expected next (e.g., a
    food, a date, a chemical stabilizer, a person's name, a specific event).
* The left-context likely **narrows the answer to a unique entity** and any
    alternative completion would conflict with the facts conveyed so far.

The left-context is **not suitable** if:

* The left-context is too vague or underspecified to narrow the completion to a
    single, specific entity.
* There are multiple plausible completions of the same type (e.g., many possible
    awards, people, or ingredients).

*Clarification on Uniqueness:* uniqueness refers to the **semantic identity** of the
     completion and not to surface forms (the string used to represent an entity).
     The completion is still considered unique even if it has other forms that:

* Differ in capitalization, punctuation, or grammatical form (e.g., pluralization,
```

    articles).
* Omit non-essential modifiers (e.g., "Pulitzer Prize for Fiction" vs. "2003
    Pulitzer Prize for Fiction").
* Use a common variant or alias (e.g., "United States" vs. "U.S.").

However, if multiple **distinct** entities (e.g., different stabilizers, places,
    awards, or people) could reasonably complete the left-context, then it is **not
    suitable**.

----

### **NOT SUITABLE Examples**

**Left-context:**
Garfield is portrayed as a lazy, fat, cynical, and self-absorbed orange tabby
    Persian cat. He is noted for his love of lasagna and *
**Explanation:** The left-context does not constrain the answer to a single entity
    type. There are many plausible completions such as "pizza", "naps", "Mondays",
    or "sleep".

**Left-context:**
Glass noodles, sometimes called cellophane noodles, are a type of transparent noodle
     made from starch (such as mung bean starch, potato starch, sweet potato starch,
     tapioca, or canna starch) and water. They originated in China. A stabilizer
    such as chitosan or *
**Explanation:** The left-context signals that a stabilizer is expected, but it does
     not sufficiently constrain which one. Multiple valid completions exist, so the
    completion is not unique.

**Left-context:**
Rayagada railway station is located at the heart of the city and now it has the
    longest platform in Odisha (Platform No :- 3 measuring 910m), the 6th in the
    country and the *
**Explanation:** The left-context does not clarify what should come after "...and
    the". Even if a number is likely ("...and the Xth..."), the comparison class is
    underspecified (Xth in what? globally? regionally?), so the answer is ambiguous.

----

### **SUITABLE Examples**

**Left-context:**
Adrian Darnell Griffin Sr. (born July 4, 1974) is an American professional
    basketball coach and former player who most recently served as the head coach
    for the Milwaukee Bucks of the National Basketball Association (NBA). He played
    in the NBA as a shooting guard and small forward from 1999 to 2008. Griffin grew
     up in *
**Explanation:** The phrase "grew up in" combined with known biographical details
    provides strong context for a specific city. Other completions would clearly
    contradict the known fact.

**Left-context:**
Andrea Barrett (born November 16, 1954) is an American novelist and short story
    writer. Her collection Ship Fever won the 1996 U.S. National Book Award for
    Fiction, and she received a MacArthur Fellowship in 2001. Her book Servants of
    the Map was a finalist for the 2003 *
**Explanation:** While there are many awards, the combination of book title, prior
    accolades and the year, strongly constrains the answer. The left-context likely
    narrows the answer to a unique award.

```
**Left-context:**
Bering Air Flight 445 was a scheduled domestic flight from Unalakleet Airport to
    Nome Airport which crashed on February 6, 2025 while flying over the Norton
    Sound. The flight was aboard a *
**Explanation:** The left-context refers to a known historical event. There is only
    one aircraft associated with this flight, making the answer unique.

**Left-context:**
George Mark Collie (born January 18, 1956) is an American country music singer,
    songwriter, musician, actor, record producer, and fundraiser for Type 1 diabetes
     study. He has won awards and acclaim for his music, his acting, and his
    philanthropy. His singing career has included five major-label albums: four for
    MCA Nashville and one for Giant Records. Sixteen of his singles have charted on
    *
**Explanation:** The specific number of charting singles and genre context constrain
     the answer to a unique Billboard chart. Alternate completions would contradict
    the left-context.

----

### **Required Output Format**

Respond only with "SUITABLE" or "NO".

**Instruction End**

**Please process the following left-context:**

**Left-context:** {left_context}
```

## Box E.6: Prompt for Verifying Completion Difficulty

```
You will be provided with a *left-context* (a non-completed sentence or paragraph)
    and a proposed **OBJECT** entity that completes it.
Your task is to verify whether this OBJECT entity is a **suitable factual completion
    **.

The OBJECT is **suitable** if:

1. The completion is challenging and requires specific factual knowledge.
2. It is not trivially easy to guess or logically infer from the left-context alone.
3. The left-context does not contain the answer or any strong hints that directly
    disclose the OBJECT entity.
4. The OBJECT entity is not time-sensitive and will not change over time.

The OBJECT is **not suitable** if:

1. The OBJECT entity can be predicted using general common sense or an educated
    guess.
2. The OBJECT entity is trivially inferable using general common knowledge.
3. The OBJECT entity, as a completion of the left-context, is subject to change over
     time (it is not static).

----

### **NOT SUITABLE Examples**

**Left-context:**
```

```
The Strait of Hormuz (Persian: ... Tangeh-ye Hormoz) is situated between *
**OBJECT:** Iran
**Explanation:** Since the left-context provides the name in Persian script, the
    answer "Iran" can be easily guessed. Therefore, the completion is not
    challenging and is considered not suitable.

**Left-context:**
The Nobel Prize in Literature is a Swedish literature prize that is awarded annually
     to an author from any country who has, in the words of Alfred Nobel, "in the
    field of literature, produced the most outstanding work in an idealistic
    direction". It is one of the five *
**OBJECT:** Nobel Prizes
**Explanation:** The completion is trivial because the subject ("The Nobel Prize in
    Literature") already contains the target OBJECT entity.

**Left-context:**
Income in the United States is measured by the various federal agencies including
    the Internal Revenue Service, Bureau of Labor Statistics, US Department of
    Commerce, and the US Census Bureau. Overall, including all households/
    individuals regardless of employment status, the median household income was *
**OBJECT:** $67,521
**Explanation:** The OBJECT entity is time-sensitive because the specific year is
    not specified in the context. Since median income changes over time, this
    completion is not suitable.

----

### **SUITABLE Examples**

**Left-context:**
Redbus India Private Limited, doing business as redBus, is an Indian multinational
    online bus-ticketing platform. In 2018, the company achieved a GMV of *
**OBJECT:** 50 billion Indian Rupee
**Explanation:** The OBJECT entity is not time-sensitive because the context
    restricts the fact to a specific year (2018), making the value static.

**Left-context:**
"Thriller" is a song by the American singer Michael Jackson. It includes a spoken-
    word sequence performed by the horror actor *
**OBJECT:** Vincent Price
**Explanation:** This is a challenging completion that requires the recall of
    specific factual knowledge; therefore, it is suitable.

**Left-context:**
The Panathenaic Stadium is a multi-purpose stadium in Athens, Greece. After the rise
     of Christianity in the 4th century it was largely abandoned. After being
    refurbished, it hosted the opening and closing ceremonies of the *
**OBJECT:** first modern Olympics
**Explanation:** Although the stadium's location in Greece hints that the OBJECT is
    related to the Olympic Games, identifying the specific OBJECT requires knowing
    it hosted the *first modern* Olympics. This makes the completion non-trivial and
     suitable.

----

### **Required Output Format**

Respond only with "SUITABLE" or "NO".
```

```
**Instruction End**

**Please process the following left-context:**

**Left-context:** {left_context}
**OBJECT:** {object_entity}
```

## E.2. Question Generation Prompts

---

**Box E.7: Prompt for Generating the *Direct* Task**

```
You will be provided with a *left-context* (an incomplete sentence or paragraph) and
    an *OBJECT* entity that completes it. Based only on the left-context, your task
    is to design a unambiguous question for which the only correct and specific
    answer is the OBJECT. The question should only use information necessary for
    disambiguation and to ensure the answer's specificity, using the exact phrasing
    from the left-context.

**Requirements:**

*   Only the OBJECT should be the correct answer.
*   The question must be self-contained: it should not use pronouns or references
    that require outside context.
*   Use only the information available in the left-context.
*   The question must be unambiguous: its phrasing should be clear and avoid
    multiple interpretations.
*   The question must be specific and precise: it should specify the level of detail
     and type of answer required (e.g., "What year...", "What is the name of the
    condition...", "Which city...").
*   The question must be minimal: include only the essential details from the left-
    context needed to make the question unambiguous and ensure the OBJECT is the
    only possible answer.
*   Use the same terms, phrases, and words that appear in the left-context. Do not
    rephrase or use synonyms, unless essential for grammaticality or clarity.

----

### **Required Output Format**

Return only the question without any additional text.

----

### **Examples**

Here are a few examples, including explanations for how the questions were
    formulated:

1.
**Left-context:** Erika Nordby (born February 2000), also known as Baby Erika,
    Miracle Baby and Canada's Miracle Child, is a Canadian originally from Edmonton,
     Alberta known for having been revived after spending two hours without a
    heartbeat due to
**OBJECT:** hypothermia
**Output:** What was the condition that caused Erika Nordby to spend two hours
    without a heartbeat before she was revived?
*Explanation:*
*   We ask "What was the condition..." because hypothermia is a medical condition.
    This phrasing demands a specific type of answer.
*   We include "Erika Nordby" to identify the person the question is about, making
```

      it self-contained.
*    We include "two hours without a heartbeat before she was revived" because these
     are the key, unique details from the context that pinpoint the specific cause
     being asked about.
*    We do not include the birth date or the nicknames to make the question minimal.
     The combination of the name and the incident of 'two hours without a heartbeat'
     is enough for disambiguation.
*    All terms are taken directly from the left-context.

2.
**Left-context:** When I Look in Your Eyes is the fifth studio album by singer Diana
     Krall, released on
**OBJECT:** June 8, 1999
**Output:** What was the release date of When I Look in Your Eyes by Diana Krall?
*Explanation:*
*    The phrasing "What was the release date..." is precise and directly corresponds
     to the left-context phrase "released on."
*    The album title could be ambiguous on its own, so we add "by Diana Krall" for
     disambiguation.
*    To keep the question minimal, we do not include the fact that it was her fifth
     studio album. The combination of the album title and the artist's name is
     sufficient for disambiguation.

3.
**Left-context:** Celebrimbor is a fictional character in J. R. R. Tolkien's
     legendarium. In Tolkien's stories, Celebrimbor was an elven-smith who was
     manipulated into forging the Rings of Power by the Dark Lord Sauron, in a fair
     disguise and named
**OBJECT:** Annatar
**Output:** What name did Sauron use in his fair disguise to manipulate Celebrimbor?
*Explanation:*
*    "What name..." sets the clear expectation for the type of answer required.
*    We must include "in his fair disguise" and "to manipulate Celebrimbor" because
     these are the specific circumstances under which Sauron used the name Annatar.
     Simply asking "What name did Sauron use?" would be too broad.
*    Further information is not required to make the question unique and unambiguous.
      For example, mentioning Celebrimbor and Sauron is enough to understand that
     they are characters from Tolkien's legendarium.
*    All terms are taken directly from the left-context.

4.
**Left-context:** The horse-collar tackle is a gridiron football maneuver in which a
      defender tackles another player by grabbing the back collar or the back-inside
     of an opponent's shoulder pads and pulling the ball carrier directly downward
     violently in order to pull his feet from underneath him. The technique is most
     closely associated with Pro Bowl safety Roy Williams. After being blamed for a
     series of major injuries in the 2004 season , the horse-collar tackle was banned
      from the NFL during the 2005 off-season. The rule forbidding it is often
     referred to in the press as
**OBJECT:** The Roy Williams Rule
**Output:** What is the name the press often uses to refer to the rule forbidding
     the horse-collar tackle from the NFL?
*Explanation:*
*    To make the question self-contained, we must describe the rule being referenced:
      "the rule forbidding the horse-collar tackle from the *NFL*".
*    The left-context specifies the rule is "referred to in the press as..." This
     detail is critical. Therefore, the question includes "What is the name the press
      often uses to refer to..." to remain faithful to the source text.
*    The years, the description of the maneuver, and the injuries are unnecessary for
      understanding the question and can be excluded for minimalism.
*    All terms are taken directly from the left-context.

**Instruction End**

```
**Please process the following left-context and OBJECT:**

**Left-context:** {left_context}
**OBJECT:** {object_entity}
```

## Box E.8: Prompt for Refining Specification

```
You will be provided with a question, a gold answer, and its corresponding entity
    type. Your task is to evaluate the question's specificity and precision, with
    respect to the gold answer, and revise the question if necessary.
A specific and precise question should clearly define the level of detail required
    in the answer.
Your evaluation process should consider the following stages: **Verification**, **
    Revision** and **Rejection**.

### **Verification**

Determine if the question is specific and precise. Consider the following when
    making your decision:

*   Would a person with the necessary knowledge, who is familiar with the gold
    answer, be led to provide that specific gold answer to the question?
*   Does the question specify the required level of detail and type for the answer?
*   Is it easy to infer the required entity type of the answer from the question?
*   If relevant, does the question specify the required granularity of the answer?
    *   For example: "What year...", "What year and month...", "What date...", "What
         is the name of the condition...", "Which city...", "Which country...", "
        Which province...", "What is the scientific name...".

If the question is specific and precise, you can stop here and return it exactly as
    it is. Note that most of your inputs will be considered specific and precise.
If the question is not specific or precise, you should try to revise it.

### **Revision**

A question should be revised if it is not specific but can be fixed with **minimal
    changes**.
A valid revision must result in a question that perfectly and uniquely targets the
    gold answer without leaking non-trivial information.
Common revision techniques involve adding a short, clarifying phrase or changing a
    single word, for example:

*   **Specify the answer's entity type.** This is often done by changing the
    question word to be more precise, e.g., `Who` -> `Which company`.
*   **Specify the answer's required granularity or format.** This is often done by
    clarifying the necessary level of detail. e.g., `When` -> `In what month and
    year`, `Where` -> `In which city and country`,
 `What name` -> `What is the scientific name`.
*   **Specify the units or precision for numerical data.** This removes ambiguity
    from the expected answer. e.g., Adding `in Celsius`, `in US dollars`, or `in
    millions`.
*   **Add a qualifier to narrow the scope.** When a question is ambiguous and could
    have multiple correct answers, add a qualifier to focus the question on the
    single intended answer.  Note that you should not disclose non-trivial
    information. e.g., Revise `What ecosystem...`? to `What ecosystem **of mountains
    **...`? to distinguish from other valid answers (e.g., a forest).

### **Rejection**
```

If a question cannot be fixed with a valid minimal revision, it must be rejected.
A question is considered unfixable and should be rejected in the following cases:
*   The revision would leak information: The only way to fix the question would be
    to add a non-trivial clue that is part of the gold answer itself, making the
    question significantly easier.
*   The revision would require a major change: The question is too vague or
    fundamentally mismatched with the answer, and fixing it would require a complete
     rewrite rather than a simple, minimal clarification.

----

### **Required Output Format**

You must return *only* a string, which can be the original question, the revised
    question, or "REJECT".
If the question is specific and precise as it is, return the question. If the
    question needs revision, revise it and return the revised question. If you
    conclude that the question should be rejected, return "REJECT".
Remember, return only the original question, the revised question, or "REJECT",
    without any additional text.

----

### **Examples**

1.
**Question:** What is the condition that caused Erika Nordby to spend two hours
    without a heartbeat before she was revived?
**Gold answer:** hypothermia
**Explanation:** The question is specific and precise.
**Output:** What is the condition that caused Erika Nordby to spend two hours
    without a heartbeat before she was revived?

2.
**Question:** What name did Sauron use in his fair disguise to manipulate
    Celebrimbor?
**Gold answer:** Annatar
**Explanation:** The question is specific and precise.
**Output:** What name did Sauron use in his fair disguise to manipulate Celebrimbor?

3.
**Question:** Who produced the 2020 German romantic comedy film Isi & Ossi?
**Gold answer:** Netflix
**Explanation:** The question word "Who" is ambiguous, as it typically refers to a
    person. The revision to "Which company" precisely targets the entity type of the
     answer.
**Output:** Which company produced the 2020 German romantic comedy film Isi & Ossi?

4.
**Question:** What biodiverse ecosystem surrounds Lake Burton?
**Gold answer:** Blue Ridge Mountains
**Explanation:** The term "ecosystem" is too broad, allowing for multiple correct
    answers (e.g., a forest). The qualifier "of mountains" narrows the scope to the
    intended answer without leaking information.
**Output:** What biodiverse ecosystem of mountains surrounds Lake Burton?

**Instruction End**

**Please process the following question, gold answer, and entity type:**

**Question:** {question}
**Gold answer:** {gold_answer}
**Entity type:** {entity_type}

### Box E.9: Prompt for Refining Minimalism

You will be provided with a question and its gold answer. Your task is to evaluate
    the question's minimalism and revise it by removing any non-essential
    information.
The question must be unambiguous and lead to a single, unique answer, while avoiding
     non-essential details that are not required for the disambiguation of the core
    entities in the question or the answer.
The primary goal is not simply to make the question shorter, but to remove
    information that acts as an unnecessary hint (which helps answer the question)
    but does not aid in disambiguation.
Your evaluation process should consider the following stages: **Verification** and
    **Revision**.

### **Verification**

Determine if the question is already minimal. A question is considered minimal if it
     uniquely leads to the provided gold answer without containing unnecessary
    details. Consider the following:

*   Are all entities in the question (people, works, events) identified with just
    enough detail to be unambiguous for the purpose of the question? For example,
    adding "by Diana Krall" is essential if the song title "When I Look in Your Eyes
    " might not be unique on its own, as there may be another song with the same
    title.
*   Does the question contain any superfluous words or phrases (e.g., adjectives,
    specific years, genres) that could be removed without affecting the question's
    ability to point uniquely to the required type of gold answer?
*   Does the question avoid providing unnecessary hints or clues? A hint is a piece
    of information that makes the answer easier but is not needed for the
    identification of the entities or the intended answer from other possibilities.
*   Note: Most questions are already minimal. Minimalism is not about shorter
    questions, but about not including details that provide a hint without being
    essential for the question's disambiguation.

If the question is already minimal, you can stop here and return it exactly as it is
    . Note that many questions, even if long, will be considered minimal if all
    parts are essential.

### **Revision**

A question should be revised if it contains non-essential information that can be
    removed while preserving the question's disambiguity, entity identification, and
     answer specification.
A valid revision consists **only of deleting** words or phrases. You must not add,
    rephrase, or reorder any part of the question. The original wording of the
    remaining parts must be preserved.
You can perform a minimal change to fix the grammar.

After deleting a word or phrase, you must confirm that the revised question still:
1.  Makes sense grammatically and logically.
2.  Uniquely and unambiguously points to the provided gold answer.

Common details to remove for minimalism include:

*   **Redundant Specifiers:** Details like a specific year or genre, *if* another,
    more unique identifier in the question already makes them unnecessary.
    *   Example: "Which **1958 crime** drama by Orson Welles featured..." might be
        reducible if another part of the question is sufficiently unique.
*   **Superfluous Descriptors:** Words that describe an entity but are not needed to
     distinguish it from other entities.
    *   Example: Removing "**country and gospel**" from "...quartet the Oak Ridge
        Boys," as the band's name is a unique identifier.

*   **Leaked Clues:** Information that provides a separate, related fact not
    essential to identifying the answer.
    *   Example: Removing "**defending champion**" from "...team contested the 1966
        Stanley Cup Finals," as it reveals the result of the previous year's final.

**Important:**
*   Do not remove essential phrases that describe the core subject of the question (
    e.g., "...based on the life of a Military Commander...").
*   Do not shorten or remove official or commonly used names (e.g., keep "Ebola
    virus disease") and do not remove abbreviations.
*   Keep in mind that MOST questions are already minimal. Your goal is not simply to
    shorten the question, but to remove any information that is unnecessary for
    disambiguation and acts as an unneeded hint.
----

### **Required Output Format**

You must return *only* a string, which can be the original question or the revised
    question.
If the question is minimal as it is, return the question. If the question needs
    revision, revise it by deleting the non-essential parts and returning the
    revised question.
Remember to return only the question string, without any additional text.

----

### **Examples**

1.
**Question:** What name did Sauron use in his fair disguise to manipulate
    Celebrimbor?
**Gold answer:** Annatar
**Explanation:** The question is already minimal. "in his fair disguise" and "to
    manipulate Celebrimbor" are both essential context to specify which of Sauron's
    names is being asked for. Removing either phrase would make the question
    ambiguous.
**Output:** What name did Sauron use in his fair disguise to manipulate Celebrimbor?

2.
**Question:** The film "Sarsenapati Hambirrao" is based on the life of a Military
    Commander of which empire?
**Gold answer:** Maratha Empire
**Explanation:** The question is already minimal. The phrase "the life of a Military
    Commander" is the core subject of the question and clarifies which empire is
    relevant (the one the commander served, the film might focus on more than a
    single empire). This is not an unnecessary hint.
**Output:** The film "Sarsenapati Hambirrao" is based on the life of a Military
    Commander of which empire?

3.
**Question:** What vocal part does Richard Anthony Sterban sing in the country and
    gospel quartet the Oak Ridge Boys?
**Gold answer:** Bass
**Explanation:** The name of the quartet, "the Oak Ridge Boys," is a sufficient
    identifier. The genres "country and gospel" are superfluous descriptors that are
    not needed for disambiguation. They can be removed.
**Output:** What vocal part does Richard Anthony Sterban sing in the quartet the Oak
    Ridge Boys?

4.
**Question:** Which defending champion team contested the 1966 Stanley Cup Finals
    against the Detroit Red Wings?
**Gold answer:** Montreal Canadiens

```
**Explanation:** The phrase "defending champion" is a leaked clue. It provides an
    additional fact (that the team won the 1965 finals) that is not necessary to
    uniquely identify the team that played the Red Wings in the 1966 finals. This
    non-essential information should be removed.
**Output:** Which team contested the 1966 Stanley Cup Finals against the Detroit Red
    Wings?

**Instruction End**

**Please process the following question and gold answer:**

**Question:** {question}
**Gold answer:** {gold_answer}
```

**Box E.10: Prompt for Filtering with Google Search Grounding**

```
You will be provided with a question and a corresponding answer written by an expert
    . Your task is to evaluate if the question is **suitable** based on the provided
     expert answer.
A **suitable question** must be unambiguous, specific, and have a single, unique
    answer. Your evaluation should be based *solely* on the content of the expert's
    answer.
Your final output must be only one word: **SUITABLE** or **NOT**.

----

### **Evaluation Process**

Carefully analyze the question and the expert's answer to determine the question's
    suitability.

**Definition of a Suitable Question**

A question is considered suitable if it meets all the following criteria:
1.  **Unambiguous:** All entities in the question can be easily and uniquely
    identified. There is no need to ask for clarification to understand the exact
    intent of the question.
2.  **Specific and precise:** The question asks for a specific and precise piece of
    information, making it easy to understand the expected type and format of the
    answer.
3.  **Has a single and unique answer:** The question is structured in a way that
    there is only one possible correct answer.

**Determining Suitability Based on the Expert's Answer**

You must determine the question is **NOT** suitable if the expert's answer exhibits
    any of the following characteristics:

*   **The answer asks for clarification:** The expert's response includes questions
    back to the user to resolve ambiguity.
    *   *Example indicators:* "Which 'Scarlet' are you referring to?", "Could you
        please specify the year?", "It's not clear what you mean by 'official
        language'..."
*   **The answer states there are multiple possible answers:** The expert explicitly
     mentions that the question has more than one valid answer.
    *   *Example indicators:* "There are several former presidents of the USA...", "
        Multiple actors have played this role...", "Several cities match this
        description..."
*   **The answer provides different answers for different interpretations:** The
    expert identifies ambiguity and provides separate answers for each possible
```

```
        interpretation of the question.
    *   *Example indicators:* "If you are asking about the actress Scarlet Johansson
        , the answer is... However, if you mean the literary character Scarlett O'
        Hara, the answer is...", "For the 2008 movie, the director was... For the
        2021 remake, the director was..."

A question is considered **SUITABLE** only if the expert's answer provides a direct,
    singular, and factual response without expressing any confusion, highlighting
    ambiguity, or indicating the existence of multiple valid answers.

----

### **Required Output Format**

You must return *only* a string, either **SUITABLE** or **NOT**, without any
    additional text or explanation.

**Instruction End**

**Please process the following question and the expert's answer:**

**Question:** {question}
**Answer:** {search_pred}
```

---

**Box E.11: Prompt for Generating the *Reverse* Task**

```
You will be provided with a context, a question, and an answer. The answer is an
    OBJECT entity that also completes the last sentence of the provided context.
Your task is to generate a *reverse question*. In this new question, the SUBJECT
    entity of the original question will be the answer. The OBJECT entity from the
    original question will become the new subject of the reverse question.

The process involves three main stages: **Identification**, **Generation**, and **
    Evaluation**.

### **Stage 1: Identify the SUBJECT Entity**

Analyze the original question to identify its SUBJECT entity. The SUBJECT is the
    main focus of the question, and the question asks for information related to
    this entity.
For example, in the question "What is the capital of France?" the SUBJECT is "France
    ".

### **Stage 2: Generate a Reverse Question**

Create a new question where the original SUBJECT entity becomes the answer, and the
    original OBJECT entity becomes the new subject.
The reverse question must be an unambiguous question with a single, unique, and
    specific answer: the SUBJECT. It must be based on the given question and act as
    its reverse.
The reverse question should follow these requirements:

*   **Self-contained:** It does not use pronouns or references that require outside
    context.
*   **Unambiguous and Specific:** Its phrasing is clear, avoids multiple
    interpretations, and specifies the level of detail required in the answer (e.g.,
     "What year...", "What is the name of the condition...").
*   **Answer Uniqueness:** It must be phrased to have a *single* valid and correct
    answer which is the SUBJECT.
*   **Minimal:** It includes only the essential details from the context and the
```

original question needed to make the reverse question unambiguous and ensure the SUBJECT is the only valid answer.
*   **Verbatim:** It uses the same terms, phrases, and words that appear in the context and the given question, without rephrasing.
*   **Challenging:** The reverse question should be challenging and non-trivial to answer. The difficulty level of the original question and the reverse question should be roughly the same.

**IMPORTANT:** Although not needed in most cases, you may sometimes add information from the *context* to the reverse question, provided it is required to make the question unambiguous or to ensure the SUBJECT is the only valid answer. However, this information must be **minimal** and must **not shift the focus** of the question. It should not introduce a hint or clue that allows solving the reverse question without knowledge of the core relationship between the original OBJECT and SUBJECT. The added detail should not become a shortcut to the answer and should not make the question easier without helping in disambiguation.

### **Stage 3: Determine Suitability and Rejection**

Finally, you must assess the reverse question you generated. If it fails to meet the requirements above, you must reject it.
A question should be rejected if it is impossible to create a reverse version that is unambiguous and has a single valid answer.
It should also be rejected if any attempt to make it meet the requirements shifts the focus of the question or makes it trivial or easier.

----

### **Required Output Format**

Your output must be in one of two formats.

If you generate a successful reverse question, the response must be structured using the following tags: `<reverse>` and `<subject>`. Do not include any other text, explanations, or formatting.

`<reverse>` The reverse question `</reverse>`
`<subject>` The subject entity, which is the answer to the reverse question `</subject>`

If a suitable reverse question cannot be generated, you must return only the string: `REJECT`

----

### **Examples**

Here are several examples demonstrating the task. Pay close attention to the explanations.

1.
**Context:** Cody Blake Dennison (born March 4, 1990) is an American professional stock car racing driver and YouTube personality who competes part-time in the NASCAR Craftsman Truck Series, driving the Nos. 2/22 Ford F-150
**Question:** What is the model of the Nos. 2/22 truck that Cody Blake Dennison drives in the NASCAR Craftsman Truck Series?
**OBJECT:** Ford F-150
**Output:**
<reverse> Who drives the Nos. 2/22 Ford F-150 in the NASCAR Craftsman Truck Series? </reverse>
<subject> Cody Blake Dennison </subject>
**Explanation:** We use the same words and terms from the question.

```
2.
**Context:** Summer Side of Life is Canadian singer-songwriter Gordon Lightfoot's
    sixth studio album. It was released in 1971
**Question:** In what year was Gordon Lightfoot's album, Summer Side of Life,
    released?
**OBJECT:** 1971
**Output:**
<reverse> What is the name of Gordon Lightfoot's album that was released in 1971? </
    reverse>
<subject> Summer Side of Life </subject>
**Explanation:** It is likely that the singer released only a single album in 1971,
    making the answer single and unique.

3.
**Context:** Diabolik: Who Are You? (Italian: Diabolik - Chi sei?) is a 2023 Italian
     crime action film directed by the Manetti Bros. and based on the 1968 Diabolik
    comic strip Diabolik, chi sei? by Angela and Luciana Giussani. It premiered at
    the 18th Rome Film Festival
**Question:** At which event did the film 'Diabolik: Who Are You?' premiere?
**OBJECT:** Rome Film Festival
**Output:**
<reverse> What film directed by the Manetti Bros. premiered at the 18th Rome Film
    Festival? </reverse>
<subject> Diabolik: Who Are You? </subject>
**Explanation:** A reverse question like, "What film premiered at the Rome Film
    Festival?" is ambiguous. To ensure a unique answer, we add two key details,
    forming the question: "What film directed by the Manetti Bros. premiered at the
    18th Rome Film Festival?". We specify the "18th" edition because the festival is
     held annually and would otherwise have multiple answers over the years. We add
    "directed by the Manetti Bros." to narrow the options at that specific event,
    ensuring a single valid answer. In contrast, adding the detail "...based on the
    1968 Diabolik comic strip Diabolik, chi sei?" would have been incorrect because
    it shifts the focus. The question would no longer be about the film's premiere
    but would become a simple trivia question about identifying a comic book
    adaptation.

4.
**Context:** WWWQ (99.7 FM) is a commercial radio station carrying a contemporary
    hit radio format known as "Q99.7". Owned by Cumulus Media
**Question:** What company owns the commercial radio station WWWQ?
**OBJECT:** Cumulus Media
**Output:**
REJECT
**Explanation:** A reverse question like, "What commercial radio station is owned by
     Cumulus Media?" is not suitable because the company owns many stations. Adding
    a detail like "...that is known as 'Q99.7'?" or "...that operates on 99.7 FM?"
    shifts the focus. The question is no longer about ownership but about knowing
    the station's slogan or frequency. This turns the added detail into a shortcut.
    Since no additional context can be provided, this item must be rejected.

**Instruction End**

**Please process the following context, question, and its answer (OBJECT entity):**

**Context:** {context}
**Question:** {question}
**OBJECT:** {answer}
```

**Box E.12: Prompt for Verifying a Valid *Reverse* and *Direct* Pair**

You will be provided with a *Direct Question* (with its answer) and a *Reverse
    Question* (with its answer).
Your task is to verify whether this pair represents a *suitable factual evaluation
    pair*.

A fact is defined as a proposition between two entities: a subject and an object.
* In the *Direct Question*, the target is the **Object** entity.
* In the *Reverse Question*, the roles are swapped: the Object of the direct
    question becomes the Subject (or is explicitly mentioned) in the reverse
    question, and the target is the original Subject entity.

The pair is **suitable** if:

1.  The questions represent a valid direct/reverse relationship. The answer (Object)
     of the Direct Question is included in the Reverse Question.
    * *Note:* It is acceptable if the information conveys different hints to resolve
        ambiguity, or if the Object of the direct question is not the main entity,
        topic, or subject of the reverse question, as long as it is explicitly
        present.
2.  The questions are challenging and require specific factual knowledge.
    * The answers are not contained within the questions themselves.
    * The answers cannot be easily guessed using general common sense or linguistic
        clues.
3.  The answers are not time-sensitive.
    * If the fact is subject to change (e.g., revenue, statistics, roles), the
        question must specify a date/year to make the answer static.

The pair is **not suitable** if:

1.  The answer to the Direct Question (the Object) is missing from the Reverse
    Question.
2.  One of the answers can be easily predicted via an educated guess.
3.  One of the questions asks for dynamic information (like statistics or current
    roles) without specifying a timeframe, making the answer subject to decay.

----

### **NOT SUITABLE Examples**

**Direct Question:** What language's complete form was published in the book Toki
    Pona: The Language of Good?
**Answer (Object):** Toki Pona
**Reverse Question:** ...
**Answer (Subject):** ...
**Explanation:** The answer "Toki Pona" is explicitly contained within the book
    title mentioned in the question. Therefore, it is trivial.

**Direct Question:** ...
**Answer (Object):** ...
**Reverse Question:** What is the name of the annual celebration for the date the
    National flag of Canada officially appeared?
**Answer (Subject):** National Flag of Canada Day
**Explanation:** The answer is trivial because it simply repeats the key terms ("
    National flag of Canada") found in the question.

**Direct Question:** What is the median household income in the United States
    according to the Census Bureau?
**Answer (Object):** $67,521
**Reverse Question:** ...
**Answer (Subject):** ...

**Explanation:** The answer is time-sensitive. Without a specific year (e.g., "in 2020"), this value changes over time and is not static.

**Direct Question:** What was the release date of Ice Age: Dawn of the Dinosaurs?
**Answer (Object):** July 1, 2009
**Reverse Question:** What film is the third installment in the Ice Age film series?
**Answer (Subject):** Ice Age: Dawn of the Dinosaurs
**Explanation:** This fails the structural criteria. The answer to the Direct Question ("July 1, 2009") is not included in the Reverse Question.

----

### **SUITABLE Examples**

**Direct Question:** ...
**Answer (Object):** ...
**Reverse Question:** What DVI variant is compatible with VGA?
**Answer (Subject):** DVI-I
**Explanation:** Although the answer shares the term "DVI" with the question, it requires specific technical knowledge to identify the exact variant ("I") compatible with VGA. It is non-trivial.

**Direct Question:** What was the GMV achieved by the Redbus India Private Limited company in 2018?
**Answer (Object):** 50 billion Indian Rupee
**Reverse Question:** ...
**Answer (Subject):** ...
**Explanation:** The answer is not time-sensitive because the context restricts the fact to a specific year (2018), making the value static.

**Direct Question:** With what batting average did Carl Furillo win the batting title?
**Answer (Object):** .344
**Reverse Question:** Who won the 1953 batting title with a .344 average?
**Answer (Subject):** Carl Furillo
**Explanation:** This is suitable. Even though the Direct Answer (.344) is not the main subject of the Reverse Question, it is included in the Reverse Question to resolve ambiguity.

----

### **Required Output Format**

Respond only with "SUITABLE" or "NO".

**Instruction End**

**Please process the following direct and reverse questions and their answers:**

**Direct Question:** {direct}
**Answer (Object):** {object_entity}
**Reverse Question:** {reverse}
**Answer (Subject):** {subject_entity}

**Box E.13: Prompt for Generating the *Natural* Questions**

You will be provided with a question and its answer. Your task is to rephrase the
    question.
The question provided to you is an unambiguous question that has a single, unique,
    and specific answer. It is:

*    **Self-contained:** It does not use pronouns or references that require outside
     context.
*    **Unambiguous and Specific:** Its phrasing is clear, avoids multiple
     interpretations, and specifies the level of detail required in the answer (e.g.,
     "What year...", "What is the name of the condition...").
*    **Minimal:** It includes only the essential details needed to make the question
     unambiguous and ensure there is a single possible answer.

### **Natural Question**

Rephrase the provided question to sound natural and conversational, as if a person
    were genuinely asking.

**Requirements:**

*    The natural question must include all the critical information from the provided
     question, but rewritten in a natural way.
*    **Crucially, the natural question must remain unambiguous and lead to the exact
     same specific answer as the provided question.**
*    To ensure the answer is specific, you might need to add clarifying phrases. For
     example, if the answer is a date, a vague question like "When did X happen?"
     must be rephrased as **"When did X happen? Answer with a date."** If the answer
     is a year, it would be **"When did X happen? Answer with a year."**
*    It is acceptable to use shorter, more common terms (e.g., Ebola instead of Ebola
     virus disease) or common abbreviations if they are widely understood (e.g., USA
     , NFL, LLM).
*    While you should generally not remove information, you **can** discard minor
     words or clauses if they are not necessary for disambiguation and a plausible
     person would consider the canonical and casual questions to be asking for the
     exact same thing. This should be done sparingly, only when it significantly
     improves the naturalness of the question.

----

### **Required Output Format**

You must return *only* a string with the natural question.

----

### **Examples**

1.
*    **Question:** What was the condition that caused Erika Nordby to spend two hours
     without a heartbeat before she was revived?
*    **Answer:** hypothermia
*    **Output:**
What condition caused Erika Nordby to have no heartbeat for two hours?
*Explanation:* The phrase "before she was revived" is omitted as it is implied by
    the fact that she had "no heartbeat for two hours" and the context is about her
    survival. A casual speaker would likely not add this detail.

2.
*    **Question:** What was the release date of When I Look in Your Eyes by Diana
     Krall?
*    **Answer:** June 8, 1999

```
*   **Output:**
When did Diana Krall's album "When I Look in Your Eyes" come out? Answer with a date
    .

3.
*   **Question:** What name did Sauron use in his fair disguise to manipulate
    Celebrimbor?
*   **Answer:** Annatar
*   **Output:**
What name did Sauron use as part of his disguise to trick Celebrimbor?
*Explanation:* The term "manipulated" is rephrased to the more common "trick." The
    sentence structure is changed to a more natural flow, but the key disambiguating
     elements "Sauron," "Celebrimbor," and "disguise" are all retained.

4.
*   **Question:** What is the name the press often uses to refer to the rule
    forbidding the horse-collar tackle from the NFL?
*   **Answer:** The Roy Williams Rule
*   **Output:**
What is the NFL rule that bans horse-collar tackles called?
*Explanation:* This is an example of carefully removing a clause. The phrase "the
    press often uses to refer to" is simplified to "what is the common name for". A
    natural speaker asking this question would assume the common name for the rule
    is the the name the press call the rule, making both questions the same.

**Instruction End**

**Please process the following question and answer:**

**Question:** {question}
**Answer:** {answer}
```

---

**Box E.14: Prompt for Generating the *Contextual* Questions**

```
You will be provided with a context, a question, and an answer. The answer also
    completes the last sentence of the provided context.
Your task is to augment the question with the context.
The question provided to you is an unambiguous question that has a single, unique,
    and specific answer. It is:

*   **Self-contained:** It does not use pronouns or references that require outside
    context.
*   **Unambiguous and Specific:** Its phrasing is clear, avoids multiple
    interpretations, and specifies the level of detail required in the answer (e.g.,
     "What year...", "What is the name of the condition...").
*   **Minimal:** It includes only the essential details needed to make the question
    unambiguous and ensure there is a single possible answer.

### **Contextual Question**

Create a question that tests comprehension within the given context. This is
    achieved by combining the context with the question, making minor edits to
    enhance fluency and minimize redundancy.

The typical process for generating this question is:
1.  Take the entire context, excluding the final incomplete sentence that the answer
     completes.
2.  Append the provided question to the end.
3.  Minor rephrasing of the context or question may be needed to improve the flow
```

    and remove redundancy.

----

### **Required Output Format**

You must return *only* a string with the contextual question.

----

### **Examples**

1.
*   **Context:** Erika Nordby (born February 2000), also known as Baby Erika,
    Miracle Baby and Canada's Miracle Child, is a Canadian originally from Edmonton,
     Alberta known for having been revived after spending two hours without a
    heartbeat due to
*   **Question:** What was the condition that caused Erika Nordby to spend two hours
     without a heartbeat before she was revived?
*   **Answer:** hypothermia
*   **Output:** Erika Nordby (born February 2000), also known as Baby Erika, Miracle
     Baby and Canada's Miracle Child, is a Canadian originally from Edmonton,
    Alberta. What was the condition that caused Erika Nordby to spend two hours
    without a heartbeat before she was revived?

2.
*   **Context:** When I Look in Your Eyes is the fifth studio album by singer Diana
    Krall, released on
*   **Question:** What was the release date of When I Look in Your Eyes by Diana
    Krall?
*   **Answer:** June 8, 1999
*   **Output:** When I Look in Your Eyes is the fifth studio album by singer Diana
    Krall. What was the release date of When I Look in Your Eyes?
*Explanation:* We removed the singer's name from the question to improve fluency, as
     it already appears in the preceding context and its repetition would be
    redundant.

3.
*   **Context:** Celebrimbor is a fictional character in J. R. R. Tolkien's
    legendarium. In Tolkien's stories, Celebrimbor was an elven-smith who was
    manipulated into forging the Rings of Power by the Dark Lord Sauron, in a fair
    disguise and named
*   **Question:** What name did Sauron use in his fair disguise to manipulate
    Celebrimbor?
*   **Answer:** Annatar
*   **Output:** Celebrimbor is a fictional character in J. R. R. Tolkien's
    legendarium. In Tolkien's stories, Celebrimbor was an elven-smith who was
    manipulated into forging the Rings of Power by the Dark Lord Sauron. What name
    did Sauron use in his fair disguise to manipulate Celebrimbor?

4.
*   **Context:** The horse-collar tackle is a gridiron football maneuver in which a
    defender tackles another player by grabbing the back collar or the back-inside
    of an opponent's shoulder pads and pulling the ball carrier directly downward
    violently in order to pull his feet from underneath him. The technique is most
    closely associated with Pro Bowl safety Roy Williams. After being blamed for a
    series of major injuries in the 2004 season , the horse-collar tackle was banned
     from the NFL during the 2005 off-season. The rule forbidding it is often
    referred to in the press as

*   **Question:** What is the name the press often uses to refer to the rule
    forbidding the horse-collar tackle from the NFL?
*   **Answer:** The Roy Williams Rule
*   **Output:** The horse-collar tackle is a gridiron football maneuver in which a
    defender tackles another player by grabbing the back collar or the back-inside
    of an opponent's shoulder pads and pulling the ball carrier directly downward
    violently. The technique is most closely associated with Pro Bowl safety Roy
    Williams and is blamed for a series of major injuries in the 2004 season,
    leading to it being banned. What is the name the press often uses to refer to
    the rule forbidding the horse-collar tackle from the NFL?

**Instruction End**

**Please process the following context, question and answer:**

**Context:** {context}
**Question:** {question}
**Answer:** {answer}

---

**Box E.15: Prompt for Generating Options for Multiple-Choice Questions**

You will be provided with a context, a question, and its gold answer. Your task is
    to generate three additional plausible but incorrect answers to create a
    complete multiple-choice question.
The primary goal is to create challenging distractors that are confusing but
    definitely incorrect.
The provided question should be used as is, and the provided gold answer must be the
     first option in your output.

----

### **Requirements**

Your task is to generate three incorrect but confusing answer options. These
    distractors should make the question challenging. Consider the following
    strategies when creating them:

*   **Plausibility is key:** The incorrect answers should seem like they *could* be
    correct. They must share the same type, category, and granularity as the gold
    answer (e.g., if the answer is a person's name, all options should be names of
    people, if the answer is a city, all options should be cities).
*   **Must be incorrect:** There must be no ambiguity. Only the provided gold answer
     can be the correct answer to the question. The distractors must be factually
    wrong in the given context.
*   **Use contextual hints:** The distractors can be based on other entities or
    details mentioned in the provided context. For example, if the context is a film
    's summary and the question asks for the director, a good distractor would be
    the name of the producer or a lead actor mentioned in the text.
*   **Use similarity:** The distractors can be based on entities with similar names,
     or concepts that are thematically related. For example, if the answer is "The
    Battle of Hastings," a distractor could be "The Battle of Stamford Bridge,"
    which occurred in the same year.
*   **Use popular associations:** The distractors can be other very popular options
    that might come to mind for the general topic, but are incorrect for the
    specific question asked.

----

### **Required Output Format**

```
You must return only the four answer options, each on a new line.
*   The first line must be the original gold answer provided to you.
*   The following three lines must be the new incorrect but confusing answer options
    .
*   Do not include any numbering or lettering (e.g., A., B., C., D.).

Remember to return only the list of strings (each on a new line), without any
    additional text.

**Instruction End**

**Please process the following context, question, and gold answer:**

**Context:** {context}
**Question:** {question}
**Gold answer:** {gold_answer}
```

---

**Box E.16: Prompt for Generating the *Reverse* Task**

```
You will be provided with a question and its corresponding answer.
Your task is to categorize the answer by assigning the most appropriate entity type
    from the list below:

*   **PERSON:** An individual human, whether real or fictional.
*   **ORGANIZATION:** A structured company, institution, or collective with a common
     purpose.
*   **LOCATION:** A specific geographical place, region, or feature on Earth.
*   **EVENT:** A notable occurrence or incident that happens at a particular time
    and place.
*   **WORK_OF_ART:** A specific creative work, such as a book, movie, song, or
    painting.
*   **PRODUCT:** A commercially produced good, food, service, or piece of technology
    .
*   **ORGANISM:** A non-human biological life form, including animals, plants, and
    microbes.
*   **SUBSTANCE:** A material with a distinct physical or chemical composition.
*   **DATE:** A specific point in time, including full dates, years, or historical
    periods.
*   **NUMERIC_VALUE:** A number representing a specific quantity, measurement, or
    other quantifiable value.
*   **CONDITION:** A specific state of being, typically a medical, physical, or
    operational status.
*   **TITLE:** An official position, role, award, or honorific held by a person or
    group.
*   **LANGUAGE:** Natural languages, dialects, writing systems, including phrases.
*   **GROUP:** A collective of people defined by a shared nationality, ethnicity, or
     cultural affiliation.
*   **LAW:** A specific, formally enacted rule, regulation, or statute created by a
    governing body.
*   **IDEA:** An abstract concept, belief system, named phenomenon, or cultural
    practice.
*   **CLASSIFICATION:** A specific type, genre, format, class or list used to
    categorize other entities.
*   **IDENTIFIER:** A unique name, code, or alias used as a formal reference or
    label for another entity
*   **OTHER:** A catch-all category for named entities that do not fit into any of
    the other defined types.

----
```

```
### **Required Output Format**

You must return only a string containing a single entity type (e.g., "GROUP", "DATE
    "). Do not include any additional text or punctuation.

**Instruction End**

Please categorize the answer to the following question:

**Question:** {question}
**Answer:** {answer}
```

## E.3. Response Instructions and Grader Prompts

### Box E.17: Task Instructions

**Completions:**

```
Respond only with the completion of the last sentence
```

**Completions CoT:**

```
Think step by step. Complete the last sentence of the text. End your response with
    Answer: [Your Completion]
```

**Questions:**

```
Respond only with the answer to the question
```

**Questions CoT:**

```
Think step by step. End your response with Answer: [Your Answer]
```

**Multiple-Choice:**

```
Respond only with the letter of the correct answer
```

**Multiple-Choice CoT:**

```
Think step by step. End your response with: Answer: [Letter]
```

### Box E.18: Autorater Prompt for Text Completion Tasks

```
>>> SYSTEM:
You will be provided with an incomplete text (referred to as the `left context`), a
    gold entity that represents the expected continuation or completion (`gold
    completion`), and several predicted completions generated by a model. Your task
    is to evaluate each predicted completion in relation to the gold completion and
    the left context, and assign one of the following labels:
["CORRECT", "PARTIALLY", "INCORRECT", "OTHER"].

First, I will give examples of each grade, and then you will grade a new example.

----

The following are examples of CORRECT predicted completions.

```
Left context: The song "Walk This Way" was peaked at number 10 on the
Gold completion: Billboard Hot 100
Completion 1: on the Billboard Hot 100
Completion 2: Billboard hot 100 in early 1976.
Completion 3: prestigious U.S. chart, Billboard Hot 100 in 1977.
```

```

```

```
Left context: Frequently portions of the term would remain only upon a U.S. senator'
    s death or resignation. From 1914 and the enforcement of the
Gold completion: Seventeenth Amendment to the United States Constitution
Completion 1: Seventeenth Amendment to the United States Constitution, senators have
     been popularly elected.
Completion 2: 17th Amendment, U.S.
Completion 3: Seventeenth Amendment
Completion 4: Amendment XVII
```

```
Left context: Nova Scotia lies in the mid-temperate zone, and although the province
    is almost surrounded by water, the climate is closer to
Gold completion: continental climate
Completion 1: continental than to maritime.
Completion 2: continental than to maritime because the prevailing winds are from the
     continent.
```

```
These predicted completions are all CORRECT because:
    - The predicted completion contains the important information of the gold
       completion.
    - The predicted completion does not contain any information that contradicts the
       gold completion or the left context.
    - The gold completion may appear anywhere within the predicted completion as
       long as it clearly plays the same semantic role as the focus of the
       completion.
    - Minor omissions to the gold completion are allowed if the missing information
       is either easily inferable from the left context or not critical to
       identifying the gold completion (e.g., dropping a redundant modifier or
       title).
    - Only the semantic meaning matters: The completion may contain a variant,
       abbreviation, or alternative name for the gold completion. Differences in
       capitalization, punctuation, spelling, or minor grammatical variation (e.g.,
       articles, pluralization) are allowed.
    - Completions may contain small factual inaccuracies or hallucinated details, as
       long as they do not contradict the left context.
    - Hedging and guessing are permissible, provided that the gold completion is
       included and the response does not contradict it or the left context.

----

The following are examples of INCORRECT predicted completions.
```

```
Left context: The Grand Caravan was reported missing, and a search was made, ending
    with the wreckage being found the next day, with all
Gold completion: 10
Completion 1: nine occupants of the aircraft dead.
Completion 2: five people on board deceased.
Completion 3: 11
```

```
Left context:  Andrew Paul Gosden (born 10 July 1993) disappeared from Central
    London on
Gold completion: 14 September 2007
Completion 1: 19 March 2007, aged 14.
Completion 2: 2007-09-15
```

```
Left context:  The Toplica Uprising was a mass uprising by Serbian rebels against
    the Bulgarian occupation forces that took place in Bulgarian-occupied Serbia
    during the First World War. The revolt was supported by Serbian guerrilla
    fighters known as
Gold completion: Chetniks
Completion 1: Komitadjis
Completion 2: Komitadjis who were the enemies of the Chetnik bands
```

These predicted completions are all INCORRECT because:
    - A factual statement in the predicted completion contradicts the left context
        or the gold completion.
    - Incorrect statements that have some hedging (e.g., "it is possible that", "
        although i'm not sure, i think") are also considered incorrect.

----

The following are examples of PARTIALLY predicted completions.

```
Left context:  IgA vasculitis (HSP), previously known as Henoch-Schonlein purpura,
    is an autoimmune disease that most commonly affects children. With kidney
    involvement, there may be a loss of small amounts of blood and protein in the
    urine (hematuria and proteinuria), but this usually goes unnoticed; in a small
    proportion of cases, the kidney involvement proceeds to
Gold completion: chronic kidney disease
Completion 1: end-stage kidney disease
Completion 2: kidney disease
Completion 3: chronic kidney disease and kidney failure - This is CORRECT.
```

```
Left context:  Dopamine dysregulation syndrome (DDS) is a dysfunction of the reward
    system observed in some individuals taking dopaminergic medications for an
    extended length of time. It typically occurs in people with Parkinson's disease
    (PD) or restless legs syndrome (RLS) who have taken
Gold completion: dopamine agonist medications
Completion 1: dopaminergic medication
Completion 2: dopaminergic medication for more than six months.
```

```
Left context:  Franki Raffles (17 October 1955 - 6 December 1994) was an English
    feminist and a
Gold completion: documentary photographer
Completion 1: photographer
Completion 2: photographer and artist.
Completion 3: professional artist and photographer.
```

These predicted answers are all PARTIALLY because:
    - The predicted completion contains a more general or less specific version of
        the gold completion (e.g., "dopamine agonist medications" is a specific type
         of "dopaminergic medication").
    - The predicted completion identifies the core entity and the primary meaning of
         the gold completion, but omits a descriptive detail or modifier (e.g., "
        photographer" instead of "documentary photographer").
    - The predicted completion is closely related to the gold completion and
        represents a common or expected outcome or progression, even if not strictly
         identical in every case (e.g., "end-stage kidney disease" as a severe form
        often associated with "chronic kidney disease").

```
----

The following are examples of OTHER predicted completions.

```
Left context:  Dirk Koperlager van Erp (1862-1933) was a Dutch American artisan,
Gold completion: coppersmith
Completion 1: noted for his distinctive work.
Completion 2: designer, and teacher.
Completion 3: I don't know.
Completion 4: I need more context about Dirk Koperlager.
Completion 5: Without researching the web, I cannot answer this. However, I can tell
    you that Dirk Koperlager van Erp was an artisan.
Completion 6: who became known for his innovative metalwork, particularly his use of
    silver.
```

```
Left context:  T33 and CP3 are disability sport classification for disability
    athletics. Athletes in this class have moderate
Gold completion: quadriplegia
Completion 1: to severe hypertonia, ataxia, or athetosis
Completion 2: athetosis, ataxia, or dystonia.
Completion 3: or good functional strength and range of motion in their trunk
```

These predicted completions are all OTHER because:
    - The important information in the gold completion is not included in the
        predicted completion.
    - No statements in the predicted completion contradict the gold completion or
        the left context.

----

Also note the following things:
- For completions where the gold completion is a number, the predicted completion
    needs to be correct to the last significant figure in the gold completion. For
    example, consider the left context "The number of citations the Transformer
    paper has is?" with gold completion "120k".
    - Predictions "120k", "124k", and 115k" are all CORRECT.
    - Predictions "100k" and "113k" are INCORRECT.
    - Predictions "around 100k" and "more than 50k" are considered OTHER because
        they neither confirm nor contradict the gold completion.
- Do not punish predicted completions if they omit information that would be clearly
    inferred from the left context. For example, consider the left context: "The
    paper A pretrainer's guide to training data: Measuring the effects of data age,
    domain coverage, quality, & toxicity" won the NAACL '24", and the gold
    completion "Outstanding Paper Award". The predicted completion "Outstanding
    Paper" would be considered CORRECT, because "award" is presumed in the left
    context.
- Do not punish typos in people's names if it's clearly the same name.
    - For example, if the gold completion is "Hyung Won Chung", you can consider the
        following predicted completions as correct: "Hyoong Won Choong", "Hyungwon
        Chung", or "Hyun Won Chung".

----

**Required Output Format**

Your entire response must be structured using the following tags:
<completion_{i}>the grade of Completion {i}</completion_{i}>
```

```
>>> USER:
Here is a new example. Use the output format above to grade each completion with
    either "CORRECT", "INCORRECT", "PARTIALLY", or "OTHER". Don't apologize or
    correct yourself if there was a mistake; we are just trying to grade the
    completions.

```
Left context:  {left_context}
Gold completion: {gold_target}
{completions}
```

---

**Box E.19: Autorater Prompt for Closed-book QA Tasks**

```
>>> SYSTEM:
You will be provided with a question, a gold target, and several predicted answers
    generated by a model. Your task is to evaluate each predicted answer in relation
     to the gold target, and assign one of the following labels:
["CORRECT", "PARTIALLY", "INCORRECT", "OTHER"].

First, I will give examples of each grade, and then you will grade a new example.

----

The following are examples of CORRECT predicted answers.

```
Question: What chart did Aerosmith's "Walk This Way" hit number 10 on?
Gold target: Billboard Hot 100
Answer 1: On the Billboard Hot 100
Answer 2: Billboard hot 100 in early 1976.
Answer 3: The prestigious U.S. chart, Billboard Hot 100 in 1977.
```

```
Question: What was enforced from 1914 that affected the election of U.S. senators?
Gold target: Seventeenth Amendment to the United States Constitution
Answer 1: Seventeenth Amendment to the United States Constitution, senators have
    been popularly elected.
Answer 2: 17th Amendment, U.S.
Answer 3: Seventeenth Amendment
Answer 4: Amendment XVII
```

```
Question: Nova Scotia lies in the mid-temperate zone. What type of climate is Nova
    Scotia's climate closer to, despite the province being almost surrounded by
    water?
Gold target: continental climate
Answer 1: Continental than to maritime.
Answer 2: Continental than to maritime because the prevailing winds are from the
    continent.
```

These predicted answers are all CORRECT because:
    - The predicted answer contains the important information of the gold target.
    - The predicted answer does not contain any information that contradicts the
        gold target or the question.
    - The gold target may appear anywhere within the predicted answer as long as it
        clearly plays the same semantic role as intended by the question.
    - Minor omissions to the gold target in the predicted answer are allowed if the
```

```
        missing information is either easily inferable from the question or not
        critical to identifying the gold target (e.g., dropping a redundant modifier
         or title).
    - Only the semantic meaning matters: The predicted answer may contain a variant,
        abbreviation, or alternative name for the gold target. Differences in
        capitalization, punctuation, spelling, or minor grammatical variation (e.g.,
        articles, pluralization) are allowed.
    - Predicted answers may contain small factual inaccuracies or hallucinated
        details, as long as they do not contradict the question.
    - Hedging and guessing are permissible, provided that the gold target is
        included and the predicted answer does not contradict it or the question.

----

The following are examples of INCORRECT predicted answers.

```
Question: How many people were on Bering Air Flight 445 when it crashed on February
    6, 2025?
Gold target: 10
Answer 1: Nine occupants of the aircraft dead.
Answer 2: Five people on board deceased.
Answer 3: 11
```

```
Question:  Andrew Paul Gosden (born 10 July 1993) disappeared from Central London.
    On what date did this disappearance occur?
Gold target: 14 September 2007
Answer 1: 19 March 2007, aged 14.
Answer 2: 2007-09-15
```

```
Question:  What was the name of the Serbian guerrilla fighters who supported the
    Toplica Uprising?
Gold target: Chetniks
Answer 1: Komitadjis
Answer 2: Komitadjis who were the enemies of the Chetnik bands
```

These predicted answers are all INCORRECT because:
    - A factual statement in the predicted answer contradicts the question or the
        gold target.
    - Incorrect answers that have some hedging (e.g., "it is possible that", "
        although i'm not sure, i think") are also considered incorrect.

----

The following are examples of PARTIALLY predicted answers.

```
Question:  What is a rare but serious kidney complication of IgA vasculitis?
Gold target: chronic kidney disease
Answer 1: End-stage kidney disease
Answer 2: Kidney disease
Answer 3: Chronic kidney disease and kidney failure - This is CORRECT.
```

```
Question:  What drugs can lead to Dopamine dysregulation syndrome in patients with
    Parkinson's or restless legs syndrome?
Gold target: dopamine agonist medications
```
```

```
Answer 1: Dopaminergic medication
Answer 2: Dopaminergic medication for more than six months.
```

```
Question:  What was Franki Raffles' profession?
Gold target: documentary photographer
Answer 1: Photographer
Answer 2: Photographer and artist.
Answer 3: Professional artist and photographer.
```

These predicted answers are all PARTIALLY because:
   - The predicted answer contains a more general or less specific version of the
       gold target (e.g., "dopamine agonist medications" is a specific type of "
       dopaminergic medication").
   - The predicted answer identifies the core entity and the primary meaning of the
        gold target, but omits a descriptive detail or modifier (e.g., "
        photographer" instead of "documentary photographer").
   - The predicted answer is closely related to the gold target and represents a
       common or expected outcome or progression, even if not strictly identical in
        every case (e.g., "end-stage kidney disease" as a severe form often
       associated with "chronic kidney disease").

----

The following are examples of OTHER predicted completions.

```
Question:  Dirk Koperlager van Erp (1862-1933) was a Dutch American artisan. What
    type of artisan was he?
Gold completion: coppersmith
Answer 1: Noted for his distinctive work.
Answer 2: Designer, and teacher.
Answer 3: I don't know.
Answer 4: I need more context about Dirk Koperlager.
Answer 5: Without researching the web, I cannot answer this. However, I can tell you
     that Dirk Koperlager van Erp was an artisan.
Answer 6: He became known for his innovative metalwork, particularly his use of
    silver.
```

```
Question:  What condition is described as moderate in athletes in the T33 and CP3
    disability sport classification?
Gold completion: quadriplegia
Answer 1: Severe hypertonia, ataxia, or athetosis
Answer 2: Athetosis, ataxia, or dystonia.
Answer 3: Good functional strength and range of motion in their trunk
```

These predicted completions are all OTHER because:
   - The important information in the gold target is not included in the predicted
       answer.
   - No statements in the predicted answer contradict the gold target or the
       question.

----

Also note the following things:
- For answers where the gold target is a number, the predicted answer needs to be
    correct to the last significant figure in the gold target. For example, consider
     the question "What is the number of citations the Transformer paper has?" with

```
        gold target "120k".
        - Predictions "120k", "124k", and 115k" are all CORRECT.
        - Predictions "100k" and "113k" are INCORRECT.
        - Predictions "around 100k" and "more than 50k" are considered OTHER because
            they neither confirm nor contradict the gold target.
- Do not punish predicted answers if they omit information that would be clearly
    inferred from the question. For example, consider the left context: "What NAACL
    '24 award did the paper A pretrainer's guide to training data: Measuring the
    effects of data age, domain coverage, quality, & toxicity" won?", and the gold
    target "Outstanding Paper Award". The predicted answer "Outstanding Paper" would
     be considered CORRECT, because "award" is presumed in the question.
- Do not punish typos in people's names if it's clearly the same name.
        - For example, if the gold target is "Hyung Won Chung", you can consider the
            following predicted answers as correct: "Hyoong Won Choong", "Hyungwon Chung
            ", or "Hyun Won Chung".

----

**Required Output Format**

Your entire response must be structured using the following tags:
<answer_{i}>the grade of Answer {i}</answer_{i}>

>>> USER:
Here is a new example. Use the output format above to grade each answer with either
    "CORRECT", "INCORRECT", "PARTIALLY", or "OTHER". Don't apologize or correct
    yourself if there was a mistake; we are just trying to grade the answers.

```

Question:  {question}
Gold target: {gold_target}
{answers}
```
```

