# OpenReview forum: "Empty Shelves or Lost Keys? Recall Is the Bottleneck for Parametric Factuality"
_ICML.cc/2026/Conference — ICML 2026 regular_

### Official Review · Reviewer_qc4a · 2026-02-20

**Soundness:** 4
**Presentation:** 4
**Significance:** 4
**Originality:** 4
**Overall Recommendation:** 5
**Confidence:** 5

**Summary:**

This paper proposes to disentangle factuality errors in LLMs as encoding errors (i.e., the LM does not have this information in its parameters) and recall errors (i.e., the LM has the information, but cannot elicit it due to various factors, e.g., the reversal curse). The former is solvable only with pre-training interventions (e.g., scaling model size or pre-training data), whereas the latter can be addressed through post-training (e.g., learning how to elicit the right answers that are already acquired earlier during pre-training).

The authors introduce a concept called knowledge profiling, which categorizes any atomic fact into one of five profiles: encoding failure, recall failure, direct recall (a fact is encoded and can be recalled directly), recall with thinking (a fact is encoded and can be recalled only with thinking), and inference without encoding (a fact is not encoded but is known when thinking is enabled, e.g., resolvable via chain-of-thought or reasoning). Given this taxonomy, the authors present WikiProfile, a benchmark for evaluating and profiling LLM factual knowledge, which consists of 2150 facts extracted from natural documents, paired with 10 single-hop questions each. When evaluated on 13 LLMs, this paper finds that frontier models exhibit near-saturated encoded knowledge, with recall being the driving problem.

Further analysis reveals that (1) tail knowledge exhibits a small encoding gap compared to head knowledge, but a large recall gap; (2) LLMs struggle with generating responses to reverse questions, but not when framed as a multiple-choice task. Finally, the authors show that with thinking enabled, 40-65% of encoded facts are recoverable, suggesting that when it comes to factual tasks, thinking functions more as a recall mechanism.

**Compliance With Llm Reviewing Policy:**

Affirmed.

**Final Justification:**

I have a pretty positive view of the paper, and my original positive score remains unchanged. I had a few minor questions that the authors answered completely during rebuttal.

**Key Questions For Authors:**

*Questions:*

- The WikiProfile per-category distribution in Figure 9 is interesting to me. Do different categories lead to different per-profile breakdowns, or are they all roughly similar? For example, I would expect philosophy & religion (which would have more long-tail knowledge) to have a higher rate of recall errors than, say, everyday life.
- Retrieval-augmented generation (RAG) is a common way to improve test-time factuality, and allows for small LMs to catch up to large LMs. Under WikiProfile, would incorporating RAG help reduce recall error? Further, would the effects of RAG be complementary with the thinking mode?
- Another type of test-time scaling is best-of-N sampling (sampling multiple times under high diversity conditions), and I wonder if this could also help improve recall recovery?  The benefit to this is that you would not need an LM equipped with thinking mode capabilities.
- From Figure 12, we see that for higher $p$, enabling the thinking mode eventually leads to lower correct $\geq p$ for all the Gemma 3 models. Why do you think that is? Is it because Gemma 3 maybe a smaller or less performant model than the others, which are all frontier models?
- How expensive is it to evaluate a model or method on WikiProfile? It would be useful to have some idea of cost estimates. Given that evaluation is prompting-based and requires $n$ samples per question, evaluation could be costly and untenable within an academic budget.


Other comments:
- In Figure 3, there is a type in Box D.10 (“Filerting”)

**Limitations:**

yes

**Strengths And Weaknesses:**

Soundness: This submission is technically sound, and I think the claims are all well-supported. One minor quibble I have is that for Gemini 2.5 Pro w/ thinking is used as graders for WikiProfile response evaluation, and there may be a slight bias when evaluating the Gemini line of models. Would WikiProfile evaluation still result in similar trends using different LLMs for autorating?

Presentation: I found this paper well-written and easy to follow, and well-situated among the recent literature. In particular, I like how Figs.1-3 convey the various contributions of this work in an intuitive and easily accessible way. One suggestion is that for the thinking-enabled studies, it would be nice to see examples of the thinking traces that lead to the right answer (for instances that the LM would not get correct otherwise).


Significance: I like this paper a lot! Hallucination in LLMs and ensuring factual generations are still very active areas of research. Given that pre-training is becoming increasingly computationally expensive (as training tokens and model parameters continue to scale), it is encouraging to see that most factual errors may be recoverable via post-training. I think this paper has broad impact and can open up (1) further methodological research on eliciting factuality in instruction-tuning / RL stages; (2) further analysis of these error modes, particularly in the more challenging long-form generation setting.


Originality: I think this paper is original and contributes a new understanding of factuality that can drive progress in the field. WikiProfile seems like a useful contribution that many researchers may use in the future. In particular, identifying the thinking mode as a possible way to recover from recall errors is very timely, given that a lot of works have focused on unlocking thinking capabilities for reasoning-intensive (not knowledge-intensive) tasks.

---

> ### Author Rebuttal · Authors · 2026-03-29
>
> We thank the reviewer for their feedback and are especially glad they liked the paper.
>
> **W1 (Different Autoraters):**
> In Appendix D.3 (page 22), we compared the Gemini-2.5-Pro grader with an alternative GPT-5-based grader,  and observed 98.2% agreement. Most disagreements were on OTHER and PARTIALLY labels (see Figure 15, and the error analysis on pages 27–28), which are excluded from question grading. Therefore, we believe any bias is minimal and will not affect our overall conclusions. We will clarify this in the final manuscript.
>
> **Suggestion (thinking traces):**
> We agree that such qualitative analysis would be valuable. Thank you for this suggestion. Unfortunately, the raw traces are not disclosed for closed-source LLMs, which are the primary focus of our work. The only available signal is the "thought summary", which we found neither informative nor revealing of meaningful patterns. In the final revision, we will include illustrative examples using GPT-4.1 and Gemma3, where raw traces are available.
>
> **Q1 (Categories Analysis):** Performance on different categories varies, but this may be confounded by factors such as popularity or answer type (e.g., the People category often involves date-related questions, which are generally more challenging for LLMs). To avoid over-interpreting confounded results, we decided not to include such analysis. We include a table with knowledge measures on different categories:
> https://anonymous.4open.science/r/icml26_rebuttal-D5A9/categories_table.md
>
> **Q2 (Incorporating RAG):** RAG improves factuality by retrieving external knowledge, bypassing the encoding/recall distinction, while thinking specifically improves parametric recall of encoded facts. RAG therefore does not fix recall failures, but sidesteps them. Moreover, parametric knowledge remains important even in RAG systems: knowing what to search, formulating effective queries, and integrating retrieved information all rely on encoded knowledge. Thus, RAG and thinking are complementary. On WikiProfile, RAG would be highly effective since the source documents are known, but this is expected, as the benchmark is designed to measure parametric knowledge. Extending the setup to study how the two (RAG, thinking) interact in agentic systems is an exciting direction for future work. We thank the reviewer for this insightful question, and we will further extend our discussion on RAG (see the discussion section).
>
> **Q3 (Best-of-N):**
> Our evaluation uses "some form of" Best-of-N: each question is answered with N=8 samples, and the score is computed as correct/(correct+incorrect), requiring a value above 0.5 (a correct majority). This differs from standard majority voting (since aggregation is done at the grading level rather than directly over answers). Our analysis of thinking (Appendix B, Figure 12) also provides some insight into its relationship with this "grade-level" Best-of-N, although this is not a formal analysis. Intuitively, the correct@p metric can be interpreted as follows: correct@0.125 without thinking approximates the performance of an oracle Best-of-N selector (i.e., one that always picks a correct answer if present), while correct@0.5 with thinking reflects portions of questions where thinking is more likely to produce a correct answer. For some LLMs, correct@0.125 without thinking exceeds correct@0.5 with thinking, suggesting that strong Best-of-N strategies could partially replace thinking. The non-trivial question, however, is which strategy to use, though the potential is clear.
>
> **Q4 (Gemma3 and Thinking Degradation):** There are two possible explanations for why Gemma3 benefits less from thinking. First, as suggested, Gemma3 models may simply be smaller or less capable. Second, they are not “thinking-optimized” models, meaning they do not undergo a dedicated post-training phase for reasoning. This likely limits the effectiveness of thinking (CoT).
>
> **Q5 (Costs):** The evaluation is indeed computationally expensive. Profiling a single "pro" model can cost around 500 USD.
> Generating all responses (8 samples X 2,150 facts X 10 questions) across five GPT models costs approximately
> 1,200 USD, almost entirely due to thinking tokens (the inputs and outputs are relatively short). We will include a detailed cost breakdown in the final version. That said, costs can be reduced substantially. Excluding multiple-choice questions reduces costs by ~40% (4 out of 10 tasks). Additionally, since we did not observe meaningful differences across question phrasings, this could yield a further ~20% reduction. Finally, reducing the number of samples per question (e.g., from 8 to 4) would further decrease costs, at the expense of higher variance (see Appendix D.4).
>
> Once again, we thank the reviewer for their thoughtful feedback and for the positive and encouraging review.

---

> > ### Author Rebuttal · Reviewer_qc4a · 2026-04-01
> >
> > Thanks for the response! My concerns have been addressed; I have also read the other concerns raised by my fellow reviewers and found the authors' responses convincing. As such, I've elected to keep my original, positive score.

---

### Official Review · Reviewer_gAs5 · 2026-03-05

**Soundness:** 3
**Presentation:** 3
**Significance:** 2
**Originality:** 3
**Overall Recommendation:** 4
**Confidence:** 4

**Summary:**

The paper proposes a new benchmark, WikiProfile, designed to characterize LLM’s knowledge retrieval in a fine-grained manner. The benchmark contains 2150 facts, each paired with 10 questions querying that fact in diverse formats. The paper provides an in-depth, comprehensive evaluation of 13 LLMs (querying them a total of 4M times!), and analyze the results. The paper is well-organized, well-written, mostly clear, and experimental decisions are sound. Despite these values, I am reluctant to recommend the paper's acceptance as the study does not really provide new insights, rather a comprehensive evidence of previously identified aspects about knowledge recall. I also think some of the claims are a bit not well supported, please see weaknesses.

**Compliance With Llm Reviewing Policy:**

Affirmed.

**Final Justification:**

This is a nicely presented paper, and I like some ideas from it. I enjoyed reading it overall.

I still find the definition of recall a bit challenging -- if scoped too tight (i.e., currently as 4 questions per question with minimal paraphrases), it doesn't really mean "recall" in a colloquial sense ppl use. People mean "recall" when LLMs can *use* that knowledge in diverse contexts and settings, including knowledge propagation. But then, if you really try to measure "recall", it's impossible to operationalize.... Reading the paper again after the author's response, I think they tried what they could do (though could have been a bit clearer in intro/etc that they are measuring "recall" in a very limited manner.

I also have an issue with some of the claims, given the scope of the experiments. They only evaluate with Wikipedia facts (which are somewhat prominent facts to begin with), and making broader claims based on it (i.e., encodings are saturated in frontier models) not very convincing.

Having said that, I think the paper has some good ideas and well-executed, so increasing my score a little bit. I hope authors will discuss these two points a bit more carefully in the final revision.

**Key Questions For Authors:**

* For “encoding”, what’s the statistics of the length of “left context”?
* To make a claim that model “encodes” these facts, it *assumes* models are trained on these Wikipedia pages? I think this is reasonable assumption but has to be clarified, as we do not know data many of these models are trained on.

**Limitations:**

yes

**Strengths And Weaknesses:**

Strength:
* The paper is clearly written and provides a nice overview about the related work.
* The data construction methodology is clearly explained and mostly sound.
* The evaluation is thorough and carefully designed, such as employing multiple LLM as a judge from different family.
* I appreciate the clear notation and definition of concepts in this work, especially about the knowledge “Encoding”.
Weakness:
* While I appreciate the efforts, I am not sure I’m fully convinced about the methodology. I think “Encoding” is well-defined and measured, but the way “knowledge” is defined, I do not think it is *possible* to measure, as you cannot come up with all possible questions about any fact.
* Novelty: While this provides a thorough study on multiple, SoTA LLMs, I am not sure what is the new insight it provides. Many of the key arguments (e.g., rare facts are harder to recall, reverse fact are harder to retrieve, models perform good at multiple choice questions but cannot answer free-form answers, thinking improves model performances, etc) are already well studied in the field. I understand that the paper formalizes and provides stronger evidence for such claims, but not sure what new behavior about LLMs it reveals.
* Section 2.2. is not very clear for me. Could you describe how measuring of these encoding failure, recall failure, etc are implemented? It should be linked to Questions generated in Figure 3.
* I do not agree with framing that “rare facts are encoded”. Figure 5 shows that rare facts are indeed encoded to a much lesser degree than popular facts. Figure 16 (in appendix) further shows if you go to smaller model like Gemma3-4B (almost half as well encoded, 56.2 vs. 31.7).
* I do not think the authors should make generalization claim to other domains (Section 3) of using the pipeline, as Wikipedia is fairly unique domain with encyclopedic facts. Many other corpora is much more tricky to deal with, with high dependency on temporal aspects, etc. To include such claim, the authors should test pipeline in other domains.
minor: * Figure 3 typo “Clarrification”

---

> ### Author Rebuttal · Authors · 2026-03-29
>
> We thank the reviewer for their thorough review and for recognizing the clarity, soundness, and rigor of our study. We believe the concern about novelty stems from a misunderstanding of our central contribution, particularly regarding the term “recall”.
>
> **W2 (Novelty):** In prior work, “recall” is measured by accuracy. In our work, we introduce a framework that separates encoding failures from recall failures, which prior evaluations conflate. “Recall” here refers to knowing an *encoded fact*. While we agree that the individual observations (rare facts being harder, the reversal curse, MC vs. CBQA, thinking helps) are known, our primary contribution is not to replicate them but to reinterpret them through the encoding-vs-recall lens, which is itself novel, and our work is the first to study this.
>
> Our main insights are about encoding being saturated and recall being the bottleneck, and that scaling mainly improves encoding (all are novel to the best of our knowledge). The other insights mentioned are about understanding when recall is challenging, and we would like to clarify how they are reinterpreted. For rare facts, this reinterpretation has direct practical consequences: prior work has naturally framed these as encoding/memorization problems (e.g., Kandpal et al., 2023; Mallen et al., 2023; Berglund et al., 2024), motivating pre-training interventions. We show they are predominantly recall failures, shifting the implied intervention toward post-training.
>
> LLMs struggle with reverse questions and (separately) perform better on multiple-choice is not our core insight. We demonstrate that in multiple-choice, there is no reversal curse, reframing the curse as a recall asymmetry rather than a lack of bidirectional knowledge.
>
> Additionally, prior evidence for thinking's benefits has centered on reasoning-intensive tasks; our results suggest thinking can function as a recall mechanism, a novel mechanistic insight with practical implications for when and why to deploy it. We show that if a fact is encoded, there is a 40–65% chance of recovery with thinking, but if not encoded, less than 20%.
>
> **W1 (Measurability of knowledge):** We appreciate this thoughtful concern. Our definition of *knows*  requires correctness across two key axes, relational direction (direct/reverse) and phrasing. This follows standard practices in NLP evaluation (e.g., memorization, robustness). We do not claim that Q_f is complete; we claim that it captures the relevant dimensions of factual recall. Extending it is a natural direction for future work, and our setup readily supports it. Note that we show (§5.2, Appendix C.3) that phrasing has a negligible impact, reducing concerns about exhaustive coverage.
>
> **W3 (Clarity of Section 2.2):** We thank the reviewer for flagging this. We will revise Section 2.2 to explicitly link it to Figure 3 as suggested. A fact is encoded if the model can reproduce it in a pre-training-like context (encoding tasks: Proposition Completion, Contextual Question in Figure 3); a fact is known if the model can answer questions about it across different phrasings and directions (Direct Question, reverse Question, Direct Natural Question, reverse Natural Question in Figure 3). Failure modes follow directly: if a fact is neither encoded nor known, it is an encoding failure; if it is encoded but not known, it is a recall failure.
>
> **W4 (Encoding of rare facts):** The claim is scoped to frontier LLMs, as stated in the paper (R313 "this pattern holds for all frontier LLMs"). For these models, the encoding gap between popular and rare facts is small (Δ=5–8 points) while the recall gap is substantially larger (Δ=19–25 points), supporting our framing. The reviewer is correct that for smaller models like Gemma3-4B, encoding gaps are much larger; this is consistent with our broader finding that scaling primarily improves encoding. We will make the frontier-model scope of this claim more explicit to avoid confusion.
>
> **W5 (Generalization of pipeline to other domains):** We thank the reviewer for raising this concern. We will soften this claim and reframe it as future work.
>
> **Q1 (Left-context length statistics):** The median is 100 words, mean is 116 words (±56 std). We will add full statistics to the paper.
>
> **Q2 (Does encoding assume training on Wikipedia?):** If a model was not exposed to the fact during pre-training, the chance of it passing our encoding tasks is low, so the assumption is implicit in the results rather than explicitly stated in the definition. As you suggested, we selected Wikipedia as a trusted source that pre-training pipelines are likely to prioritize (lines 170–173).
>
> We thank the reviewer for their time and thoughtful feedback and hope our responses have addressed their concerns. We hope the reviewer will consider revising their score in light of the clarifications above, particularly regarding the novelty of the encoding-vs-recall framework and its practical implications.

---

> > ### Author Rebuttal · Reviewer_gAs5 · 2026-04-03
> >
> > Thank you for the response.
> >
> > W2: I disagree with the characterization that prior work has not studied "encoding being saturated and recall being the bottleneck". The reversal curse line of work points this exact thing, i.e., some facts can be recovered with certain queries but not with other semantically equivalent queries, suggests that encoded facts cannot be recalled.
> >
> > W1: I do not think "recall", the way it is defined in this paper, can be concretely measured in a principled manner. The line 167 defines for "a set of questions associated with a fact f". This is a very wide set, which should also includes facts that are impacted by propagation of the fact that is tested (e.g., if the fact is John Doe is born in San Francisco, would the question "Was John Doe born in the USA?" be included?). The format of the question / as well as instructions prepended to them, will also impact the model behaviors significantly. It is just very hard to formally define such a set, and while I appreciate the efforts, the rough conceptualization here can mislead the readers. There are so many ways a fact can be queried, and how it is getting queried will impact whether it can be recalled or not.
> >
> > W4: I was responding to section right above Figure 5, where statements about rare facts are not scoped to premier models, but I agree other parts of the paper the claim was scoped well.

---

> > > ### Author Response · Authors · 2026-04-03
> > >
> > > Thanks for engaging in the rebuttal.
> > >
> > > **W2:** We agree that we are not the first to observe that some facts can be recalled under certain queries but not under other semantically equivalent ones, and we are transparent about this in our paper. The reversal curse is one manifestation of this phenomenon. However, this is not what we claim is our contribution. We propose a principled and operational framework to measure and quantify encoding and recall separately and, to our knowledge, conduct the first large-scale study of this distinction on frontier LLMs. This enables a key finding that prior work could not establish: for frontier LLMs, encoding is near-saturated, while recall is the dominant bottleneck.
> > > We believe this is both novel and timely. As our results show, encoding is not saturated for smaller or earlier-generation models, but becomes so at the frontier. This distinction has practical implications, suggesting a shift in focus from knowledge acquisition to knowledge utilization. In contrast, prior work, such as the reversal curse, identifies specific failure modes but does not quantify or disentangle encoding and recall failures.
> > >
> > > **W1:** We agree that “knowledge” cannot be exhaustively defined over all possible queries, and our definition is therefore intentionally operational and bounded (otherwise, it is not practically measurable). Therefore, our goal is to capture the prevalent axes of variation for single-hop subject–object facts. We focus on directionality and phrasing, and empirically show that phrasing has a negligible effect across models, suggesting that additional phrasing variations would not change the conclusions. We agree that other axes, such as granularity (e.g., derived or entailed facts), could affect recall. However, these often correspond to multi-hop inference rather than direct instantiation of the extracted fact, and are beyond the scope of this work. More broadly, any benchmark requires operational choices. To avoid potential misinterpretation, we will clarify in the paper that our definition targets these axes for single-hop facts and does not aim to fully characterize all forms of knowledge.
> > >
> > > **W4:** Thanks for pointing to the less-scoped part. We will revise the bold title of the paragraph.
> > >
> > > We thank the reviewer for the thoughtful discussion and for helping us clarify these points.

---

### Official Review · Reviewer_NGyj · 2026-03-30

**Soundness:** 3
**Presentation:** 4
**Significance:** 3
**Originality:** 3
**Overall Recommendation:** 4
**Confidence:** 3

**Summary:**

The paper presents a comprehensive analysis on the failures of fact recall of LLMs. The authors propose an evaluation framework, create a benchmark dataset, and demonstrate that failure to recall encoded knowledge is the primary reason for factuality errors, and that thinking helps LLM to recall knowledge.

**Compliance With Llm Reviewing Policy:**

Affirmed.

**Final Justification:**

Rebuttal has addressed my concerns; I increased the originality score a bit.

**Key Questions For Authors:**

1. See the weakness about the originality. Please expand on how the claimed two contributions are original given the cited & missed related works.

2. Regarding the motivation of creating WikiProfile compared to reusing existing factuality benchmarks. In introduction, the rationales is explained as "Unlike other benchmarks that rely on synthetic or schema-restricted knowledge-base triplets, [the dataset] enables evaluation in more realistic settings, where facts are expressed indirectly, and may involve complex, context-dependent relations"

Why can't benchmarks in prior works enable "complex / context-dependent" evaluation of factuality? Please explain more in detail with exemplars.

**Limitations:**

I did not find a dedicated limitations section. However, the paper adequately the address all in-scope research questions proposed in the paper. I encourage the authors to include a limitation section on which parts can break out of the studied scope in the next version of the paper.

**Strengths And Weaknesses:**

**Soundness - strength**

The claims in the paper are well supported by the experiments. Experiments are well-designed to answer research questions such as the predominant cause of factuality errors, effect of thinking on fact recall, impact of reversed questions, and what makes fact more susceptible to recall failure. Experiment results are significant enough to support the claims.

**Presentation - strength**

The paper is clearly written and well-structured. A minor issue is to define "thinking" more formally and in the main text, rather than in the footnote.

**Significance - neutral**

The issue studied in paper, factuality, is crucial in many LLM applications; insights to cause and failure modes in factuality is relevant and significant to the community. However, the paper does not present new actionable insights on improving factuality of LLMs beyond "turn on thinking during inference", which I believe is already a prevalent practice when users care about factuality, except for highly cost and latency sensitive scenarios.

**Originality - weakness**

1. The authors include "we show that the bottleneck for factuality is recall, not encoding" in the "contribution" paragraph. Can I understand the claim as "knowledge is often learned, but recall depends on how we elicit the knowledge (like with better prompting)"? If so, I wonder whether the claim can be considered a novel contribution given the cited papers like Berglund et al. 2024 (reversal curse).

I also noticed the authors did not cite a well-cited work in the field: [1] Jiang et al. How Can We Know What Language Models Know? in TACL 2020, which I suspect is making the same claim.

2. In addition, regarding the contribution of "showing thinking helps recover otherwise inaccessible knowledge" in the introduction - doesn't quite seem accurate, as the authors included in the related works section, their are prior works that show CoT improves factuality. The difference explained there seems to be the more precise statement of the novelty ("Our work offers a unifying perspective: thinking functions as a recall mechanism, not only a reasoning mechanism")

Possibly missing reference [2] Wang et al. Unveiling Factual Recall Behaviors of Large Language Models through Knowledge Neurons, which makes a claim that CoT improves factuality by enhancing reliability.

3. Despite the concerns above, the clean behavioral framework to study the problem makes the paper still original and offer useful new insights compared to prior works.

---

> ### Author Rebuttal · Authors · 2026-03-30
>
> We thank the reviewer for their careful and thorough reading, and for the encouraging assessment of the paper's soundness and presentation. We address each of the concerns and questions below.
>
> **Originality**: We thank the reviewer for the careful reading and the suggested references. The reviewer's interpretation ("knowledge is often learned, but recall depends on how we elicit it") is partially correct, but we believe it undersells the contribution. Prior work has shown that better prompting can surface knowledge that naive prompting misses, treating this as a *measurement problem*. Our contribution is different: we ask why prompting affects recall, and whether failures stem from knowledge never being encoded, or from knowledge being encoded but inaccessible. This reframing turns a measurement question into a *mechanistic* one, and the answer has direct implications for where to intervene.
>
> Jiang et al. (2020), which we will cite, is a good example of this gap: it studies prompt sensitivity to better surface knowledge, but operates entirely within the *recall regime*, never asking whether a failure might be at the encoding level. Our framework explicitly separates *encoding* from *recall*, and shows empirically that for frontier LLMs, encoding is near-saturated, so the focus should shift to recall. So our core contribution is not claiming that “better prompting can elicit more gains”, but to diagnose knowledge and failure modes of LLMs, which sheds new light on prior findings.
>
> Berglund et al. (2024) interpret the reversal curse as missing bidirectional knowledge; we show that models can recognize the correct reverse answer in multiple-choice but fail to generate it. This suggests the curse is a *recall asymmetry*, not a knowledge gap, with different implications for how to fix it. Indeed, we show the reversal curse is almost entirely eliminated when thinking is enabled, further supporting this interpretation.
>
> Wang et al. (2024) show CoT helps factuality in *multi-hop reasoning*, where reasoning is naturally helpful. We show thinking improves recall of *single-hop* facts where no reasoning is needed, pointing to a distinct mechanism of *recall facilitation* that prior work does not identify. We show that thinking primarily functions as a recall mechanism: it recovers 40–65% of encoded facts, but <20% when the fact is not encoded (Figure 11 in the appendix, will be moved to the main body). Accordingly, thinking selectively targets recall failures. We also demonstrate that thinking improves performance in harder recall settings (e.g., reverse and rare facts). We will cite both papers and clarify these distinctions in the related work section.
>
> **Significance**: We appreciate this concern and agree that "turn on thinking" alone would not be a significant contribution. We would like to clarify that thinking in our work serves as a *proof of concept* that recall failures are recoverable, not as the final recommendation. The deeper contribution is identifying recall as the tractable bottleneck and demonstrating that interventions targeting it can succeed (like thinking). This opens a concrete research agenda: developing post-training methods (e.g., RL) that improve *knowledge utilization* rather than *knowledge acquisition*. This is a fundamentally different than what prior work points to.
>
> **WikiProfile vs. existing benchmarks**: WikiProfile was designed to support our profiling setup, which has two structural requirements that existing benchmarks cannot satisfy: (i) encoding tasks require  the original pre-training document to construct the left context (ii) determining relational direction (direct vs. reverse) requires knowing how the fact is expressed in that document. Benchmarks based on knowledge-base triplets (e.g., PopQA, LAMA) do not provide this. Beyond this, such benchmarks are inherently limited to atomic, predefined relations (e.g., BornIn, ChildOf). Many real-world facts are richer, implicit propositions that depend on context and integrate multiple pieces of information. For example: "What condition caused Erika Nordby to spend two hours without a heartbeat before she was revived?" (Appendix, line 861) is a fact that cannot be reduced to a single predefined relation. Finally, to our knowledge, WikiProfile is the first benchmark to study the reversal curse beyond a constrained set of relations; Berglund et al. explicitly identify this as future work.
>
> **Presentation**: We agree and will move the definition of "thinking" from the footnote to the main text.
>
> **Limitations**: We thank the reviewer for this suggestion and will add a dedicated limitations section discussing what falls outside our studied scope, including smaller models, non-Wikipedia domains, and long-form generation settings.
>
> We once again thank the reviewer for the thoughtful feedback, and hope our responses have addressed their concerns. We hope the reviewer will consider revising their score in light of our clarifications.

---

> > ### Author Rebuttal · Reviewer_NGyj · 2026-04-02
> >
> > Thank you for the response. I updated the originality score a bit. Please update the writing accordingly in the final version of the paper.

---

### Decision · Program_Chairs · 2026-04-30

**Decision:**

Accept (regular)

**Comment:**

This paper introduces WikiProfile, a new benchmark and behavioral framework for profiling factual knowledge in LLMs by distinguishing whether errors stem from missing encoding ("empty shelves") or limited recall access ("lost keys"), evaluated across 4M responses from 13 LLMs. Its main contribution is an encoding-vs-recall reframing that empirically demonstrates encoding is near-saturated in frontier models (95–98% for GPT-5 and Gemini-3) while recall remains the primary bottleneck, and that inference-time thinking can recover a substantial fraction of recall failures.

**Strengths:**
- The paper is technically sound, well-written, and well-organized, with claims that are well-supported by a thorough and carefully designed evaluation at large scale (4M responses, 13 LLMs, multiple LLM-as-judge graders from different families), as noted by all three reviewers.
- The behavioral framework separating encoding from recall is clean and provides a useful lens for reinterpreting factuality errors, which Reviewer qc4a highlighted as offering "a new understanding of factuality that can drive progress in the field."
- The work is timely and has broad potential impact, opening methodological directions for post-training interventions targeting knowledge utilization rather than acquisition, and WikiProfile is likely to be a useful community resource.

**Weaknesses:**
- Concerns about originality relative to prior work (Reviewer NGyj and gAs5), particularly regarding the reversal curse literature (Berglund et al.) and prompt-sensitivity work (Jiang et al. 2020), with some arguing that individual observations (rare facts are harder, reverse questions are harder, thinking helps) are already established.
- The measurability and operationalization of "knowledge/recall" is contested (Reviewer gAs5), as the set of possible queries for a given fact cannot be exhaustively defined, and the current scoping (directionality + limited phrasings) may be too narrow to match colloquial notions of recall.
- Some claims may be overgeneralized given the scope of experiments, particularly around encoding being "saturated" (based only on Wikipedia facts, which are prominent) and the applicability of the pipeline to non-Wikipedia domains with temporal or non-encyclopedic content.
- Limited new actionable insights beyond "enable thinking at inference" (Reviewer NGyj), and minor presentation issues such as the definition of "thinking" being relegated to a footnote, a missing dedicated limitations section, and possible autorater bias from using Gemini-2.5-Pro as the judge.

All three reviewers recommended acceptance, with Reviewer NGyj's concerns fully resolved after rebuttal and Reviewer gAs5 partially resolved but still raising their score. The consensus is that despite legitimate concerns about the novelty framing and the operational definition of recall, the paper is technically solid, clearly written, and provides a useful reframing with sufficient empirical support to be valuable to the community.